# EF21 WITH BELLS & WHISTLES: PRACTICAL ALGORITHMIC EXTENSIONS OF MODERN ERROR FEEDBACK

## ABSTRACT

First proposed by Seide et al. (2014) as a heuristic, error feedback (EF) is a very popular mechanism for enforcing convergence of distributed gradient-based optimization methods enhanced with communication compression strategies based on the application of contractive compression operators. However, existing theory of EF relies on very strong assumptions (e.g., bounded gradients), and provides pessimistic convergence rates (e.g., while the best known rate for EF in the smooth nonconvex regime, and when full gradients are compressed, is $O(1/T^{2/3})$, the rate of gradient descent in the same regime is $O(1/T)$). Recently, Richtárik et al. (2021) (2021) proposed a new error feedback mechanism, EF21, based on the construction of a Markov compressor induced by a contractive compressor. EF21 removes the aforementioned theoretical deficiencies of EF and at the same time works better in practice. In this work we propose six practical extensions of EF21, all supported by strong convergence theory: partial participation, stochastic approximation, variance reduction, proximal setting, momentum and bidirectional compression. Several of these techniques were never analyzed in conjunction with EF before, and in cases where they were (e.g., bidirectional compression), our rates are vastly superior.

## 1 INTRODUCTION

In this paper, we consider the nonconvex distributed/federated optimization problem of the form

$$\min_{x \in \mathbb{R}^d} \left\{ f(x) \stackrel{\text{def}}{=} \frac{1}{n} \sum_{i=1}^{n} f_i(x) \right\}, \tag{1}$$

where $n$ denotes the number of clients/workers/devices/nodes connected with a server/master and client $i$ has an access to the local loss function $f_i$ only. The local loss of each client is allowed to have the online/expectation form

$$f_i(x) = \mathbb{E}_{\xi_i \sim \mathcal{D}_i} \left[ f_{\xi_i}(x) \right], \tag{2}$$

or the finite-sum form

$$f_i(x) = \frac{1}{m} \sum_{j=1}^{m} f_{ij}(x). \tag{3}$$

Problems of this structure appear in federated learning (Konečný et al., 2016; Kairouz, 2019), where training is performed directly on the clients' devices. In a quest for state-of-the-art performance, machine learning practitioners develop elaborate model architectures and train their models on enormous data sets. Naturally, for training at this scale to be possible, one needs to rely on distributed computing (Goyal et al., 2017; You et al., 2020). Since in recent years remarkable empirical successes were obtained with massively over-parameterized models (Arora et al., 2018), which puts an extra strain on the communication links during training, recent research activity and practice focuses on developing distributed optimization methods and systems capitalizing on (deterministic or randomized) *lossy communication compression* techniques to reduce the amount of communication traffic.

A compression mechanism is typically formalized as an operator $\mathcal{C} : \mathbb{R}^d \mapsto \mathbb{R}^d$ mapping hard-to-communicate (e.g., dense) input messages into easy-to-communicate (e.g., sparse) output messages. The operator is allowed to be randomized, and typically operates on models Khaled & Richtárik

(2019) or on gradients Alistarh et al. (2017); Beznosikov et al. (2020), both of which can be described as vectors in $\mathbb{R}^d$. Besides sparsification (Alistarh et al., 2018), typical examples of useful compression mechanisms include quantization (Alistarh et al., 2017; Horváth et al., 2019a) and low-rank approximation (Vogels et al., 2019; Safaryan et al., 2021).

There are two large classes of compression operators often studied in the literature: i) *unbiased* compression operators $\mathcal{C}$, meaning that there exists $\omega \geq 0$ such that

$$\mathbb{E}\left[\mathcal{C}(x)\right] = x, \qquad \mathbb{E}\left[\|\mathcal{C}(x) - x\|^2\right] \leq \omega\|x\|^2, \qquad \forall x \in \mathbb{R}^d; \tag{4}$$

and ii) *biased* compression operators $\mathcal{C}$, meaning that there exists $0 < \alpha \leq 1$ such that

$$\mathbb{E}\left[\|\mathcal{C}(x) - x\|^2\right] \leq (1 - \alpha)\|x\|^2, \qquad \forall x \in \mathbb{R}^d. \tag{5}$$

Note that the latter "biased" class contains the former one, i.e., if $\mathcal{C}$ satisfies (4) with $\omega$, then a scaled version $(1 + \omega)^{-1}\mathcal{C}$ satisfies (5) with $\alpha = 1/(1+\omega)$. While distributed optimization methods with unbiased compressors (4) are well understood (Alistarh et al., 2017; Khirirat et al., 2018; Mishchenko et al., 2019; Horváth et al., 2019b; Li et al., 2020; Li & Richtárik, 2021a; Li & Richtárik, 2020; Islamov et al., 2021; Gorbunov et al., 2021), *biased* compressors (5) are significantly harder to analyze. One of the main reasons behind this is rooted in the observation that when deployed within distributed gradient descent in a naive way, biased compresors may lead to (even exponential) divergence (Karimireddy et al., 2019; Beznosikov et al., 2020). *Error Feedback* (EF) (or *Error Compensation* (EC))—a technique originally proposed by Seide et al. (2014)—emerged as an empirical fix of this problem. However, this technique remained poorly understood until very recently.

Although several theoretical results were obtained supporting the EF framework in recent years (Stich et al., 2018; Alistarh et al., 2018; Beznosikov et al., 2020; Gorbunov et al., 2020; Qian et al., 2020; Tang et al., 2020; Koloskova et al., 2020), they use strong assumptions (e.g., convexity, bounded gradients, bounded dissimilarity), and do not get $\mathcal{O}(1/\alpha T)$ convergence rates in the smooth nonconvex regime. Very recently, Richtárik et al. (2021) proposed a new EF mechanism called EF21, which uses standard smoothness assumptions only, and also enjoys the desirable $O(1/\alpha T)$ convergence rate for the nonconvex case (in terms of number of communication rounds $T$ this matches the best-known rate $\mathcal{O}((1+\omega/\sqrt{n})/T)$ obtained by Gorbunov et al. (2021) using unbiased compressors), improving the previous $O(1/(\alpha T)^{2/3})$ rate of the standard EF mechanism (Koloskova et al., 2020).

## 2   OUR CONTRIBUTIONS

While Richtárik et al. (2021) provide a new theoretical SOTA for error feedback based methods, the authors only study their EF21 mechanism in a pure form, without any additional "bells and whistles" which are of importance in practice. In this paper, we aim to push the EF21 framework beyond its pure form by extending it in several directions of high theoretical and practical importance. In particular, we further enhance the EF21 mechanism with the following six useful and practical algorithmic extensions: *stochastic approximation*, *variance reduction*, *partial participation*, *bidirectional compression*, *momentum*, and *proximal (regularization)*. We do not stop at merely proposing these algorithmic enhancements: we derive *strong convergence results for all of these extensions*. Several of these techniques were never analyzed in conjunction with the original EF mechanism before, and in cases where they were, our new results with EF21 are vastly superior. See Table 1 for an overview of our results. In summary, our results constitute the new algorithmic and theoretical state-of-the-art in the area of error feedback.

We now briefly comment on each extension proposed in this paper:

◇ **Stochastic approximation.** The vanilla EF21 method requires all clients to compute the exact/full gradient in each round. While Richtárik et al. (2021) do consider a stochastic extension of EF21, they do not formalize their result, and only consider the simplistic scenario of uniformly bounded variance, which does not in general hold for stochasticity coming from subsampling (Khaled & Richtárik, 2020). However, exact gradients are not available in the stochastic/online setting (2), and in the finite-sum setting (3) it is more efficient in practice to use subsampling and work with stochastic gradients instead. In our paper, we extend EF21 to a more general stochastic approximation framework than the simplistic framework considered in the original paper. Our method is called EF21-SGD (Algorithm 2); see Appendix D for more details.

| Setup | Method | Citation | Compl. (NC) | Compl. (PL) | Comment |
|---|---|---|---|---|---|
| Full grads | EF21 | Richtárik et al. (2021) | $\frac{1}{\alpha\varepsilon^2}$ | $\frac{1}{\alpha\mu}$ | |
| Stoch. grads | Choco-SGD | Koloskova et al. (2020) | $\frac{1}{\varepsilon^2} + \frac{G}{\alpha\varepsilon^3} + \frac{\sigma^2}{n\varepsilon^4}$ | N/A | $\|\nabla f_i(x)\| \le G$ |
| | EF21-SGD | Richtárik et al. (2021) | $\frac{1}{\alpha\varepsilon^2} + \frac{\sigma^2}{\alpha^3\varepsilon^4}$ | $\frac{1}{\alpha\mu} + \frac{\sigma^2}{\mu^2\alpha^3\varepsilon}$ | UBV (Ex. 1) |
| | EF21-SGD | **NEW** | $\frac{1}{\alpha\varepsilon^2} + \frac{1+\Delta^{\text{inf}}}{\alpha^3\varepsilon^4}$ | $\frac{1}{\alpha\mu} + \frac{1+\Delta^{\text{inf}}}{\mu^2\alpha^3\varepsilon}$ | IS (Ex. 2) |
| | EF21-PAGE | **NEW** | $\frac{\sqrt{m}+1/\alpha}{\varepsilon^2} + m$ | $\frac{\sqrt{m}+1/\alpha}{\mu} + m$ | $f_i(x) = \frac{1}{m}\sum\limits_{j=1}^{m} f_{ij}(x)$ |
| PP | EF21-PP | **NEW** | $\frac{1}{p\alpha\varepsilon^2}^{(1)} + \frac{1}{\alpha\varepsilon^2}$ | $\frac{1}{p\alpha\mu}^{(1)} + \frac{1}{\alpha\mu}$ | Full grads |
| BC | DoubleSqueeze | Tang et al. (2020) | $\frac{1}{\varepsilon^2} + \frac{\Delta}{\varepsilon^3} + \frac{\sigma^2}{n\varepsilon^4}$ | N/A | $\mathbb{E}\left[\|\mathcal{C}(x) - x\|\right] \le \Delta$ |
| | EF21-BC | **NEW** | $\frac{1}{\alpha_w\alpha_M\varepsilon^2}$ | $\frac{1}{\alpha_w\alpha_M\mu}$ | Full grads |
| Mom. | M-CSER | Xie et al. (2020)[2] | $\frac{1}{\varepsilon^2} + \frac{G}{(1-\eta)\alpha\varepsilon^3}$ | N/A | $\|\nabla f_i(x)\| \le G$ |
| | EF21-HB | **NEW** | $\frac{1}{\varepsilon^2}\left(\frac{1}{1-\eta} + \frac{1}{\alpha}\right)$ | N/A | Full grads |
| Prox | EF21-Prox | **NEW** | $\frac{1}{\alpha\varepsilon^2}$ | $\frac{1}{\alpha\mu}^{(3)}$ | Full grads |

[1] Red term = number of communication rounds, blue term = expected number of gradient computations per client.

[2] Xie et al. (2020) consider Nesterov's momentum. Moreover, they analyzed the version with stochastic gradients, bidirectional compression and local steps. However, the derived result is not better than state-of-the-art ones with either stochastic gradients or bidirectional compression. Therefore, to maintain the table compact, we do not include the results of Xie et al. (2020) in the other parts of the table.

[3] This result is obtained under the generalized PŁ-condition for composite optimization problems (see Assumption 5 from Appendix I.2).

Table 1: Summary of the state-of-the-art complexity results for finding an $\varepsilon$-**stationary point**, i.e., such a point $\hat{x}$ that $\mathbb{E}\left[\|\nabla f(\hat{x})\|^2\right] \le \varepsilon^2$, for generally non-convex functions and an $\varepsilon$-**solution**, i.e., such a point $\hat{x}$ that $\mathbb{E}\left[f(\hat{x}) - f(x^*)\right] \le \varepsilon$, for functions satisfying PŁ-condition using error-feedback type methods. By (computation) complexity we mean the average number of (stochastic) first-order oracle calls needed to find an $\varepsilon$-stationary point ("Compl. (NC)") or $\varepsilon$-solution ("Compl. (PŁ)"). Removing the terms colored in blue from the complexity bounds shown in the table, one can get communication complexity bounds, i.e., the total number of communication rounds needed to find an $\varepsilon$-stationary point ("Compl. (NC)") or $\varepsilon$-solution ("Compl. (PŁ)"). Dependences on the numerical constants, "quality" of the starting point, and smoothness constants are omitted in the complexity bounds. Moreover, dependencies on $\log(1/\varepsilon)$ are also omitted in the column "Compl. (PŁ)". Abbreviations: "BC" = bidirectional compression, "PP" = partial participation; "Mom." = momentum; $T$ = the number of communications rounds needed to find an $\varepsilon$-stationary point; $\overline{\#\text{grads}}$ = the number of (stochastic) first-order oracle calls needed to find an $\varepsilon$-stationary point. Notation: $\alpha$ = the compression parameter, $\alpha_w$ and $\alpha_M$ = the compression parameters of worker and master nodes respectively for EF21-BC, $\sigma^2 = \frac{1}{n}\sum_{i=1}^{n}\sigma_i^2$ (see Example 1), $\Delta^{\text{inf}} = f^{\text{inf}} - \frac{1}{n}\sum_{i=1}^{n}\frac{1}{m_i}\sum_{j=1}^{m_i} f_{ij}^{\text{inf}}$ (see Example 2), $p$ = probability of sampling the client in EF21-PP, $\eta$ = momentum parameter. To the best of our knowledge, combinations of error feedback with partial participation (EF21-PP) and proximal versions of error feedback (EF21-Prox) were never analyzed in the literature.

◇ **Variance reduction.** As mentioned above, EF21 relies on full gradient computations at all clients. This incurs a high or unaffordable computation cost, especially when local clients hold large training sets, i.e., if $m$ is very large in (3). In the finite-sum setting (3), we enhance EF21 with a variance reduction technique to reduce the computational complexity. In particular, we adopt the simple and efficient variance-reduced method PAGE (Li et al., 2021; Li, 2021b) (which is optimal for solving problems (3)) into EF21, and call the resulting method EF21-PAGE (Algorithm 3). See Appendix E for more details.

◇ **Partial participation.** The EF21 method proposed by Richtárik et al. (2021) requires *full participation* of clients for solving problem (1), i.e., in each round, the server needs to communicate with all $n$ clients. However, full participation is usually impractical or very hard to achieve in massively distributed (e.g., federated) learning problems (Konečný et al., 2016; Cho et al., 2020; Kairouz, 2019; Li & Richtárik, 2021b; Zhao et al., 2021). To remedy this situation, we propose a *partial participation* (PP) variant of EF21, which we call EF21-PP (Algorithm 4). See Appendix F for more details.

◇ **Bidirectional compression.** The vanilla EF21 method only considers *upstream* compression of the messages sent by the clients to the server. However, in some situations, downstream communication is also costly (Horváth et al., 2019a; Tang et al., 2020; Philippenko & Dieuleveut, 2020). In order to cater to these situations, we modify EF21 so that the server can also optionally compresses messages before communication. Our master compression is intelligent in that it employs the Markov compressor proposed in EF21 to be used at the devices. The proposed method, based on *bidirectional compression*, is EF21-BC (Algorithm 5). See Appendix G for more details.

| Update | Method | Alg. # | $c_i^t$ | Comment |
|---|---|---|---|---|
| $x^{t+1} = x^t - \gamma g^t,$ $g^t = \frac{1}{n}\sum_{i=1}^n g_i^t,$ $g_i^{t+1} = g_i^t + c_i^t$ | EF21 | Alg. 1 | $\mathcal{C}(\nabla f_i(x^{t+1}) - g_i^t)$ | |
| | EF21-SGD | Alg. 2 | $\mathcal{C}(\hat{g}_i(x^{t+1}) - g_i^t)$ | $\hat{g}_i(x^{t+1})$ satisfies As. 2 |
| | EF21-PAGE | Alg. 3 | $\mathcal{C}(v_i^{t+1} - g_i^t)$ | $b_i^t \sim \text{Be}(p),$ $v_i^{t+1} = \nabla f_i(x^{t+1}),\ \text{if } b_i^t = 1,$ $v_i^{t+1} = v_i^t + \frac{1}{\tau_i}\sum_{j \in I_i^t} \nabla f_{ij}(x^{t+1})$ $-\frac{1}{\tau_i}\sum_{j \in I_i^t} \nabla f_{ij}(x^t),\ \text{if } b_i^t = 0,$ $I_i^t$ is a minibatch, $|I_i^t| = \tau_i$ |
| | EF21-PP | Alg. 4 | $\mathcal{C}(\nabla f_i(x^{t+1}) - g_i^t)$ $0$ | if $i \in S_t$ if $i \notin S_t$ |
| $x^{t+1} = x^t - \gamma g^t,$ $g^{t+1} = g^t + b^{t+1},$ $b^{t+1} = \mathcal{C}_M(\widetilde{g}^{t+1} - g^t),$ $\widetilde{g}^{t+1} = \frac{1}{n}\sum_{i=1}^n \widetilde{g}_i^{t+1},$ $\widetilde{g}_i^{t+1} = \widetilde{g}_i^t + c_i^t$ | EF21-BC | Alg. 5 | $\mathcal{C}_w(\nabla f_i(x^{t+1}) - \widetilde{g}_i^t)$ | Master broadcasts $b^{t+1}$; $\mathcal{C}_w$ is used on the workers' side, $\mathcal{C}_M$ is used on the master's side |
| $x^{t+1} = x^t - \gamma v^t,$ $v^{t+1} = \eta v^t + g^{t+1},$ $g^{t+1} = \frac{1}{n}\sum_{i=1}^n g_i^{t+1},$ $g_i^{t+1} = g_i^t + c_i^t$ | EF21-HB | Alg. 6 | $\mathcal{C}(\nabla f_i(x^{t+1}) - g_i^t)$ | $\eta \in [0,1)$ – momentum parameter |
| $x^{t+1} = \text{prox}_{\gamma r}(x^t - \gamma g^t),$ $g^{t+1} = \frac{1}{n}\sum_{i=1}^n g_i^{t+1},$ $g_i^{t+1} = g_i^t + c_i^t$ | EF21-Prox | Alg. 7 | $\mathcal{C}(\nabla f_i(x^{t+1}) - g_i^t)$ | For problem (6); $\text{prox}_{\gamma r}(x)$ is defined in (91) |

Table 2: Description of the methods developed and analyzed in the paper. For the ease of comparison, we also provide a description of EF21. In all methods only compressed vectors $c_i^t$ are transmitted from workers to the master and the master broadcasts non-compressed iterates $x^{t+1}$ (except EF21-BC, where the master broadcasts compressed vector $b^{t+1}$). Initialization of $g_i^0$, $i = 1, \ldots, n$ can be arbitrary (possibly randomized). One possible choice is $g_i^0 = \mathcal{C}(\nabla f_i(x^0))$. The pseudocodes for each method are given in the appendix.

⋄ **Momentum.** A very successful and popular technique for enhancing both optimization and generalization is momentum/acceleration (Polyak, 1964; Nesterov, 1983; Lan & Zhou, 2015; Allen-Zhu, 2017; Lan et al., 2019; Li, 2021a). For instance, momentum is a key building block behind the widely-used Adam method (Kingma & Ba, 2014). In this paper, we add the well-known (Polyak) heavy ball momentum (Polyak, 1964; Loizou & Richtárik, 2020) to EF21, and call the resulting method EF21-HB (Algorithm 6). See Appendix H for more details.

⋄ **Proximal setting.** It is common practice to solve *regularized* versions of empirical risk minimization problems instead of their vanilla variants (Shalev-Shwartz & Ben-David, 2014). We thus consider the composite/regularized/proximal problem

$$\min_{x \in \mathbb{R}^d}\left\{ \Phi(x) \overset{\text{def}}{=} \frac{1}{n}\sum_{i=1}^n f_i(x) + r(x) \right\}, \tag{6}$$

where $r(x) : \mathbb{R}^d \to \mathbb{R} \cup \{+\infty\}$ is a regularizer, e.g., $\ell_1$ regularizer $\|x\|_1$ or $\ell_2$ regularizer $\|x\|_2^2$. To broaden the applicability of EF21 to such problems, we propose a proximal variant of EF21 to solve the more general composite problems (6). We call this new method EF21-Prox (Algorithm 7). See Appendix I for more details.

Our theoretical complexity results are summarized in Table 1. In addition, we also analyze EF21-SGD, EF21-PAGE, EF21-PP, EF21-BC under Polyak-Łojasiewicz (PŁ) condition (Polyak, 1963; Lojasiewicz, 1963) and EF21-Prox under the generalized PŁ-condition (Li & Li, 2018) for composite optimization problems. Due to space limitations, we defer all the details about the analysis under the PŁ-condition to the appendix and provide only simplified rates in Table 1. We comment on some preliminary experimental results in Section 5. More experiments including deep learning experiments are presented in Appendix A.

## 3 METHODS

Since our methods are modifications of EF21, they share many features, and are presented in a unified way in Table 2. At each iteration of the proposed methods, worker $i$ computes the compressed vector $c_i^t$ and sends it to the master. The methods differ in the way of computing $c_i^t$ but have similar (in case

of EF21-SGD, EF21-PAGE, EF21-PP – exactly the same) update rules to the one of EF21:

$$x^{t+1} = x^t - \gamma g^t, \qquad g_i^{t+1} = g_i^t + c_i^t, \qquad g^{t+1} = \frac{1}{n} \sum_{i=1}^{n} g_i^{t+1} = g^t + \frac{1}{n} \sum_{i=1}^{n} c_i^t. \qquad (7)$$

The pseudocodes of the methods are given in the appendix. Below we briefly describe each method.

⋄ EF21-SGD: **Error feedback and** SGD**.** EF21-SGD is essentially EF21 but instead of the full gradients $\nabla f_i(x^{t+1})$, workers compute the stochastic gradients $\hat{g}_i(x^{t+1})$, and use them to compute $c_i^t = \mathcal{C}(\hat{g}_i(x^{t+1}) - g_i^t)$. Despite the seeming simplicity of this extension, it is highly important for various applications of machine learning and statistics where exact gradients are either unavailable or prohibitively expensive to compute.

⋄ EF21-PAGE: **Error feedback and variance reduction.** In the finite-sum regime (3), variance reduced methods usually perform better than vanilla SGD in many situations (Gower et al., 2020). Therefore, for this setup we modify EF21 and combine it with variance reduction. In particular, this time we replace $\nabla f_i(x^{t+1})$ in the formula for $c_i^t$ with the PAGE estimator (Li et al., 2021) $v_i^{t+1}$. With (typically small) probability $p$ this estimator equals the full gradient $v_i^{t+1} = \nabla f_i(x^{t+1})$, and with probability $1 - p$ it is set to

$$v_i^{t+1} = v_i^t + \frac{1}{\tau_i} \sum_{j \in I_i^t} \left( \nabla f_{ij}(x^{t+1}) - \nabla f_{ij}(x^t) \right),$$

where $I_i^t$ is a minibatch of size $\tau_i$. Typically, the number of data points $m$ owned by each client is large, and $p \leq 1/m$ when $\tau_i \equiv 1$. As a result, computation of full gradients rarely happens during the optimization procedure: on average, once in every $m$ iterations only. Although it is possible to use other variance-reduced estimators like in SVRG or SAGA, we use the PAGE-estimator: unlike SVRG or SAGA, PAGE is optimal for smooth nonconvex optimization, and therefore gives the best theoretical guarantees (we have obtained results for both SVRG and SAGA and indeed, they are worse, and hence we do not include them).

Notice that unlike VR-MARINA (Gorbunov et al., 2021), which is a state-of-the-art distributed optimization method designed specifically for unbiased compressors and which *also* uses the PAGE-estimator, EF21-PAGE does not require the communication of full (non-compressed) vectors at all. This is an important property of the algorithm since, in some distributed networks, and especially when $d$ is very large, as is the case in modern over-parameterized deep learning, full vector communication is prohibitive. However, unlike the rate of VR-MARINA, the rate of EF21-PAGE does not improve with increasing $n$. This is not a flaw of our method, but rather an inevitable drawback of distributed methods that rely on *biased* compressors such as Top-$k$.

⋄ EF21-PP: **Error feedback and partial participation.** The extension of EF21 to the case of partial participation of the clients is mathematically identical to EF21 up to the following change: $c_i^t = 0$ for all clients $i \notin S_t \subseteq \{1, \dots, n\}$ that are *not* selected for communication at iteration $t$. In practice, $c_i^t = 0$ means that client $i$ does not take part in the $t$-th communication round. Here the set $S_t \subseteq \{1, \dots, n\}$ is formed randomly such that $\mathbf{Prob}(i \in S_t) = p_i > 0$ for all $i = 1, \dots, n$.

⋄ EF21-BC: **Error feedback and bidirectional compression.** The simplicity of the EF21 mechanism allows us to naturally extend it to the case when it is desirable to have efficient/compressed communication between the clients and the server in *both directions*. At each iteration of EF21-BC, clients compute and send to the master node $c_i^t = \mathcal{C}_w(\nabla f_i(x^{t+1}) - \widetilde{g}_i^t)$ and update $\widetilde{g}_i^{t+1} = \widetilde{g}_i^t + c_i^t$ in the usual way, i.e., workers apply the EF21 mechanism. The key difference between EF21 and EF21-BC is that the master node in EF21-BC also uses this mechanism: it computes and broadcasts to the workers the compressed vector $b^{t+1} = \mathcal{C}_M(\widetilde{g}^{t+1} - g^t)$ and updates $g^{t+1} = g^t + b^{t+1}$, where $\widetilde{g}^{t+1} = \frac{1}{n} \sum_{i=1}^{n} \widetilde{g}_i^{t+1}$. Vector $g^t$ is maintained by the master *and* workers. Therefore, the clients are able to update it via using $g^{t+1} = g^t + b^{t+1}$ and compute $x^{t+1} = x^t - \gamma g^t$ once they receive $b^{t+1}$.

⋄ EF21-HB: **Error feedback with momentum.** We consider classical Heavy-ball method (Polyak, 1964) with EF21 estimator $g^t$:

$$x^{t+1} = x^t - \gamma v^t, \quad v^{t+1} = \eta v^t + g^{t+1}, \quad g_i^{t+1} = g_i^t + c_i^t, \quad g^{t+1} = \frac{1}{n} \sum_{i=1}^{n} g_i^{t+1} = g^t + \frac{1}{n} \sum_{i=1}^{n} c_i^t.$$

The resulting method is not better than EF21 in terms of the complexity of finding $\varepsilon$-stationary point, i.e., momentum does not improve the theoretical convergence rate. Unfortunately, this is common

issue for a wide range of results for momentum methods Loizou & Richtárik (2020). However, it is important to theoretically analyze momentum-extensions such as EF21-HB due to their importance in practice and generalization behaviour.

⋄ EF21-Prox: **Error feedback for composite problems.** Finally, we make EF21 applicable to the composite optimization problems (6) by simply taking the prox-operator from the right-hand side of the $x^{t+1}$ update rule (7): $x^{t+1} = \text{prox}_{\gamma r}(x^t - \gamma g^t) = \arg\min_{x \in \mathbb{R}^d} \{\gamma r(x) + \|x - x^t + \gamma g^t\|^2/2\}$. This trick is simple, but, surprisingly, EF21-Prox is the first distributed method with error-feedback that provably converges for composite problems (6).

# 4 THEORETICAL CONVERGENCE RESULTS

In this section, we formulate a single corollary derived from the main convergence theorems for our six enhancements of EF21, and formulate the assumptions that we use in the analysis. The complete statements of the theorems and their proofs are provided in the appendices. In Table 1 we compare our new results with existing results.

## 4.1 ASSUMPTIONS

In this subsection, we list and discuss the assumptions that we use in the analysis.

### 4.1.1 GENERAL ASSUMPTIONS

To derive our convergence results, we invoke the following standard smoothness assumption.

**Assumption 1** (Smoothness and lower boundedness). *Every $f_i$ has $L_i$-Lipschitz gradient, i.e.,* $\|\nabla f_i(x) - \nabla f_i(y)\| \le L_i \|x - y\|$ *for all $i \in [n], x, y \in \mathbb{R}^d$, and $f^{\text{inf}} \stackrel{def}{=} \inf_{x \in \mathbb{R}^d} f(x) > -\infty$.*

We also assume that the compression operators used by all algorithms satisfy the following property.

**Definition 1** (Contractive compressors). *We say that a (possibly randomized) map $\mathcal{C} : \mathbb{R}^d \to \mathbb{R}^d$ is a* contractive compression operator, *or simply* contractive compressor, *if there exists a constant $0 < \alpha \le 1$ such that*

$$\mathbb{E}\left[\|\mathcal{C}(x) - x\|^2\right] \le (1 - \alpha)\|x\|^2, \qquad \forall x \in \mathbb{R}^d. \tag{8}$$

We emphasize that we do *not* assume $\mathcal{C}$ to be unbiased. Hence, our theory works with the Top-$k$ (Alistarh et al., 2018) and the Rank-$r$ (Safaryan et al., 2021) compressors, for example.

### 4.1.2 ADDTIONAL ASSUMPTIONS FOR EF21-SGD

We analyze EF21-SGD under the assumption that local stochastic gradients $\nabla f_{\xi_{ij}^t}(x^t)$ satisfy the following inequality (see Assumption 2 of Khaled & Richtárik (2020)).

**Assumption 2** (General assumption for stochastic gradients). *We assume that for all $i = 1, \ldots, n$ there exist parameters $A_i, C_i \ge 0$, $B_i \ge 1$ such that*

$$\mathbb{E}\left[\|\nabla f_{\xi_{ij}^t}(x^t)\|^2 \mid x^t\right] \le 2A_i\left(f_i(x^t) - f_i^{\text{inf}}\right) + B_i\|\nabla f_i(x^t)\|^2 + C_i, \tag{9}$$

*where[1] $f_i^{\text{inf}} = \inf_{x \in \mathbb{R}^d} f_i(x) > -\infty$.*

Below we provide two examples of stochastic gradients fitting this assumption (for more detail, see (Khaled & Richtárik, 2020)).

**Example 1.** *Consider $\nabla f_{\xi_{ij}^t}(x^t)$ such that*

$$\mathbb{E}\left[\nabla f_{\xi_{ij}^t}(x^t) \mid x^t\right] = \nabla f_i(x^t) \quad and \quad \mathbb{E}\left[\left\|\nabla f_{\xi_{ij}^t}(x^t) - \nabla f_i(x^t)\right\|^2 \mid x^t\right] \le \sigma_i^2$$

*for some $\sigma_i \ge 0$. Then, due to variance decomposition,(9) holds with $A_i = 0$, $B_i = 0$, $C_i = \sigma_i^2$.*

---

[1]When $A_i = 0$ one can ignore the first term in the right-hand side of (9), i.e., assumption $\inf_{x \in \mathbb{R}^d} f_i(x) > -\infty$ is not required in this case.

**Example 2.** *Let $f_i(x) = \frac{1}{m_i} \sum_{j=1}^{m_i} f_{ij}(x)$, $f_{ij}$ be $L_{ij}$-smooth and $f_{ij}^{\inf} = \inf_{x \in \mathbb{R}^d} f_{ij}(x) > -\infty$. Following Gower et al. (2019), we consider a stochastic reformulation*

$$f_i(x) = \mathbb{E}_{v_i \sim \mathcal{D}_i}[f_{v_i}(x)] = \mathbb{E}_{v_i \sim \mathcal{D}_i}\left[\frac{1}{m_i} \sum_{j=1}^{m_i} f_{v_{ij}}(x)\right], \tag{10}$$

*where $\mathbb{E}_{v_i \sim \mathcal{D}_i}[v_{ij}] = 1$. One can show (see Proposition 2 of Khaled & Richtárik (2020)) that under the assumption that $\mathbb{E}_{v_i \sim \mathcal{D}_i}[v_{ij}^2]$ is finite for all $j$ stochastic gradient $\nabla f_{\xi_{ij}^t}(x^t) = \nabla f_{v_i^t}(x^t)$ with $v_i^t$ sampled from $\mathcal{D}_i$ satisfies (9) with $A_i = \max_j L_{ij} \mathbb{E}_{v_i \sim \mathcal{D}_i}[v_{ij}^2]$, $B_i = 1$, $C_i = 2A_i \Delta_i^{\inf}$, where $\Delta_i^{\inf} = \frac{1}{m_i} \sum_{j=1}^{m_i}(f_i^{\inf} - f_{ij}^{\inf})$. In particular, if $\mathbf{Prob}(\nabla f_{\xi_{ij}^t}(x^t) = \nabla f_{ij}(x^t)) = \frac{L_{ij}}{\sum_{l=1}^{m_i} L_{il}}$, then $A_i = \overline{L}_i = \frac{1}{m_i} \sum_{j=1}^{m_i} L_{ij}$, $B_i = 1$, and $C_i = 2A_i \Delta_i^{\inf}$.*

Stochastic gradient $\hat{g}_i(x^t)$ is computed using a mini-batch of $\tau_i$ independent samples satisfying (9):

$$\hat{g}_i(x^t) \stackrel{\text{def}}{=} \frac{1}{\tau_i} \sum_{j=1}^{\tau_i} \nabla f_{\xi_{ij}^t}(x^t).$$

### 4.1.3 Additional Assumptions for EF21-PAGE

In the analysis of EF21-PAGE, we rely on the following assumption.

**Assumption 3** (Average $\mathcal{L}$-smoothness). *Let every $f_i$ have the form (3). Assume that for all $t \geq 0$, $i = 1, \dots, n$, and batch $I_i^t$ (of size $\tau_i$), the minibatch stochastic gradients difference $\widetilde{\Delta}_i^t \stackrel{\text{def}}{=} \frac{1}{\tau_i} \sum_{j \in I_i^t}(\nabla f_{ij}(x^{t+1}) - \nabla f_{ij}(x^t))$ computed on the node $i$, satisfies $\mathbb{E}\left[\widetilde{\Delta}_i^t \mid x^t, x^{t+1}\right] = \Delta_i^t$ and*

$$\mathbb{E}\left[\left\|\widetilde{\Delta}_i^t - \Delta_i^t\right\|^2 \mid x^t, x^{t+1}\right] \leq \frac{\mathcal{L}_i^2}{\tau_i}\|x^{t+1} - x^t\|^2 \tag{11}$$

*with some $\mathcal{L}_i \geq 0$, where $\Delta_i^t \stackrel{\text{def}}{=} \nabla f_i(x^{t+1}) - \nabla f_i(x^t)$. We also define $\widetilde{\mathcal{L}} \stackrel{\text{def}}{=} \frac{1}{n} \sum_{i=1}^n \frac{(1-p_i)\mathcal{L}_i^2}{\tau_i}$.*

This assumption is satisfied for many standard/popular sampling strategies. For example, if $I_i^t$ is a full batch, then $\mathcal{L}_i = 0$. Another example is *uniform sampling* on $\{1, \dots, m\}$, and each $f_{ij}$ is $L_{ij}$-smooth. In this regime, one may verify that $\mathcal{L}_i \leq \max_{1 \leq j \leq m} L_{ij}$.

### 4.2 Main Results

Below we formulate the corollary establishing the complexities for each method. The complete version of this result is formulated and rigorously derived for each method in the appendix.

**Corollary 1.** *Suppose that Assumption 1 holds. Then, there exist appropriate choices of parameters for EF21-PP, EF21-BC, EF21-HB, EF21-Prox such that the number of communication rounds $T$ and the (expected) number of gradient computations at each node #grad for these methods to find an $\varepsilon$-stationary point, i.e., a point $\hat{x}^T$ such that $\mathbb{E}[\|\nabla f(\hat{x}^T)\|^2] \leq \varepsilon^2$ for EF21-PP, EF21-BC, EF21-HB and $\mathbb{E}[\|\mathcal{G}_\gamma(\hat{x}^T)\|^2] \leq \varepsilon^2$ for EF21-Prox, where $\mathcal{G}_\gamma(x) = 1/\gamma\left(x - \text{prox}_{\gamma r}(x - \gamma \nabla f(x))\right)$, are*

$$\text{EF21-PP:} \qquad T = \mathcal{O}\left(\frac{\widetilde{L}\delta^0}{p\alpha\varepsilon^2}\right), \quad \#grad = \mathcal{O}\left(\frac{\widetilde{L}\delta^0}{\alpha\varepsilon^2}\right)$$

$$\text{EF21-BC:} \qquad T = \#grad = \mathcal{O}\left(\frac{\widetilde{L}\delta^0}{\alpha_w \alpha_M \varepsilon^2}\right)$$

$$\text{EF21-HB:} \qquad T = \#grad = \mathcal{O}\left(\frac{\widetilde{L}\delta^0}{\varepsilon^2}\left(\frac{1}{\alpha} + \frac{1}{1-\eta}\right)\right)$$

$$\text{EF21-Prox:} \qquad T = \#grad = \mathcal{O}\left(\frac{\widetilde{L}\delta^0}{\alpha\varepsilon^2}\right),$$

*where $\widetilde{L} \stackrel{\text{def}}{=} \sqrt{\frac{1}{n} \sum_{i=1}^n L_i^2}$, $\delta_0 \stackrel{\text{def}}{=} f(x^0) - f^{\inf}$ (for EF21-Prox $\delta^0 = \Phi(x^0) - \Phi^{inf}$), $p$ is the probability of sampling the client in EF21-PP, $\alpha_w$ and $\alpha_M$ are contraction factors for compressors applied on the workers' and the master's sides respectively in EF21-BC, and $\eta \in [0,1)$ is the momentum parameter in EF21-HB.*

*If Assumptions 1 and 2 in the setup from Example 1 hold, then there exist appropriate choices of parameters for EF21-SGD such that the corresponding $T$ and the averaged number of gradient computations at each node $\overline{\#grad}$ are*

$$\text{EF21-SGD:} \qquad T = \mathcal{O}\left(\frac{\widetilde{L}\delta^0}{\alpha\varepsilon^2}\right), \quad \overline{\#grad} = \mathcal{O}\left(\frac{\widetilde{L}\delta^0}{\alpha\varepsilon^2} + \frac{\widetilde{L}\delta^0\sigma^2}{\alpha^3\varepsilon^4}\right),$$

*where $\sigma = \frac{1}{n}\sum_{i=1}^n \sigma_i^2$.*

*If Assumptions 1 and 3 hold, then there exist appropriate choices of parameters for EF21-PAGE such that the corresponding $T$ and $\overline{\#grad}$ are*

$$\text{EF21-PAGE:} \qquad T = \mathcal{O}\left(\frac{(\widetilde{L}+\widetilde{\mathcal{L}})\delta^0}{\alpha\varepsilon^2} + \frac{\sqrt{m}\widetilde{\mathcal{L}}\delta^0}{\varepsilon^2}\right), \quad \overline{\#grad} = \mathcal{O}\left(m + \frac{(\widetilde{L}+\widetilde{\mathcal{L}})\delta^0}{\alpha\varepsilon^2} + \frac{\sqrt{m}\widetilde{\mathcal{L}}\delta^0}{\varepsilon^2}\right),$$

*where $\widetilde{\mathcal{L}} = \sqrt{\frac{1-p}{n}\sum_{i=1}^n \mathcal{L}_i^2}$, $\tau_i \equiv \tau = 1$.*

**Remark:** We highlight some points for our results in Corollary 1 as follows:

● For EF21-PP and EF21-Prox, none of previous error feedback methods work on these two settings (partial participation and proximal/composite case). Thus, we provide the *first* convergence results for them. Moreover, we show that the gradient (computation) complexity for both EF21-PP and EF21-Prox is $\mathcal{O}(1/\alpha\varepsilon)$, matching the original vanilla EF21. It means that we extend EF21 to both settings for free.

● For EF21-BC, we show $\mathcal{O}(1/\alpha_w\alpha_M\varepsilon^2)$ complexity result. In particular, if one uses constant ratio of compression (e.g., $10\%$), then $\alpha \approx 0.1$. Then the result will be $\mathcal{O}(1/\varepsilon^2)$. However, previous result of DoubleSqueeze is $\mathcal{O}(\Delta/\varepsilon^3)$ and it also uses more strict assumption for the compressors ($\mathbb{E}\left[\|\mathcal{C}(x) - x\|\right] \leq \Delta$). Even if we ignore this, our results for EF21-BC is better than the one for DoubleSqueeze by a large factor $1/\varepsilon$.

● Similarly, our result for EF21-HB is roughly $\mathcal{O}(1/\varepsilon^2)$ (note that the momentum parameter $\eta$ is usually constant such as 0.2, 0.4, 0.9 used in our experiments). However, previous result of M-CSER is roughly $\mathcal{O}(G/\varepsilon^3)$ and it is proven under an additional bounded gradient assumption. Similarly, our EF21-HB is better by a large factor $1/\varepsilon$.

● For EF21-SGD and EF21-PAGE, we want to reduce the gradient complexity by using (variance-reduced) stochastic gradients instead of full gradient in the vanilla EF21. Note that $\sigma^2$ and $\Delta^{\inf}$ in EF21-SGD could be much smaller than $G$ in Choco-SGD since $G$ always depends on the dimension (and can be even infinite), while $\sigma^2$ and $\Delta^{\inf}$ are mostly dimension-free parameters (particularly, they are very small if the functions/data samples are similar/close). Thus, for high dimensional problems (e.g., deep neural networks), EF21-SGD can be better than Choco-SGD. Besides, in the finite-sum case (3), especially if the number of data samples $m$ on each client is not very large, then EF21-PAGE is much better since its complexity is roughly $\mathcal{O}(\sqrt{m}/\varepsilon^2)$ while EF21-SGD ones is roughly $\mathcal{O}(\sigma^2/\varepsilon^4)$.

## 5 EXPERIMENTS

In this section, we consider a logistic regression problem with a non-convex regularizer

$$\min_{x\in\mathbb{R}^d}\left\{f(x) = \frac{1}{N}\sum_{i=1}^N \log\left(1 + \exp\left(-b_i a_i^\top x\right)\right) + \lambda\sum_{j=1}^d \frac{x_j^2}{1+x_j^2}\right\}, \tag{12}$$

where $a_i \in \mathbb{R}^d$, $b_i \in \{-1,1\}$ are the training data, and $\lambda > 0$ is the regularization parameter, which is set to $\lambda = 0.1$ in all experiments. For all methods the stepsizes are initially chosen as the largest stepsize predicted by theory for EF21 (see Theorem 1), then they are tuned individually for each parameter setting. We provide more details on the datasets, hardware, experimental setups, and additional experiments, including deep learning experiments in Appendix A.

**Experiment 1: Fast convergence with variance reduction.** In our first experiment, we showcase the computation and communication superiority of EF21-PAGE (Alg. 3) over EF21-SGD.

Figure 8 illustrates that, in all cases, EF21-PAGE perfectly reduces the accumulated variance and converges to the desired tolerance, whereas EF21-SGD is stuck at some accuracy level. Moreover,

EF21-PAGE turns out to be surprisingly efficient with small bathsizes (eg, $1.5\%$ of the local data ) both in terms of the number of epochs and the # bits sent to the server per client. Interestingly, for most datasets, a further increase of bathsize does not considerably improve the convergence.

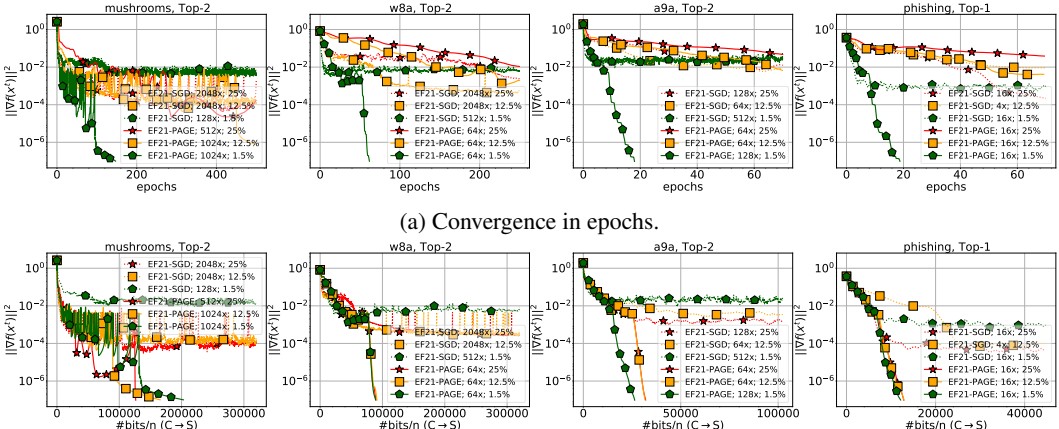

(a) Convergence in epochs.

(b) Convergence in terms of total number of bits sent from **C**lients to the **S**erver divided by $n$.

Figure 1: Comparison of EF21-PAGE and EF21-SGD with tuned parameters. By $1\times, 2\times, 4\times$ (and so on) we indicate that the stepsize was set to a multiple of the largest stepsize predicted by theory for EF21. By $25\%, 12.5\%$ and $1.5\%$ we refer to batchsizes equal $\lfloor 0.25 N_i \rfloor$, $\lfloor 0.125 N_i \rfloor$ and $\lfloor 0.015 N_i \rfloor$ for all clients $i = 1, \ldots, n$, where $N_i$ denotes the size of local dataset.

**Experiment 2: On the effect of partial participation of clients.** This experiment shows that EF21-PP (Alg. 4) can reduce communication costs and can be more practical than EF21. For this comparison, we consider $n = 100$ and, therefore, apply a different data partitioning, see Table 5 from Appendix A for more details.

It is predicted by our theory (Corollary 1) that, in terms of the number of iterations/communication rounds, *partial participation* slows down the convergence of EF21 by a fraction of participating clients . We observe this behavior in practice as well (see Figure 2a). However, since for EF21-PP the communications are considerably cheaper it outperforms EF21 in terms of # number of bits sent to the server per client on average (see Figure 2).

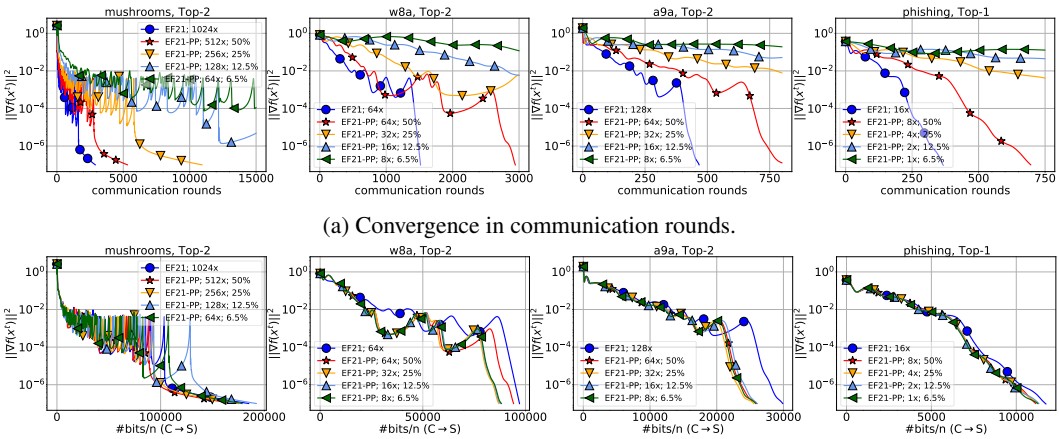

(a) Convergence in communication rounds.

(b) Convergence in terms of total number of bits sent from **C**lients to the **S**erver divided by $n$.

Figure 2: Comparison of EF21-PP and EF21 with tuned parameters. By $1\times, 2\times, 4\times$ (and so on) we indicate that the stepsize was set to a multiple of the largest stepsize predicted by theory for EF21. By $50\%, 25\%$ , $12.5\%$ and $6.5\%$ we refer to a number of participating clients equal to $\lfloor 0.5n \rfloor$, $\lfloor 0.25n \rfloor$, $\lfloor 0.125n \rfloor$ and $\lfloor 0.065n \rfloor$.

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

## TABLE OF CONTENTS

# A  EXTRA EXPERIMENTS

In this section, we give missing details on the experiments from Section 5, and provide additional experiments.

## A.1  NON-CONVEX LOGISTIC REGRESSION: ADDITIONAL EXPERIMENTS AND DETAILS

**Datasets, hardware and implementation.** We use standard LibSVM datasets (Chang & Lin, 2011), and split each dataset among $n$ clients. For experiments 1, 3, 4 and 5, we chose $n = 20$ whereas for the experiment 2 we consider $n = 100$. The first $n-1$ clients own equal parts, and the remaining part, of size $N - n \cdot \lfloor N/n \rfloor$, is assigned to the last client. We consider the heterogeneous data distribution regime (i.e. we do not make any additional assumptions on data similarity between workers). A summary of datasets and details of splitting data among workers can be found in Tables 3 and 5. The algorithms are implemented in Python 3.8; we use 3 different CPU cluster node types in all experiments: 1) AMD EPYC 7702 64-Core; 2) Intel(R) Xeon(R) Gold 6148 CPU @ 2.40GHz; 3) Intel(R) Xeon(R) Gold 6248 CPU @ 2.50GHz. In all algorithms involving compression, we use Top-$k$ (Alistarh et al., 2017) as a canonical example of contractive compressor $\mathcal{C}$, and fix the compression ratio $k/d \approx 0.01$, where $d$ is the number of features in the dataset. For all algorithms, at each iteration we compute the squared norm of the exact/full gradient for comparison of the methods performance. We terminate our algorithms either if they reach the certain number of iterations or the following stopping criterion is satisfied: $\|\nabla f(x^t)\|^2 \leq 10^{-7}$.

In all experiments, the stepsize is set to the largest stepsize predicted by theory for EF21 multiplied by some constant multiplier which was individually tuned in all cases.

| Dataset | $n$ | $N$ (total # of datapoints) | $d$ (# of features) | k | $N_i$ |
|---------|-----|-----------------------------|---------------------|-----|-------|
| mushrooms | 20 | 8,120 | 112 | 2 | 406 |
| w8a | 20 | 49,749 | 300 | 2 | 2,487 |
| a9a | 20 | 32,560 | 123 | 2 | 1,628 |
| phishing | 20 | 11,055 | 68 | 1 | 552 |
| real-sim | 20 | 72,309 | 20,958 | 210 | 3615 |

Table 3: Summary of the datasets and splitting of the data among clients for Experiments 1, 3, 4, and 5. Here $N_i$ denotes the number of datapoints per client.

**Experiment 1: Fast convergence with variance reductions (extra details).** The parameters $p_i$ of the PAGE estimator are set to $p_i = p \overset{\text{def}}{=} \frac{1}{n} \sum_{i=1}^{n} \frac{\tau_i}{\tau_i + N_i}$, where $\tau_i$ is the batchsize for clients $i = 1, \ldots, n$ (see Table 4 for details). In our experiments, we assume that the sampling of Bernoulli random variable is performed on server side (which means that at each iteration for all clients $b_i^t = 1$ or $b_i^t = 0$). And if $b_i^t = 0$, then in line 5 of Algorithm 3 $I_i^t$ is sampled without replacement uniformly at random. Table 4 shows the selection of parameter $p$ for each experiment.

For each batchsize from the set[2]

$$\{95\%, 50\%, 25\%, 12.5\%, 6.5\%, 3\%\},$$

we tune the stepsize multiplier for EF21-PAGE within the set

$$\{0.25, 0.5, 1, 2, 4, 8, 16, 32, 64, 128, 256, 512, 1024, 2048\}.$$

The best pair (batchsize, stepsize multiplier) is chosen in such a way that it gives the best convergence in terms of $\#\text{bits}/n(C \to S)$. In the rest of the experiments, fine tuning is performed in a similar fashion.

---

[2]By 50%, 25% (and so on) we refer to a batchsize, which is equals to $\lfloor 0.5N_i \rfloor$, $\lfloor 0.25N_i \rfloor$ (and so on) for all clients $i = 1, \ldots, n$.

| Dataset | 25% | 12.5% | 1.5% |
|---------|------|-------|------|
| mushrooms | 0.1992 | 0.1097 | 0.0146 |
| w8a | 0.1998 | 0.1108 | 0.0147 |
| a9a | 0.2 | 0.1109 | 0.0145 |
| phishing | 0.2 | 0.1111 | 0.0143 |
| real-sim | 0.1999 | 0.1109 | 0.0147 |

Table 4: Summary of the parameter choice of $p$.

**Experiment 2: On the effect of partial participation of clients (extra details)** In this experiment, we consider $n = 100$ and, therefore, a different data partitioning, see Table 5 for the summary.

| Dataset | $n$ | $N$ (total # of datapoints) | $d$ (# of features) | k | $N_i$ |
|---------|-----|------------------------------|----------------------|---|-------|
| mushrooms | 100 | 8,120 | 112 | 2 | 81 |
| w8a | 100 | 49,749 | 300 | 2 | 497 |
| a9a | 100 | 32,560 | 123 | 2 | 325 |
| phishing | 100 | 11,055 | 68 | 1 | 110 |

Table 5: Summary of the datasets and splitting of the data among clients for Experiment 5. Here $N_i$ denotes the number of datapoints per client.

We tune the stepsize multiplier for EF21-PP within the following set:

$$\{0.125, 0.25, 0.5, 1, 2, 4, 8, 16, 32, 64, 128, 256, 512, 1024, 2048, 4096\}.$$

**Experiment 3: On the advantages of bidirectional biased compression.** Our next experiment demonstrates that the application of the **S**erver → **C**lients compression in EF21-BC (Alg. 5) does not significantly slow down the convergence in terms of the communication rounds but requires much less bits to be transmitted. Indeed, Figure 3a illustrates that that it is sufficient to communicate only $5\% - 15\%$ of data to perform similarly to EF21 (Alg. 1).[3] Note that EF21 communicates full vectors from the **S**erver → **C**lients, and, therefore, may have slower communication at each round. In Figure 3b we take into account only the number of bits sent from clients to the server, and therefore we observe the same behavior as in Figure 3a. However, if we care about the total number of bits (see Figure 3c), then EF21-BC considerably outperforms EF21 in all cases.

---

[3]The range $5\% - 15\%$ comes from the fractions $k/d$ for each dataset.

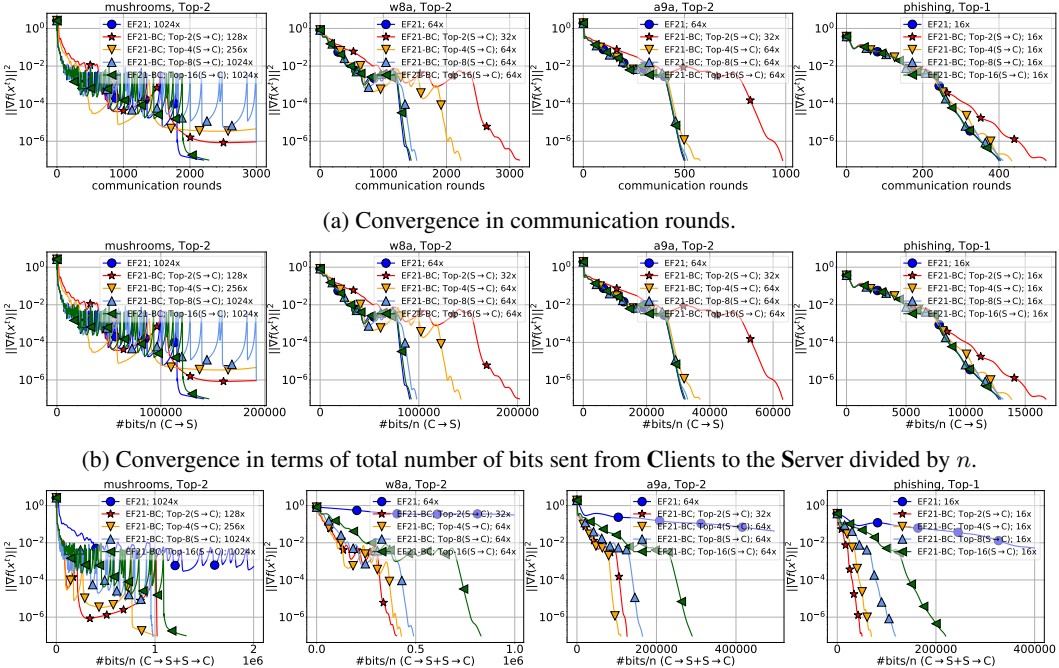

(a) Convergence in communication rounds.

(b) Convergence in terms of total number of bits sent from **C**lients to the **S**erver divided by $n$.

(c) Convergence in terms of total number of bits sent from **C**lients to the **S**erver plus the total number of bits broadcasted from **S**erver to **C**lients divided by $n$.

Figure 3: Comparison of EF21-BC and EF21 with tuned stepsizes . By $1\times, 2\times, 4\times$ (and so on) we indicate that the stepsize was set to a multiple of the largest stepsize predicted by theory for EF21 (see the Theorem 1) .

For each parameter $k$ in Server-Clients compression, we tune the stepsize multiplier for EF21-BC within the following set:

$$\{0.125, 0.25, 0.5, 1, 2, 4, 8, 16, 32, 64, 128, 256, 512, 1024, 2048\}.$$

**Experiment 4: On the cheaper computations via EF21-SGD.** The fourth experiment (see Figure 4a) illustrates that EF21-SGD (Alg. 2) is the more preferable choice than EF21 for the cases when full gradient computations are costly.

For each batchsize from the set[4]

$$\{95\%, 50\%, 25\%, 12.5\%, 6.5\%, 3\%\},$$

we tune the stepsize multiplier for EF21-SGD within the following set:

$$\{0.25, 0.5, 1, 2, 4, 8, 16, 32, 64, 128, 256, 512, 1024, 2048\}.$$

Figure 4a illustrates that EF21-SGD is able to reach a moderate tolerance in $5 - 10$ epochs.

---

[4]By $50\%, 25\%$ (and so on) we refer to a batchsize, which is equals to $\lfloor 0.5N_i \rfloor$, $\lfloor 0.25N_i \rfloor$ (and so on) for all clients $i = 1, \dots, n$.

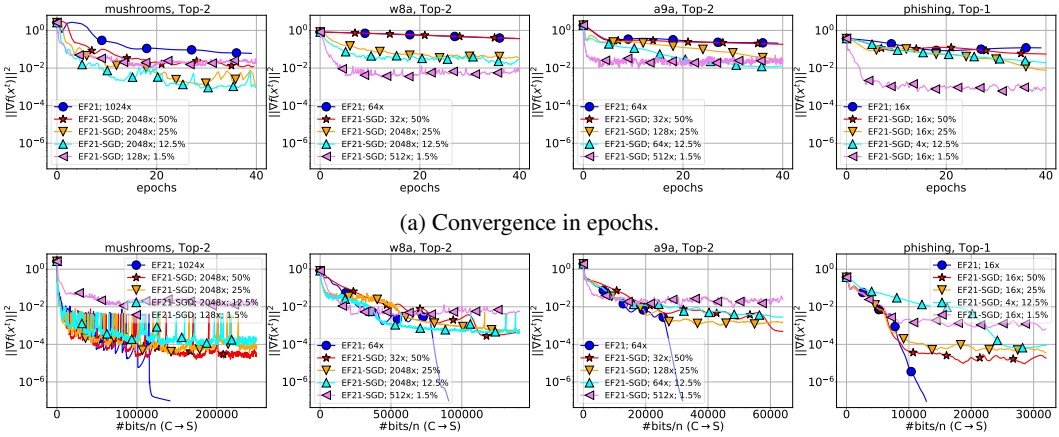

(a) Convergence in epochs.

(b) Convergence in terms of the number of bits sent from **C**lients to the **S**erver by each client.

Figure 4: Comparison of EF21-SGD and EF21 with tuned stepsizes. By $1\times, 2\times, 4\times$ (and so on) we indicate that the stepsize was set to a multiple of the largest stepsize predicted by theory for EF21. By $50\%, 25\%$ (and so on) we refer to a batchsize, which is equals to $\lfloor 0.5N_i \rfloor$, $\lfloor 0.25N_i \rfloor$ (and so on) for all clients $i = 1, \ldots, n$.

However, due to the accumulated variance introduced by SGD, estimator EF21-SGD is stuck at some accuracy level (see Figure 4b), showing the usual behavior of the SGD observed in practice.

**Experiment 5: On the effect of heavy ball momentum.** In this experiment (see Figure 5), we show that for the majority of the considered datasets heavy ball acceleration used in EF21-HB (Alg. 6) improves the convergence of EF21 method. For every dataset (and correspondingly chosen parameter $k$) we tune momentum parameter $\eta$ in EF21-HB by making a grid search over all possible parameter values from $0.05$ to $0.99$ with the step $0.05$. Finally, for our plots we pick $\eta \in \{0.05, 0.2, 0.25, 0.4, 0.9\}$ since the first four values shows the best performance and $\eta = 0.9$ is a popular choice in practice.

For each parameter $\eta$ from the set

$$\{0.05, 0.1, 0.15, 0.2, 0.25, 0.3, 0.35, 0.4, 0.45, 0.5, 0.55, 0.6, 0.65, 0.7, 0.75, 0.8, 0.85, 0.9, 0.95, 0.99\}.$$

we perform a grid search of stepsize multiplier within the powers of 2:

$$\{0.125, 0.25, 0.5, 1, 2, 4, 8, 16, 32, 64, 128, 256, 512, 1024, 2048\}.$$

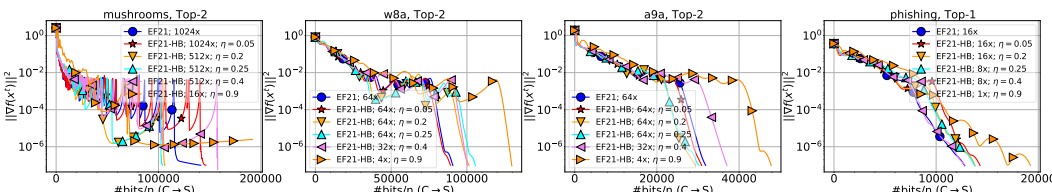

Figure 5: Comparison of EF21-HB and EF21 with tuned parameters in terms of total number of bits sent from **C**lients to the **S**erver divided by $n$. By $1\times, 2\times, 4\times$ (and so on) we indicate that the stepsize was set to a multiple of the largest stepsize predicted by theory for EF21 (see the Theorem 1) .

**Experiments on a larger dataset.** In these additional experiments, we test our methods on larger problem and dataset. The dimension of the dataset used in these experiments is $d = 20958$. Each method is run for $500$ epochs. In this case, we observe a similar behavior as in our previous experiments.

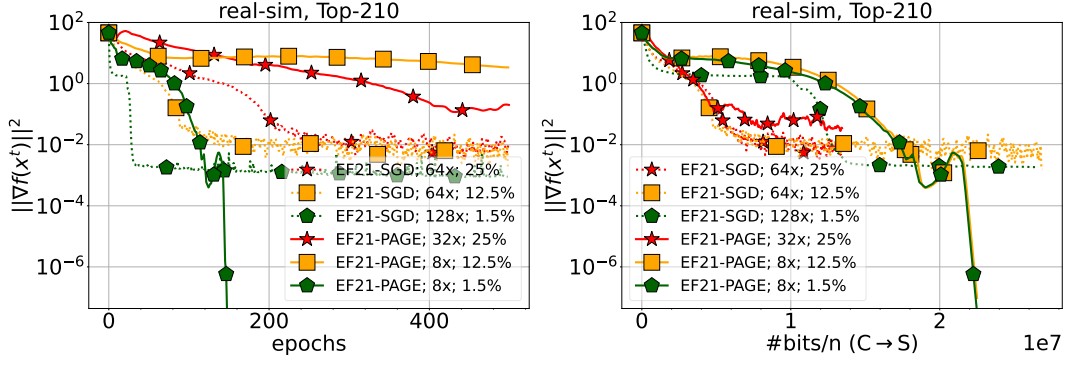

(a) Convergence in epochs.

(b) Convergence in terms of total number of bits sent from **C**lients to the **S**erver divided by $n$.

Figure 6: Comparison of EF21-PAGE and EF21-SGD with tuned parameters. By $1\times, 2\times, 4\times$ (and so on) we indicate that the stepsize was set to a multiple of the largest stepsize predicted by theory for EF21. By $25\%, 12.5\%$ and $1.5\%$ we refer to batchsizes equal $\lfloor 0.25N_i \rfloor, \lfloor 0.125N_i \rfloor$ and $\lfloor 0.015N_i \rfloor$ for all clients $i = 1, \ldots, n$, where $N_i$ denotes the size of local dataset.

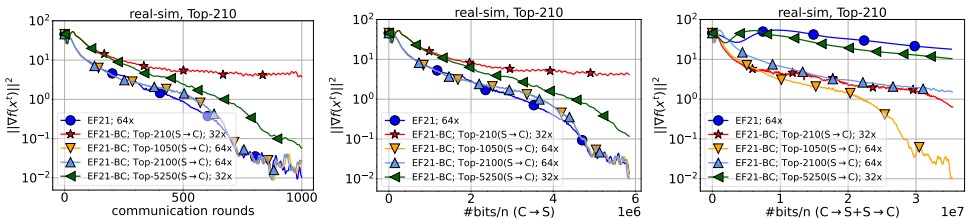

(a) Convergence in communication rounds.

(b) Convergence in terms of total number of bits sent from **C**lients to the **S**erver divided by $n$.

(c) Convergence in terms of total number of bits sent from **C**lients to the **S**erver plus the total number of bits broadcasted from **S**erver to **C**lients divided by $n$.

Figure 7: Comparison of EF21-BC and EF21 with tuned stepsizes . By $1\times, 2\times, 4\times$ (and so on) we indicate that the stepsize was set to a multiple of the largest stepsize predicted by theory for EF21 (see the Theorem 1).

**Comparison to non-compressed methods.** In addition, we compare EF21-PAGE and EF21-SGD to the baseline methods without compression: PAGE (Figure 8a) and SGD (Figure 9a). In these experiments, we observe that EF21-PAGE and EF21-SGD require much less information to transmit in order to achieve the same accuracy of the solution as the methods without compression (PAGE, SGD).

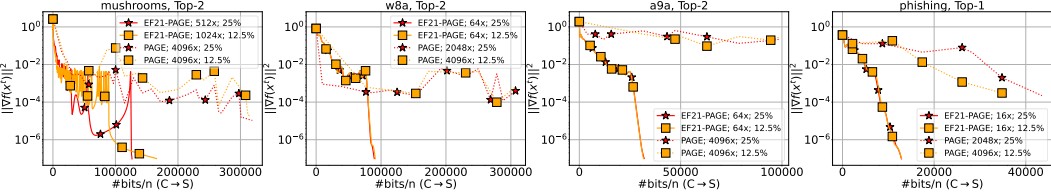

(a) Convergence in terms of total number of bits sent from **C**lients to the **S**erver divided by $n$.

Figure 8: Comparison of EF21-PAGE and PAGE with tuned parameters. By $1\times, 2\times, 4\times$ (and so on) we indicate that the stepsize was set to a multiple of the largest stepsize predicted by theory for EF21. By $25\%, 12.5\%$ and $1.5\%$ we refer to batchsizes equal $\lfloor 0.25N_i \rfloor, \lfloor 0.125N_i \rfloor$ and $\lfloor 0.015N_i \rfloor$ for all clients $i = 1, \ldots, n$, where $N_i$ denotes the size of local dataset.

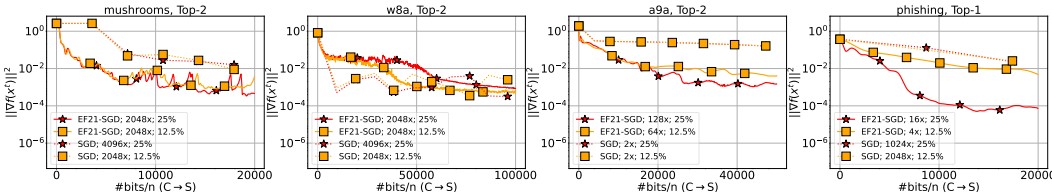

(a) Convergence in terms of total number of bits sent from **C**lients to the **S**erver divided by $n$.

Figure 9: Comparison of EF21-SGD and SGD with tuned parameters. By $1\times, 2\times, 4\times$ (and so on) we indicate that the stepsize was set to a multiple of the largest stepsize predicted by theory for EF21. By $25\%, 12.5\%$ and $1.5\%$ we refer to batchsizes equal $\lfloor 0.25N_i \rfloor, \lfloor 0.125N_i \rfloor$ and $\lfloor 0.015N_i \rfloor$ for all clients $i = 1, \ldots, n$, where $N_i$ denotes the size of local dataset.

## A.2 EXPERIMENTS WITH LEAST SQUARES

In this section, we conduct the experiments on a function satisfying the PŁ-condition (see Assumption 4). In particular, we consider the least squares problem:

$$\min_{x \in \mathbb{R}^d} \left\{ f(x) = \frac{1}{n} \sum_{i=1}^{n} (a_i^\top x - b_i)^2 \right\},$$

where $a_i \in \mathbb{R}^d, b_i \in \{-1, 1\}$ are the training data. We use the same datasets as for the logistic regression problem.

**Experiment: On the effect of heavy ball momentum in PŁ-setting.** For PŁ-setting, EF21-HB also improves the convergence over EF21 for the majority of the datasets (see Figure 10). Stepsize and momentum parameter $\eta$ are chosen using the same strategy as for the logistic regression experiments (see section A.1).

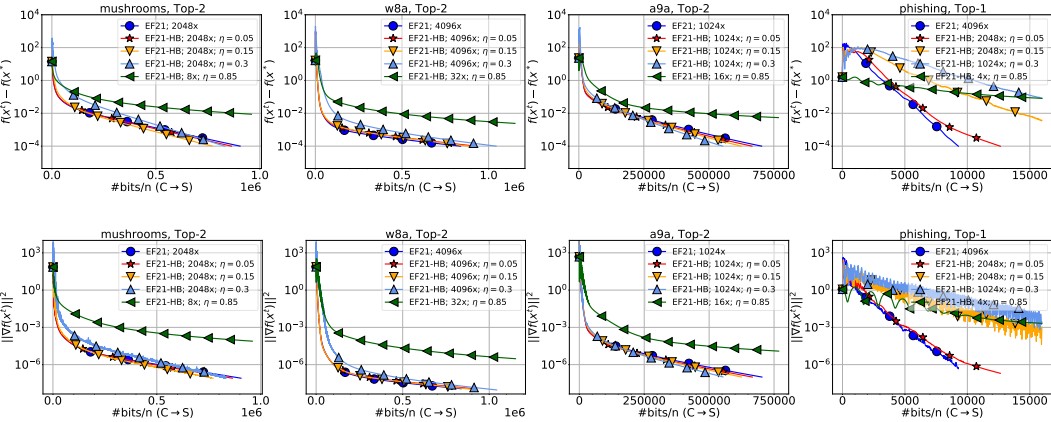

Figure 10: Comparison of EF21-HB and EF21 with tuned parameters in terms of total number of bits sent from **C**lients to the **S**erver divided by $n$. By $1\times, 2\times, 4\times$ (and so on) we indicate that the stepsize was set to a multiple of the largest stepsize predicted by theory for EF21 (see the Theorem 2).

## A.3 DEEP LEARNING EXPERIMENTS

In this experiment, the exact/full gradient $\nabla f_i(x^{k+1})$ in the algorithm EF21-HB is replaced by its stochastic estimator (we later refer to this method as EF21-SGD-HB). We compare the resulting method with some existing baselines on a deep learning multi-class image classification task. In particular, we compare our EF21-SGD-HB method to EF21+-SGD-HB[5], EF-SGD-HB[6], EF21-SGD

[5]EF21+-SGD-HB is the method obtained from EF21-SGD-HB via replacing EF21 by EF21+ compressor
[6]EF-SGD-HB is the method obtained from EF21-SGD-HB via replacing EF21 by EF compressor

and EF-SGD on the problem of training ResNet18 (He et al., 2016) model on CIFAR-10 (Krizhevsky et al., 2009) dataset. For more details about the EF21+ and EF type methods and their applications in deep learning we refer reader to (Richtárik et al., 2021). We implement the algorithms in PyTorch (Paszke et al., 2019) and run the experiments on a single GPU NVIDIA GeForce RTX 2080 Ti. The dataset is split into $n = 8$ equal parts. Total train set size for CIFAR-10 is 50,000. The test set for evaluation has 10,000 data points. The train set is split into batches of size $\tau = 32$. The first seven workers own an equal number of batches of data, while the last worker gets the rest. In our experiments, we fix $k \approx 0.05d$, $\tau = 32$ and momentum parameter $\eta = 0.9$.[7] As it is usually done in deep learning applications, stochastic gradients are generated via so-called "shuffle once" strategy, i.e., workers randomly shuffle their datasets and then select minibatches using the obtained order (Bottou, 2009; 2012; Mishchenko et al., 2020). We tune the stepsize $\gamma$ within the range $\{0.0625, 0.125, 0.25, 0.5, 1\}$ and for each method we individually chose the one $\gamma$ giving the highest accuracy score on test. For momentum methods, the best stepsize was $0.5$, whereas for the non-momentum ones it was $0.125$.

The experiments show (see Figure 11) that the train loss for momentum methods decreases slower than for the non-momentum ones, whereas for the test loss situation is the opposite. Finally, momentum methods show a considerable improvement in the accuracy score on the test set over the existing EF21-SGD and EF-SGD.

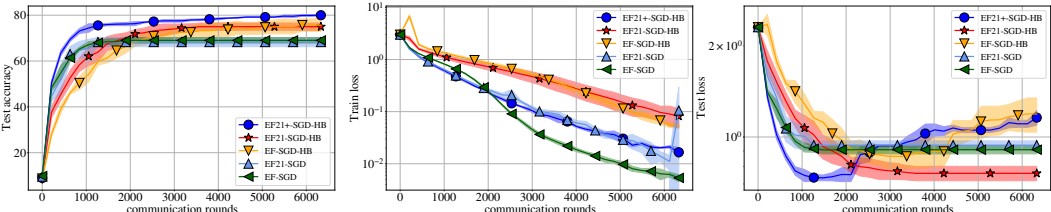

Figure 11: Comparison of EF-SGD and EF21-SGD with EF-SGD-HB, EF21-SGD-HB, and EF21+-SGD-HB with tuned stepsizes applied to train ResNet18 on CIFAR10.

---

[7]Here, $d$ is the number of model parameters. For ResNet18, $d = 11,511,784$.

| EF21 EF21-SGD EF21-PP | $R^t = \left\| x^{t+1} - x^t \right\|^2, \delta^t = f(x^t) - f^{\inf}$ |
|---|---|
| EF21-PAGE | $R^t = \left\| x^{t+1} - x^t \right\|^2, \delta^t = f(x^t) - f^{\inf},$ $P_i^t = \left\| \nabla f_i(x^t) - v_i^t \right\|^2, V_i^t = \left\| v_i^t - g_i^t \right\|^2$ |
| EF21-BC | $P_i^t = \left\| \widetilde{g}_i^t - \nabla f_i(x^t) \right\|^2, R^t = \left\| x^{t+1} - x^t \right\|^2, \delta^t = f(x^t) - f^{\inf}$ |
| EF21-HB | $R^t = (1-\eta)^2 \left\| z^{t+1} - z^t \right\|^2, \delta^t = f(x^t) - f^{\inf}$ |
| EF21-Prox | $R^t = \left\| x^{t+1} - x^t \right\|^2, \Phi(x) = f(x) + r(x), \delta^t = \Phi(x^t) - \Phi^{inf},$ $\mathcal{G}_\gamma(x) = \frac{1}{\gamma} \left( x - \mathrm{prox}_{\gamma r}(x - \gamma \nabla f(x)) \right)$ |

Table 6: Summary of frequently used notations in the proofs.

## B  NOTATIONS AND ASSUMPTIONS

We now introduce an additional assumption, which enables us to obtain a faster linear convergence result in different settings.

**Assumption 4** (Polyak-Łojasiewicz). *There exists $\mu > 0$ such that $f(x) - f(x^\star) \leq \frac{1}{2\mu} \left\| \nabla f(x) \right\|^2$ for all $x \in \mathbb{R}^d$, where $x^\star = \arg\min_{x \in \mathbb{R}^d} f$.*

Table 6 summarizes the most frequently used notations in our analysis. Additionally, we comment on the main quantities here. We define $\delta^t \stackrel{\text{def}}{=} f(x^t) - f^{\inf 8}$, $R^t \stackrel{\text{def}}{=} \left\| x^{t+1} - x^t \right\|^2$. In the analysis of EF21-HB, it is useful to adapt this notation to $R^t \stackrel{\text{def}}{=} (1-\eta)^2 \left\| z^{t+1} - z^t \right\|^2$, where $\{z^t\}_{t \geq 0}$ is the sequence of virtual iterates introduced in Section H. We denote $G_i^t \stackrel{\text{def}}{=} \left\| \nabla f_i(x^t) - g_i^t \right\|^2$, $G^t \stackrel{\text{def}}{=} \frac{1}{n} \sum_{i=1}^n G_i^t$ following Richtárik et al. (2021), where $g_i^t$ is an EF21 estimator at a node $i$. Throughout the paper $\widetilde{L}^2 \stackrel{\text{def}}{=} \frac{1}{n} \sum_{i=1}^n L_i^2$, where $L_i$ is a smoothness constant for $f_i(\cdot)$, $i = 1, \ldots, n$ (see Assumption 1).

---

[8]If, additionally, Assumption 4 holds, then $f^{\inf}$ can be replaced by $f(x^\star)$ for $x^\star = \arg\min_{x \in \mathbb{R}^d} f(x)$.

## C EF21

For completeness, we provide here the detailed proofs for EF21 (Richtárik et al., 2021).

---

**Algorithm 1** EF21

---

1: **Input:** starting point $x^0 \in \mathbb{R}^d$; $g_i^0 \in \mathbb{R}^d$ for $i = 1, \ldots, n$ (known by nodes); $g^0 = \frac{1}{n}\sum_{i=1}^n g_i^0$ (known by master); learning rate $\gamma > 0$
2: **for** $t = 0, 1, 2, \ldots, T-1$ **do**
3:     Master computes $x^{t+1} = x^t - \gamma g^t$ and broadcasts $x^{t+1}$ to all nodes
4:     **for all nodes** $i = 1, \ldots, n$ **in parallel do**
5:         Compress $c_i^t = \mathcal{C}(\nabla f_i(x^{t+1}) - g_i^t)$ and send $c_i^t$ to the master
6:         Update local state $g_i^{t+1} = g_i^t + c_i^t$
7:     **end for**
8:     Master computes $g^{t+1} = \frac{1}{n}\sum_{i=1}^n g_i^{t+1}$ via $g^{t+1} = g^t + \frac{1}{n}\sum_{i=1}^n c_i^t$
9: **end for**

---

**Lemma 1.** *Let $\mathcal{C}$ be a contractive compressor, then for all $i = 1, \ldots, n$*

$$\mathbb{E}\left[G_i^{t+1}\right] \leq (1-\theta)\mathbb{E}\left[G_i^t\right] + \beta L_i^2 \mathbb{E}\left[\left\|x^{t+1} - x^t\right\|^2\right], \text{ and} \tag{13}$$

$$\mathbb{E}\left[G^{t+1}\right] \leq (1-\theta)\mathbb{E}\left[G^t\right] + \beta \widetilde{L}^2 \mathbb{E}\left[\left\|x^{t+1} - x^t\right\|^2\right], \tag{14}$$

*where $\theta \overset{def}{=} 1 - (1-\alpha)(1+s)$, $\beta \overset{def}{=} (1-\alpha)\left(1 + s^{-1}\right)$ for any $s > 0$.*

*Proof.* Define $W^t \overset{def}{=} \{g_1^t, \ldots, g_n^t, x^t, x^{t+1}\}$, then

$$
\begin{aligned}
\mathbb{E}\left[G_i^{t+1}\right] &= \mathbb{E}\left[\mathbb{E}\left[G_i^{t+1} \mid W^t\right]\right] \\
&= \mathbb{E}\left[\mathbb{E}\left[\left\|g_i^{t+1} - \nabla f_i(x^{t+1})\right\|^2 \mid W^t\right]\right] \\
&= \mathbb{E}\left[\mathbb{E}\left[\left\|g_i^t + \mathcal{C}(\nabla f_i(x^{t+1}) - g_i^t) - \nabla f_i(x^{t+1})\right\|^2 \mid W^t\right]\right] \\
&\overset{(8)}{\leq} (1-\alpha)\mathbb{E}\left[\left\|\nabla f_i(x^{t+1}) - g_i^t\right\|^2\right] \\
&\overset{(i)}{\leq} (1-\alpha)(1+s)\mathbb{E}\left[\left\|\nabla f_i(x^t) - g_i^t\right\|^2\right] \\
&\qquad + (1-\alpha)\left(1 + s^{-1}\right)\left\|\nabla f_i(x^{t+1}) - \nabla f_i(x^t)\right\|^2 \\
&\overset{(ii)}{\leq} (1-\alpha)(1+s)\mathbb{E}\left[\left\|\nabla f_i(x^t) - g_i^t\right\|^2\right] \\
&\qquad + (1-\alpha)\left(1 + s^{-1}\right) L_i^2 \mathbb{E}\left[\left\|x^{t+1} - x^t\right\|^2\right] \\
&\overset{(iii)}{\leq} (1-\theta)\mathbb{E}\left[\left\|\nabla f_i(x^t) - g_i^t\right\|^2\right] + \beta L_i^2 \mathbb{E}\left[\left\|x^{t+1} - x^t\right\|^2\right],
\end{aligned}
\tag{15}
$$

where $(i)$ follows by Young's inequality (118), $(ii)$ holds by Assumption 1, and in $(iii)$ we apply the definition of $\theta$ and $\beta$. Averaging the above inequalities over $i = 1, \ldots, n$, we obtain (14). $\qquad\square$

### C.1 CONVERGENCE FOR GENERAL NON-CONVEX FUNCTIONS

**Theorem 1.** *Let Assumption 1 hold, and let the stepsize in Algorithm 1 be set as*

$$0 < \gamma \leq \left(L + \widetilde{L}\sqrt{\frac{\beta}{\theta}}\right)^{-1}. \tag{16}$$

*Fix $T \geq 1$ and let $\hat{x}^T$ be chosen from the iterates $x^0, x^1, \ldots, x^{T-1}$ uniformly at random. Then*

$$\mathbb{E}\left[\left\|\nabla f(\hat{x}^T)\right\|^2\right] \leq \frac{2\left(f(x^0) - f^{inf}\right)}{\gamma T} + \frac{\mathbb{E}\left[G^0\right]}{\theta T}, \tag{17}$$

*where $\widetilde{L} = \sqrt{\frac{1}{n} \sum_{i=1}^{n} L_i^2}$, $\theta = 1 - (1-\alpha)(1+s)$, $\beta = (1-\alpha)\left(1 + s^{-1}\right)$ for any $s > 0$.*

*Proof.* According to our notation, for Algorithm 1 $R^t = \left\|x^{t+1} - x^t\right\|^2$. By Lemma 1, we have

$$\mathbb{E}\left[G^{t+1}\right] \leq (1-\theta)\,\mathbb{E}\left[G^t\right] + \beta\widetilde{L}^2\mathbb{E}\left[R^t\right]. \tag{18}$$

Next, using Lemma 16 and Jensen's inequality (119), we obtain the bound

$$
\begin{aligned}
f(x^{t+1}) &\leq f(x^t) - \frac{\gamma}{2}\left\|\nabla f(x^t)\right\|^2 - \left(\frac{1}{2\gamma} - \frac{L}{2}\right)R^t + \frac{\gamma}{2}\left\|\frac{1}{n}\sum_{i=1}^{n}\left(g_i^t - \nabla f_i(x^t)\right)\right\|^2 \\
&\leq f(x^t) - \frac{\gamma}{2}\left\|\nabla f(x^t)\right\|^2 - \left(\frac{1}{2\gamma} - \frac{L}{2}\right)R^t + \frac{\gamma}{2}\frac{1}{n}\sum_{i=1}^{n}\left\|g_i^t - \nabla f_i(x^t)\right\|^2 \\
&= f(x^t) - \frac{\gamma}{2}\left\|\nabla f(x^t)\right\|^2 - \left(\frac{1}{2\gamma} - \frac{L}{2}\right)R^t + \frac{\gamma}{2}G^t. 
\end{aligned}
\tag{19}
$$

Subtracting $f^{\text{inf}}$ from both sides of the above inequality, taking expectation and using the notation $\delta^t = f(x^t) - f^{\text{inf}}$, we get

$$\mathbb{E}\left[\delta^{t+1}\right] \leq \mathbb{E}\left[\delta^t\right] - \frac{\gamma}{2}\mathbb{E}\left[\left\|\nabla f(x^t)\right\|^2\right] - \left(\frac{1}{2\gamma} - \frac{L}{2}\right)\mathbb{E}\left[R^t\right] + \frac{\gamma}{2}\mathbb{E}\left[G^t\right]. \tag{20}$$

Then by adding (20) with a $\frac{\gamma}{2\theta}$ multiple of (18) we obtain

$$
\begin{aligned}
\mathbb{E}\left[\delta^{t+1}\right] + \frac{\gamma}{2\theta}\mathbb{E}\left[G^{t+1}\right] &\leq \mathbb{E}\left[\delta^t\right] - \frac{\gamma}{2}\mathbb{E}\left[\left\|\nabla f(x^t)\right\|^2\right] - \left(\frac{1}{2\gamma} - \frac{L}{2}\right)\mathbb{E}\left[R^t\right] + \frac{\gamma}{2}\mathbb{E}\left[G^t\right] \\
&\qquad + \frac{\gamma}{2\theta}\left(\beta\widetilde{L}^2\mathbb{E}\left[R^t\right] + (1-\theta)\mathbb{E}\left[G^t\right]\right) \\
&= \mathbb{E}\left[\delta^t\right] + \frac{\gamma}{2\theta}\mathbb{E}\left[G^t\right] - \frac{\gamma}{2}\mathbb{E}\left[\left\|\nabla f(x^t)\right\|^2\right] \\
&\qquad - \left(\frac{1}{2\gamma} - \frac{L}{2} - \frac{\gamma}{2\theta}\beta\widetilde{L}^2\right)\mathbb{E}\left[R^t\right] \\
&\leq \mathbb{E}\left[\delta^t\right] + \frac{\gamma}{2\theta}\mathbb{E}\left[G^t\right] - \frac{\gamma}{2}\mathbb{E}\left[\left\|\nabla f(x^t)\right\|^2\right].
\end{aligned}
$$

The last inequality follows from the bound $\gamma^2\frac{\beta\widetilde{L}^2}{\theta} + L\gamma \leq 1$, which holds because of Lemma 15 and our assumption on the stepsize. By summing up inequalities for $t = 0, \ldots, T-1$, we get

$$0 \leq \mathbb{E}\left[\delta^T + \frac{\gamma}{2\theta}G^T\right] \leq \delta^0 + \frac{\gamma}{2\theta}\mathbb{E}\left[G^0\right] - \frac{\gamma}{2}\sum_{t=0}^{T-1}\mathbb{E}\left[\left\|\nabla f(x^t)\right\|^2\right].$$

Multiplying both sides by $\frac{2}{\gamma T}$, after rearranging we get

$$\sum_{t=0}^{T-1}\frac{1}{T}\mathbb{E}\left[\left\|\nabla f(x^t)\right\|^2\right] \leq \frac{2\delta^0}{\gamma T} + \frac{\mathbb{E}\left[G^0\right]}{\theta T}.$$

It remains to notice that the left hand side can be interpreted as $\mathbb{E}\left[\left\|\nabla f(\hat{x}^T)\right\|^2\right]$, where $\hat{x}^T$ is chosen from $x^0, x^1, \ldots, x^{T-1}$ uniformly at random. $\square$

**Corollary 2.** *Let assumptions of Theorem 1 hold,*

$$
\begin{aligned}
g_i^0 &= \nabla f_i(x^0), \qquad i = 1, \ldots, n, \\
\gamma &= \left(L + \widetilde{L}\sqrt{\beta/\theta}\right)^{-1}.
\end{aligned}
$$

*Then, after $T$ iterations/communication rounds of* EF21 *we have* $\mathbb{E}\left[\left\|\nabla f(\hat{x}^T)\right\|^2\right] \leq \varepsilon^2$. *It requires*

$$T = \#grad = \mathcal{O}\left(\frac{\widetilde{L}\delta^0}{\alpha\varepsilon^2}\right)$$

*iterations/communications rounds/gradint computations at each node, where* $\widetilde{L} = \sqrt{\frac{1}{n}\sum_{i=1}^n L_i^2}$, $\delta^0 = f(x^0) - f^{inf}$.

*Proof.* Since $g_i^0 = \nabla f_i(x^0)$, $i = 1,\ldots,n$, we have $G^0 = 0$ and by Theorem 1

$$
\begin{aligned}
\#\text{grad} &= T \stackrel{(i)}{\leq} \frac{2\delta^0}{\gamma\varepsilon^2} \stackrel{(ii)}{\leq} \frac{2\delta^0}{\varepsilon^2}\left(L + \widetilde{L}\sqrt{\frac{\beta}{\theta}}\right) \stackrel{(iii)}{\leq} \frac{2\delta^0}{\varepsilon^2}\left(L + \widetilde{L}\left(\frac{2}{\alpha} - 1\right)\right) \\
&\leq \frac{2\delta^0}{\varepsilon^2}\left(L + \frac{2\widetilde{L}}{\alpha}\right) \stackrel{(iv)}{\leq} \frac{2\delta^0}{\varepsilon^2}\left(\frac{\widetilde{L}}{\alpha} + \frac{2\widetilde{L}}{\alpha}\right) = \frac{6\widetilde{L}\delta^0}{\alpha\varepsilon^2},
\end{aligned}
$$

where in $(i)$ is due to the rate (17) given by Theorem 1. In two $(ii)$ we plug in the stepsize, in $(iii)$ we use Lemma 17, and $(iv)$ follows by the inequalities $\alpha \leq 1$, and $L \leq \widetilde{L}$. $\qquad\square$

## C.2 Convergence under Polyak-Łojasiewicz Condition

**Theorem 2.** *Let Assumptions 1 and 4 hold, and let the stepsize in Algorithm 1 be set as*

$$0 < \gamma \leq \min\left\{\left(L + \widetilde{L}\sqrt{\frac{2\beta}{\theta}}\right)^{-1}, \frac{\theta}{2\mu}\right\}. \tag{21}$$

*Let* $\Psi^t \stackrel{def}{=} f(x^t) - f(x^\star) + \frac{\gamma}{\theta}G^t$. *Then for any $T \geq 0$, we have*

$$\mathbb{E}\left[\Psi^T\right] \leq (1 - \gamma\mu)^T \mathbb{E}\left[\Psi^0\right], \tag{22}$$

*where* $\widetilde{L} = \sqrt{\frac{1}{n}\sum_{i=1}^n L_i^2}$, $\theta = 1 - (1-\alpha)(1+s)$, $\beta = (1-\alpha)\left(1 + s^{-1}\right)$ *for any $s > 0$.*

*Proof.* We proceed as in the previous proof, but use the PL inequality, subtract $f(x^\star)$ from both sides of (19) and utilize the notation $\delta^t = f(x^t) - f(x^\star)$

$$
\begin{aligned}
\delta^{t+1} &\leq \delta^t - \frac{\gamma}{2}\left\|\nabla f(x^t)\right\|^2 - \left(\frac{1}{2\gamma} - \frac{L}{2}\right)R^t + \frac{\gamma}{2}G^t \\
&\leq \delta^t - \gamma\mu\left(f(x^t) - f(x^\star)\right) - \left(\frac{1}{2\gamma} - \frac{L}{2}\right)R^t + \frac{\gamma}{2}G^t. \\
&= (1 - \gamma\mu)\delta^t - \left(\frac{1}{2\gamma} - \frac{L}{2}\right)R^t + \frac{\gamma}{2}G^t.
\end{aligned}
$$

Take expectation on both sides of the above inequality and add it with a $\frac{\gamma}{\theta}$ multiple of (18), then

$$
\begin{aligned}
\mathbb{E}\left[\delta^{t+1}\right] + \mathbb{E}\left[\frac{\gamma}{\theta}G^{t+1}\right] &\leq (1-\gamma\mu)\mathbb{E}\left[\delta^t\right] - \left(\frac{1}{2\gamma} - \frac{L}{2}\right)\mathbb{E}\left[R^t\right] + \frac{\gamma}{2}\mathbb{E}\left[G^t\right] \\
&\quad + \frac{\gamma}{\theta}\left((1-\theta)\mathbb{E}\left[G^t\right] + \beta\widetilde{L}^2\mathbb{E}\left[R^t\right]\right) \\
&= (1-\gamma\mu)\mathbb{E}\left[\delta^t\right] + \frac{\gamma}{\theta}\left(1 - \frac{\theta}{2}\right)\mathbb{E}\left[G^t\right] \\
&\quad - \left(\frac{1}{2\gamma} - \frac{L}{2} - \frac{\beta\widetilde{L}^2\gamma}{\theta}\right)\mathbb{E}\left[R^t\right].
\end{aligned}
$$

Note that our assumption on the stepsize implies that $1 - \frac{\theta}{2} \leq 1 - \gamma\mu$ and $\frac{1}{2\gamma} - \frac{L}{2} - \frac{\beta\widetilde{L}^2\gamma}{\theta} \geq 0$. The last inequality follows from the bound $\gamma^2\frac{2\beta\widetilde{L}^2}{\theta} + \gamma L \leq 1$, which holds because of Lemma 15 and our assumption on the stepsize. Thus,

$$\mathbb{E}\left[\delta^{t+1} + \frac{\gamma}{\theta}G^{t+1}\right] \leq (1 - \gamma\mu)\mathbb{E}\left[\delta^t + \frac{\gamma}{\theta}G^t\right].$$

It remains to unroll the recurrence. $\qquad\square$

**Corollary 3.** *Let assumptions of Theorem 2 hold,*

$$g_i^0 = \nabla f_i(x^0), \qquad i = 1, \ldots, n,$$

$$\gamma = \min\left\{\left(L + \widetilde{L}\sqrt{\frac{2\beta}{\theta}}\right)^{-1}, \frac{\theta}{2\mu}\right\}.$$

*Then, after $T$ iterations/communication rounds of* EF21 *we have* $\mathbb{E}\left[f(x^T) - f(x^\star)\right] \leq \varepsilon$. *It requires*

$$T = \#grad = \mathcal{O}\left(\frac{\widetilde{L}}{\alpha\mu}\log\left(\frac{\delta^0}{\varepsilon}\right)\right) \tag{23}$$

*iterations/communications rounds/gradint computations at each node, where* $\widetilde{L} = \sqrt{\frac{1}{n}\sum_{i=1}^n L_i^2}$, $\delta^0 = f(x^0) - f^{inf}$.

*Proof.* Notice that

$$\min\left\{\left(L + \widetilde{L}\sqrt{\frac{2\beta}{\theta}}\right)^{-1}, \frac{\theta}{2\mu}\right\}\mu \overset{(i)}{\geq} \min\left\{\mu\left(L + \widetilde{L}\sqrt{2}\left(\frac{2}{\alpha} - 1\right)\right)^{-1}, \frac{1 - \sqrt{1 - \alpha}}{2}\right\}$$

$$\overset{(ii)}{\geq} \min\left\{\mu\left(L + \frac{2\sqrt{2}\widetilde{L}}{\alpha}\right)^{-1}, \frac{\alpha}{4}\right\}$$

$$\overset{(iii)}{\geq} \min\left\{\mu\left(\frac{(1 + 2\sqrt{2})\widetilde{L}}{\alpha}\right)^{-1}, \frac{\alpha}{4}\right\}$$

$$= \min\left\{\frac{\alpha\mu}{(1 + 2\sqrt{2})\widetilde{L}}, \frac{\alpha}{4}\right\}$$

$$\geq \min\left\{\frac{\alpha\mu}{4\widetilde{L}}, \frac{\alpha}{4}\right\}$$

$$= \frac{\alpha\mu}{4\widetilde{L}},$$

where in $(i)$ we apply Lemma 17, and plug in $\theta = 1 - \sqrt{1 - \alpha}$ according to Lemma 17, $(ii)$ follows by $\sqrt{1 - \alpha} \leq 1 - \alpha/2$, $(iii)$ follows by the inequalities $\alpha \leq 1$, and $L \leq \widetilde{L}$.

Let $g_i^0 = \nabla f_i(x^0)$, $i = 1, \ldots, n$, then $G^0 = 0$. Thus using (22) and the above computations, we arrive at

$$\#grad = T \leq \frac{\log\left(\frac{\delta^0}{\varepsilon}\right)}{\log(1 - \gamma\mu)^{-1}} \overset{(i)}{\leq} \frac{1}{\gamma\mu}\log\left(\frac{\delta^0}{\varepsilon}\right) \leq \frac{4\widetilde{L}}{\alpha\mu}\log\left(\frac{\delta^0}{\varepsilon}\right),$$

where $(i)$ is due to (122). $\qquad\square$

# D    STOCHASTIC GRADIENTS

In this section, we study the extension of EF21 to the case when stochastic gradients are used instead of full gradients.

---
**Algorithm 2** EF21-SGD
---
1: **Input:** starting point $x^0 \in \mathbb{R}^d$; $g_i^0 \in \mathbb{R}^d$ (known by nodes); $g^0 = \frac{1}{n}\sum_{i=1}^n g_i^0$ (known by master); learning rate $\gamma > 0$
2: **for** $t = 0, 1, 2, \ldots, T-1$ **do**
3:     Master computes $x^{t+1} = x^t - \gamma g^t$ and broadcasts $x^{t+1}$ to all nodes
4:     **for all nodes** $i = 1, \ldots, n$ **in parallel do**
5:         Compute a stochastic gradient $\hat{g}_i(x^{t+1}) = \frac{1}{\tau}\sum_{j=1}^{\tau} \nabla f_{\xi_{ij}^t}(x^{t+1})$
6:         Compress $c_i^t = \mathcal{C}(\hat{g}_i(x^{t+1}) - g_i^t)$ and send $c_i^t$ to the master
7:         Update local state $g_i^{t+1} = g_i^t + c_i^t$
8:     **end for**
9:     Master computes $g^{t+1} = \frac{1}{n}\sum_{i=1}^n g_i^{t+1}$ via $g^{t+1} = g^t + \frac{1}{n}\sum_{i=1}^n c_i^t$
10: **end for**
---

**Lemma 2.** *Let Assumptions 1 and 2 hold. Then for all $t \geq 0$ and all constants $\rho, \nu > 0$ EF21-SGD satisfies*

$$
\begin{aligned}
\mathbb{E}\left[G^{t+1}\right] &\leq (1-\hat{\theta})\mathbb{E}\left[G^t\right] + \hat{\beta}_1 \widetilde{L}^2 \mathbb{E}\left[\left\|x^{t+1} - x^t\right\|^2\right] \\
&\quad + \widetilde{A}\hat{\beta}_2 \mathbb{E}\left[f(x^{t+1}) - f^{\inf}\right] + \widetilde{C}\hat{\beta}_2,
\end{aligned}
\tag{24}
$$

*where* $\hat{\theta} \overset{def}{=} 1 - (1-\alpha)(1+\rho)(1+\nu)$, $\hat{\beta}_1 \overset{def}{=} 2(1-\alpha)(1+\rho)\left(1+\frac{1}{\nu}\right)$, $\hat{\beta}_2 \overset{def}{=} 2(1-\alpha)(1+\rho)\left(1+\frac{1}{\nu}\right) + \left(1+\frac{1}{\rho}\right)$, $\widetilde{A} = \max_{i=1,\ldots,n} \frac{2(A_i + L_i(B_i-1))}{\tau_i}$, $\widetilde{C} = \frac{1}{n}\sum_{i=1}^n \left(\frac{2(A_i + L_i(B_i-1))}{\tau_i}\left(f^{\inf} - f_i^{\inf}\right) + \frac{C_i}{\tau_i}\right)$.

*Proof.* For all $\rho, \nu > 0$ we have

$$
\begin{aligned}
\mathbb{E}\left[G_i^{t+1}\right] &= \mathbb{E}\left[\left\|g_i^{t+1} - \nabla f_i(x^{t+1})\right\|^2\right] \\
&\leq (1+\rho)\mathbb{E}\left[\left\|\mathcal{C}\left(\hat{g}_i(x^{t+1}) - g_i^t\right) - \left(\hat{g}_i(x^{t+1}) - g_i^t\right)\right\|^2\right] \\
&\quad + \left(1+\frac{1}{\rho}\right)\mathbb{E}\left[\left\|\hat{g}_i(x^{t+1}) - \nabla f_i(x^{t+1})\right\|^2\right] \\
&\leq (1-\alpha)(1+\rho)\mathbb{E}\left[\left\|g_i^t - \hat{g}_i(x^{t+1})\right\|^2\right] \\
&\quad + \left(1+\frac{1}{\rho}\right)\mathbb{E}\left[\left\|\hat{g}_i(x^{t+1}) - \nabla f_i(x^{t+1})\right\|^2\right] \\
&\leq (1-\alpha)(1+\rho)(1+\nu)\mathbb{E}\left[\left\|g_i^t - \nabla f_i(x^t)\right\|^2\right] \\
&\quad + 2(1-\alpha)(1+\rho)\left(1+\frac{1}{\nu}\right)\mathbb{E}\left[\left\|\nabla f_i(x^{t+1}) - \hat{g}_i(x^{t+1})\right\|^2\right] \\
&\quad + 2(1-\alpha)(1+\rho)\left(1+\frac{1}{\nu}\right)\mathbb{E}\left[\left\|\nabla f_i(x^{t+1}) - \nabla f_i(x^t)\right\|^2\right] \\
&\quad + \left(1+\frac{1}{\rho}\right)\mathbb{E}\left[\left\|\hat{g}_i(x^{t+1}) - \nabla f_i(x^{t+1})\right\|^2\right] \\
&\leq (1-\hat{\theta})\mathbb{E}\left[G_i^t\right] + \hat{\beta}_1 L_i^2 \mathbb{E}\left[\left\|x^{t+1} - x^t\right\|^2\right] \\
&\quad + \hat{\beta}_2 \mathbb{E}\left[\left\|\hat{g}_i(x^{t+1}) - \nabla f_i(x^{t+1})\right\|^2\right],
\end{aligned}
$$

where we introduced $\hat{\theta} \overset{\text{def}}{=} 1 - (1-\alpha)(1+\rho)(1+\nu)$, $\hat{\beta}_1 \overset{\text{def}}{=} 2(1-\alpha)(1+\rho)\left(1+\frac{1}{\nu}\right)$, $\hat{\beta}_2 \overset{\text{def}}{=} 2(1-\alpha)(1+\rho)\left(1+\frac{1}{\nu}\right) + \left(1+\frac{1}{\rho}\right)$. Next we use independence of $\nabla f_{\xi_{ij}^t}(x^t)$, variance decomposition, and (9) to estimate the last term:

$$
\begin{aligned}
\mathbb{E}\left[G_i^{t+1}\right] &\leq (1-\hat{\theta})\mathbb{E}\left[G_i^t\right] + \hat{\beta}_1 L_i^2 \mathbb{E}\left[\left\|x^{t+1} - x^t\right\|^2\right] \\
&\quad + \frac{\hat{\beta}_2}{\tau_i^2} \sum_{j=1}^{\tau_i} \mathbb{E}\left[\left\|\nabla f_{\xi_{ij}^t}(x^{t+1}) - \nabla f_i(x^{t+1})\right\|^2\right] \\
&= (1-\hat{\theta})\mathbb{E}\left[G_i^t\right] + \hat{\beta}_1 L_i^2 \mathbb{E}\left[\left\|x^{t+1} - x^t\right\|^2\right] \\
&\quad + \frac{\hat{\beta}_2}{\tau_i^2} \sum_{j=1}^{\tau_i} \left(\mathbb{E}\left[\left\|\nabla f_{\xi_{ij}^t}(x^{t+1})\right\|^2\right] - \mathbb{E}\left[\left\|\nabla f_i(x^{t+1})\right\|^2\right]\right) \\
&\overset{(9)}{\leq} (1-\hat{\theta})\mathbb{E}\left[G_i^t\right] + \hat{\beta}_1 L_i^2 \mathbb{E}\left[\left\|x^{t+1} - x^t\right\|^2\right] \\
&\quad + \frac{2A_i\hat{\beta}_2}{\tau_i}\mathbb{E}\left[f_i(x^{t+1}) - f_i^{\inf}\right] + \frac{\hat{\beta}_2(B_i-1)}{\tau_i}\mathbb{E}\left[\left\|\nabla f_i(x^{t+1})\right\|^2\right] + \frac{C_i\hat{\beta}_2}{\tau_i} \\
&\leq (1-\hat{\theta})\mathbb{E}\left[G_i^t\right] + \hat{\beta}_1 L_i^2 \mathbb{E}\left[\left\|x^{t+1} - x^t\right\|^2\right] \\
&\quad + \frac{2(A_i + L_i(B_i-1))\hat{\beta}_2}{\tau_i}\mathbb{E}\left[f_i(x^{t+1}) - f_i^{\inf}\right] + \frac{C_i\hat{\beta}_2}{\tau_i}
\end{aligned}
$$

Averaging the obtained inequality for $i = 1,\dots,n$ we get

$$
\begin{aligned}
\mathbb{E}\left[G^{t+1}\right] &\leq (1-\hat{\theta})\mathbb{E}\left[G^t\right] + \hat{\beta}_1 \widetilde{L}^2 \mathbb{E}\left[\left\|x^{t+1} - x^t\right\|^2\right] \\
&\quad + \frac{1}{n}\sum_{i=1}^{n}\left(\frac{2(A_i + L_i(B_i-1))\hat{\beta}_2}{\tau_i}\mathbb{E}\left[f_i(x^{t+1}) - f_i^{\inf}\right] + \frac{C_i\hat{\beta}_2}{\tau_i}\right) \\
&\leq (1-\hat{\theta})\mathbb{E}\left[G^t\right] + \hat{\beta}_1 \widetilde{L}^2 \mathbb{E}\left[\left\|x^{t+1} - x^t\right\|^2\right] \\
&\quad + \frac{1}{n}\sum_{i=1}^{n}\left(\frac{2(A_i + L_i(B_i-1))\hat{\beta}_2}{\tau_i}\mathbb{E}\left[f_i(x^{t+1}) - f^{\inf}\right]\right) \\
&\quad + \frac{\hat{\beta}_2}{n}\sum_{i=1}^{n}\left(\frac{2(A_i + L_i(B_i-1))}{\tau_i}\left(f^{\inf} - f_i^{\inf}\right) + \frac{C_i}{\tau_i}\right) \\
&\leq (1-\hat{\theta})\mathbb{E}\left[G^t\right] + \hat{\beta}_1 \widetilde{L}^2 \mathbb{E}\left[\left\|x^{t+1} - x^t\right\|^2\right] + \widetilde{A}\hat{\beta}_2 \mathbb{E}\left[f(x^{t+1}) - f^{\inf}\right] + \widetilde{C}\hat{\beta}_2
\end{aligned}
$$

$\square$

## D.1 CONVERGENCE FOR GENERAL NON-CONVEX FUNCTIONS

**Theorem 3.** *Let Assumptions 1 and 2 hold, and let the stepsize in Algorithm 2 be set as*

$$
0 < \gamma \leq \left(L + \widetilde{L}\sqrt{\frac{\hat{\beta}_1}{\hat{\theta}}}\right)^{-1}, \tag{25}
$$

*where $\widetilde{L} = \sqrt{\frac{1}{n}\sum_{i=1}^{n} L_i^2}$, $\hat{\theta} \overset{\text{def}}{=} 1 - (1-\alpha)(1+\rho)(1+\nu)$, $\hat{\beta}_1 \overset{\text{def}}{=} 2(1-\alpha)(1+\rho)\left(1+\frac{1}{\nu}\right)$, and $\rho,\nu > 0$ are some positive numbers. Assume that batchsizes $\tau_1,\dots,\tau_i$ are such that $\frac{\gamma\widetilde{A}\hat{\beta}_2}{2\hat{\theta}} < 1$, where $\widetilde{A} = \max_{i=1,\dots,n}\frac{2(A_i + L_i(B_i-1))}{\tau_i}$ and $\hat{\beta}_2 \overset{\text{def}}{=} 2(1-\alpha)(1+\rho)\left(1+\frac{1}{\nu}\right) + \left(1+\frac{1}{\rho}\right)$. Fix $T \geq 1$ and let $\hat{x}^T$ be chosen from the iterates $x^0, x^1, \dots, x^{T-1}$ with following probabilities:*

$$
\mathbf{Prob}\left\{\hat{x}^T = x^t\right\} = \frac{w_t}{W_T}, \quad w_t = \left(1 - \frac{\gamma\widetilde{A}\hat{\beta}_2}{2\hat{\theta}}\right)^t, \quad W_T = \sum_{t=0}^{T} w_t.
$$

*Then*

$$\mathbb{E}\left[\left\|\nabla f(\hat{x}^T)\right\|^2\right] \le \frac{2(f(x^0) - f^{\inf})}{\gamma T\left(1 - \frac{\gamma \widetilde{A}\hat{\beta}_2}{2\hat{\theta}}\right)^T} + \frac{\mathbb{E}\left[G^0\right]}{\hat{\theta} T\left(1 - \frac{\gamma \widetilde{A}\hat{\beta}_2}{2\hat{\theta}}\right)^T} + \frac{\widetilde{C}\hat{\beta}_2}{\hat{\theta}}, \tag{26}$$

*where* $\widetilde{C} = \frac{1}{n}\sum_{i=1}^{n}\left(\frac{2(A_i + L_i(B_i - 1))}{\tau_i}\left(f^{\inf} - f_i^{\inf}\right) + \frac{C_i}{\tau_i}\right)$.

*Proof.* We notice that inequality (20) holds for EF21-SGD as well, i.e., we have

$$\mathbb{E}\left[\delta^{t+1}\right] \le \mathbb{E}\left[\delta^t\right] - \frac{\gamma}{2}\mathbb{E}\left[\left\|\nabla f(x^t)\right\|^2\right] - \left(\frac{1}{2\gamma} - \frac{L}{2}\right)\mathbb{E}\left[R^t\right] + \frac{\gamma}{2}\mathbb{E}\left[G^t\right].$$

Summing up the above inequality with a $\frac{\gamma}{2\hat{\theta}}$ multiple of (24), we derive

$$
\begin{aligned}
\mathbb{E}\left[\delta^{t+1} + \frac{\gamma}{2\hat{\theta}}G^{t+1}\right] &\le \mathbb{E}\left[\delta^t\right] - \frac{\gamma}{2}\mathbb{E}\left[\left\|\nabla f(x^t)\right\|^2\right] - \left(\frac{1}{2\gamma} - \frac{L}{2}\right)\mathbb{E}\left[R^t\right] + \frac{\gamma}{2}\mathbb{E}\left[G^t\right] \\
&\quad + \frac{\gamma}{2\hat{\theta}}(1 - \hat{\theta})\mathbb{E}\left[G^t\right] + \frac{\gamma}{2\hat{\theta}}\hat{\beta}_1\widetilde{L}^2\mathbb{E}\left[R^t\right] \\
&\quad + \frac{\gamma}{2\hat{\theta}}\widetilde{A}\hat{\beta}_2\mathbb{E}\left[\delta^{t+1}\right] + \frac{\gamma}{2\hat{\theta}}\widetilde{C}\hat{\beta}_2 \\
&\le \frac{\gamma\widetilde{A}\hat{\beta}_2}{2\hat{\theta}}\mathbb{E}\left[\delta^{t+1}\right] + \mathbb{E}\left[\delta^t + \frac{\gamma}{2\hat{\theta}}G^t\right] - \frac{\gamma}{2}\mathbb{E}\left[\left\|\nabla f(x^t)\right\|^2\right] + \frac{\gamma}{2\hat{\theta}}\widetilde{C}\hat{\beta}_2 \\
&\quad - \left(\frac{1}{2\gamma} - \frac{L}{2} - \frac{\gamma\hat{\beta}_1\widetilde{L}^2}{2\hat{\theta}}\right)\mathbb{E}\left[R^t\right] \\
&\overset{(25)}{\le} \frac{\gamma\widetilde{A}\hat{\beta}_2}{2\hat{\theta}}\mathbb{E}\left[\delta^{t+1}\right] + \mathbb{E}\left[\delta^t + \frac{\gamma}{2\hat{\theta}}G^t\right] - \frac{\gamma}{2}\mathbb{E}\left[\left\|\nabla f(x^t)\right\|^2\right] + \frac{\gamma}{2\hat{\theta}}\widetilde{C}\hat{\beta}_2,
\end{aligned}
$$

where $\hat{\theta} \overset{\text{def}}{=} 1 - (1 - \alpha)(1 + \rho)(1 + \nu)$, $\hat{\beta}_1 \overset{\text{def}}{=} 2(1 - \alpha)(1 + \rho)\left(1 + \frac{1}{\nu}\right)$, $\hat{\beta}_2 \overset{\text{def}}{=} 2(1 - \alpha)(1 + \rho)\left(1 + \frac{1}{\nu}\right) + \left(1 + \frac{1}{\rho}\right)$, and $\rho, \nu > 0$ are some positive numbers. Next, we rearrange the terms

$$
\begin{aligned}
\mathbb{E}\left[\left\|\nabla f(x^t)\right\|^2\right] &\le \frac{2}{\gamma}\left(\mathbb{E}\left[\delta^t + \frac{\gamma}{2\hat{\theta}}G^t\right] - \left(1 - \frac{\gamma\widetilde{A}\hat{\beta}_2}{2\hat{\theta}}\right)\mathbb{E}\left[\delta^{t+1}\right] - \frac{\gamma}{2\hat{\theta}}\mathbb{E}\left[G^{t+1}\right]\right) \\
&\quad + \frac{\widetilde{C}\hat{\beta}_2}{\hat{\theta}} \\
&\le \frac{2}{\gamma}\left(\mathbb{E}\left[\delta^t + \frac{\gamma}{2\hat{\theta}}G^t\right] - \left(1 - \frac{\gamma\widetilde{A}\hat{\beta}_2}{2\hat{\theta}}\right)\mathbb{E}\left[\delta^{t+1} + \frac{\gamma}{2\hat{\theta}}\mathbb{E}\left[G^{t+1}\right]\right]\right) \\
&\quad + \frac{\widetilde{C}\hat{\beta}_2}{\hat{\theta}},
\end{aligned}
$$

sum up the obtained inequalities for $t = 0,1,\ldots,T$ with weights $w_t/W_T$, and use the definition of $\hat{x}^T$

$$
\begin{aligned}
\mathbb{E}\left[\left\|\nabla f(\hat{x}^T)\right\|^2\right] &= \frac{1}{W_K}\sum_{t=0}^{T}w_t\mathbb{E}\left[\left\|\nabla f(x^t)\right\|^2\right] \\
&\le \frac{2}{\gamma W_T}\sum_{t=0}^{T}\left(w_t\mathbb{E}\left[\delta^t + \frac{\gamma}{2\hat{\theta}}G^t\right] - w_{t+1}\mathbb{E}\left[\delta^{t+1} + \frac{\gamma}{2\hat{\theta}}\mathbb{E}\left[G^{t+1}\right]\right]\right) \\
&\quad + \frac{\widetilde{C}\hat{\beta}_2}{\hat{\theta}} \\
&\le \frac{2\delta^0}{\gamma W_T} + \frac{\mathbb{E}\left[G^0\right]}{\hat{\theta} W_T} + \frac{\widetilde{C}\hat{\beta}_2}{\hat{\theta}}.
\end{aligned}
$$

Finally, we notice

$$W_T = \sum_{t=0}^{T} w_t \geq (T+1) \min_{t=0,1,\ldots,T} w_t > T \left(1 - \frac{\gamma \widetilde{A} \hat{\beta}_2}{2\hat{\theta}}\right)^T$$

that finishes the proof. $\qquad\square$

**Corollary 4.** *Let assumptions of Theorem 3 hold, $\rho = \alpha/2$, $\nu = \alpha/4$,*

$$\gamma = \frac{1}{L + \widetilde{L}\sqrt{\frac{\hat{\beta}_1}{\hat{\theta}}}},$$

$$\tau_i = \left\lceil \max\left\{1, \frac{2T\gamma\left(A_i + L_i(B_i - 1)\right)\hat{\beta}_2}{\hat{\theta}}, \frac{8\left(A_i + L_i(B_i - 1)\right)\hat{\beta}_2}{\hat{\theta}\varepsilon^2}\delta_i^{\inf}, \frac{4C_i\hat{\beta}_2}{\hat{\theta}\varepsilon^2}\right\} \right\rceil,$$

$$T = \left\lceil \max\left\{\frac{16\delta^0}{\gamma\varepsilon^2}, \frac{8\mathbb{E}\left[G^0\right]}{\hat{\theta}\varepsilon^2}\right\} \right\rceil,$$

*where $\delta_i^{\inf} = f^{\inf} - f_i^{\inf}$, $\delta^0 = f(x^0) - f^{\inf}$. Then, after $T$ iterations of* EF21-SGD *we have* $\mathbb{E}\left[\left\|\nabla f(\hat{x}^T)\right\|^2\right] \leq \varepsilon^2$. *It requires*

$$T = \mathcal{O}\left(\frac{\widetilde{L}\delta^0 + \mathbb{E}\left[G^0\right]}{\alpha\varepsilon^2}\right)$$

*iterations/communications rounds,*

$$\begin{aligned}
\#grad_i &= \tau_i T \\
&= \mathcal{O}\Bigg(\frac{\widetilde{L}\delta^0 + \mathbb{E}\left[G^0\right]}{\alpha\varepsilon^2} + \frac{\left(\widetilde{L}\delta^0 + \mathbb{E}\left[G^0\right]\right)\left(\hat{A}_i(\delta^0 + \delta_i^{\inf}) + C_i\right)}{\alpha^3\varepsilon^4} \\
&\qquad\qquad + \frac{(\widetilde{L}\delta^0 + \mathbb{E}\left[G^0\right])\hat{A}_i\mathbb{E}\left[G^0\right]}{\alpha^2(\alpha L + \widetilde{L})\varepsilon^4}\Bigg)
\end{aligned}$$

*stochastic oracle calls for worker $i$, and*

$$\begin{aligned}
\overline{\#grad} &= \frac{1}{n}\sum_{i=1}^{n}\tau_i T \\
&= \mathcal{O}\Bigg(\frac{\widetilde{L}\delta^0 + \mathbb{E}\left[G^0\right]}{\alpha\varepsilon^2} + \frac{1}{n}\sum_{i=1}^{n}\frac{\left(\widetilde{L}\delta^0 + \mathbb{E}\left[G^0\right]\right)\left(\hat{A}_i(\delta^0 + \delta_i^{\inf}) + C_i\right)}{\alpha^3\varepsilon^4} \\
&\qquad\qquad + \frac{1}{n}\sum_{i=1}^{n}\frac{(\widetilde{L}\delta^0 + \mathbb{E}\left[G^0\right])\hat{A}_i\mathbb{E}\left[G^0\right]}{\alpha^2(\alpha L + \widetilde{L})\varepsilon^4}\Bigg)
\end{aligned}$$

*stochastic oracle calls per worker on average, where $\hat{A}_i = A_i + L_i(B_i - 1)$.*

*Proof.* The given choice of $\tau_i$ ensures that $\left(1 - \frac{\gamma\widetilde{A}\hat{\beta}_2}{2\hat{\theta}}\right)^T = \mathcal{O}(1)$ and $\widetilde{C}\hat{\beta}_2/\hat{\theta} \leq \varepsilon/2$. Next, the choice of $T$ ensures that the right-hand side of (26) is smaller than $\varepsilon$. Finally, after simple computation we get the expression for $\tau_i T$. $\qquad\square$

**Corollary 5.** *Consider the setting described in Example 1. Let assumptions of Theorem 3 hold, $\rho = \alpha/2$, $\nu = \alpha/4$,*

$$\gamma = \frac{1}{L + \widetilde{L}\sqrt{\frac{\hat{\beta}_1}{\hat{\theta}}}}, \quad \tau_i = \left\lceil \max\left\{1, \frac{4\sigma_i^2\hat{\beta}_2}{\hat{\theta}\varepsilon^2}\right\} \right\rceil, \quad T = \left\lceil \max\left\{\frac{16\delta^0}{\gamma\varepsilon^2}, \frac{8\mathbb{E}\left[G^0\right]}{\hat{\theta}\varepsilon^2}\right\} \right\rceil,$$

where $\delta^0 = f(x^0) - f^{\inf}$. *Then, after $T$ iterations of* EF21-SGD *we have* $\mathbb{E}\left[\left\|\nabla f(\hat{x}^T)\right\|^2\right] \leq \varepsilon^2$. *It requires*

$$T = \mathcal{O}\left(\frac{\widetilde{L}\delta^0 + \mathbb{E}\left[G^0\right]}{\alpha\varepsilon^2}\right)$$

*iterations/communications rounds,*

$$\#grad_i \quad = \quad \tau_i T = \mathcal{O}\left(\frac{\widetilde{L}\delta^0 + \mathbb{E}\left[G^0\right]}{\alpha\varepsilon^2} + \frac{\left(\widetilde{L}\delta^0 + \mathbb{E}\left[G^0\right]\right)\sigma_i^2}{\alpha^3\varepsilon^4}\right)$$

*stochastic oracle calls for worker $i$, and*

$$\overline{\#grad} \quad = \quad \frac{1}{n}\sum_{i=1}^{n}\tau_i T = \mathcal{O}\left(\frac{\widetilde{L}\delta^0 + \mathbb{E}\left[G^0\right]}{\alpha\varepsilon^2} + \frac{\left(\widetilde{L}\delta^0 + \mathbb{E}\left[G^0\right]\right)\sigma^2}{\alpha^3\varepsilon^4}\right)$$

*stochastic oracle calls per worker on average, where $\sigma^2 = \frac{1}{n}\sum_{i=1}^{n}\sigma_i^2$.*

**Corollary 6.** *Consider the setting described in Example 2. Let assumptions of Theorem 3 hold, $\rho = \alpha/2, \nu = \alpha/4,$*

$$\gamma \quad = \quad \frac{1}{L + \widetilde{L}\sqrt{\frac{\hat{\beta}_1}{\hat{\theta}}}},$$

$$\tau_i \quad = \quad \left\lceil\max\left\{1, \frac{2T\gamma\overline{L}_i\hat{\beta}_2}{\hat{\theta}}, \frac{8\overline{L}_i\hat{\beta}_2}{\hat{\theta}\varepsilon^2}\delta_i^{\inf}, \frac{8\overline{L}_i\Delta_i^{\inf}\hat{\beta}_2}{\hat{\theta}\varepsilon^2}\right\}\right\rceil,$$

$$T \quad = \quad \left\lceil\max\left\{\frac{16\delta^0}{\gamma\varepsilon^2}, \frac{8\mathbb{E}\left[G^0\right]}{\hat{\theta}\varepsilon^2}\right\}\right\rceil,$$

*where $\delta_i^{\inf} = f^{\inf} - f_i^{\inf}$, $\delta^0 = f(x^0) - f^{\inf}$, $\overline{L}_i = \frac{1}{m_i}\sum_{j=1}^{m_i} L_{ij}$, $\Delta_i^{\inf} = \frac{1}{m_i}\sum_{j=1}^{m_i}(f_i^{\inf} - f_{ij}^{\inf})$.*
*Then, after $T$ iterations of* EF21-SGD *we have* $\mathbb{E}\left[\left\|\nabla f(\hat{x}^T)\right\|^2\right] \leq \varepsilon^2$. *It requires*

$$T = \mathcal{O}\left(\frac{\widetilde{L}\delta^0 + \mathbb{E}\left[G^0\right]}{\alpha\varepsilon^2}\right)$$

*iterations/communications rounds,*

$$\#grad_i \quad = \quad \tau_i T$$

$$= \quad \mathcal{O}\left(\frac{\widetilde{L}\delta^0 + \mathbb{E}\left[G^0\right]}{\alpha\varepsilon^2} + \frac{\left(\widetilde{L}\delta^0 + \mathbb{E}\left[G^0\right]\right)\left(\overline{L}_i(\delta^0 + \delta_i^{\inf}) + \overline{L}_i\Delta_i^{\inf}\right)}{\alpha^3\varepsilon^4}\right.$$

$$\left. + \frac{(\widetilde{L}\delta^0 + \mathbb{E}\left[G^0\right])\overline{L}_i\mathbb{E}\left[G^0\right]}{\alpha^2(\alpha L + \widetilde{L})\varepsilon^4}\right)$$

*stochastic oracle calls for worker $i$, and*

$$\overline{\#grad} \quad = \quad \frac{1}{n}\sum_{i=1}^{n}\tau_i T$$

$$= \quad \mathcal{O}\left(\frac{\widetilde{L}\delta^0 + \mathbb{E}\left[G^0\right]}{\alpha\varepsilon^2} + \frac{1}{n}\sum_{i=1}^{n}\frac{\left(\widetilde{L}\delta^0 + \mathbb{E}\left[G^0\right]\right)\left(\overline{L}_i(\delta^0 + \delta_i^{\inf}) + \overline{L}_i\Delta_i^{\inf}\right)}{\alpha^3\varepsilon^4}\right.$$

$$\left. + \frac{1}{n}\sum_{i=1}^{n}\frac{(\widetilde{L}\delta^0 + \mathbb{E}\left[G^0\right])\overline{L}_i\mathbb{E}\left[G^0\right]}{\alpha^2(\alpha L + \widetilde{L})\varepsilon^4}\right)$$

*stochastic oracle calls per worker on average.*

### D.2 Convergence under Polyak-Łojasiewicz Condition

**Theorem 4.** *Let Assumptions 1, 2, and 4 hold, and let the stepsize in Algorithm 2 be set as*

$$0 < \gamma \leq \min \left\{ \left( L + \widetilde{L} \sqrt{\frac{2\hat{\beta}_1}{\hat{\theta}}} \right)^{-1}, \frac{\hat{\theta}}{2\mu} \right\}, \tag{27}$$

*where $\widetilde{L} = \sqrt{\frac{1}{n} \sum_{i=1}^n L_i^2}$, $\hat{\theta} \overset{def}{=} 1 - (1-\alpha)(1+\rho)(1+\nu)$, $\hat{\beta}_1 \overset{def}{=} 2(1-\alpha)(1+\rho)\left(1+\frac{1}{\nu}\right)$, and $\rho, \nu > 0$ are some positive numbers. Assume that batchsizes $\tau_1, \ldots, \tau_i$ are such that $\frac{2\widetilde{A}\hat{\beta}_2}{\hat{\theta}} \leq \frac{\mu}{2}$, where $\widetilde{A} = \max_{i=1,\ldots,n} \frac{2(A_i + L_i(B_i - 1))}{\tau_i}$ and $\hat{\beta}_2 \overset{def}{=} 2(1-\alpha)(1+\rho)\left(1+\frac{1}{\nu}\right) + \left(1+\frac{1}{\rho}\right)$. Then for all $T \geq 1$*

$$\mathbb{E}\left[\delta^T + \frac{\gamma}{\hat{\theta}} G^T\right] \leq \left(1 - \frac{\gamma\mu}{2}\right)^T \mathbb{E}\left[\delta^0 + \frac{\gamma}{\hat{\theta}} G^0\right] + \frac{4}{\mu\hat{\theta}} \widetilde{C}\hat{\beta}_2, \tag{28}$$

*where $\widetilde{C} = \frac{1}{n} \sum_{i=1}^n \left( \frac{2(A_i + L_i(B_i - 1))}{\tau_i} \left(f^{\inf} - f_i^{\inf}\right) + \frac{C_i}{\tau_i} \right)$.*

*Proof.* We notice that inequality (20) holds for EF21-SGD as well, i.e., we have

$$\begin{aligned}
\mathbb{E}\left[\delta^{t+1}\right] &\leq \mathbb{E}\left[\delta^t\right] - \frac{\gamma}{2}\mathbb{E}\left[\|\nabla f(x^t)\|^2\right] - \left(\frac{1}{2\gamma} - \frac{L}{2}\right)\mathbb{E}\left[R^t\right] + \frac{\gamma}{2}\mathbb{E}\left[G^t\right] \\
&\overset{\text{PŁ}}{\leq} (1-\gamma\mu)\mathbb{E}\left[\delta^t\right] - \left(\frac{1}{2\gamma} - \frac{L}{2}\right)\mathbb{E}\left[R^t\right] + \frac{\gamma}{2}\mathbb{E}\left[G^t\right].
\end{aligned}$$

Summing up the above inequality with a $\frac{\gamma}{\hat{\theta}}$ multiple of (24), we derive

$$\begin{aligned}
\mathbb{E}\left[\delta^{t+1} + \frac{\gamma}{\hat{\theta}} G^{t+1}\right] &\leq (1-\gamma\mu)\mathbb{E}\left[\delta^t\right] - \left(\frac{1}{2\gamma} - \frac{L}{2}\right)\mathbb{E}\left[R^t\right] + \frac{\gamma}{2}\mathbb{E}\left[G^t\right] \\
&\quad + \frac{\gamma}{\hat{\theta}}(1-\hat{\theta})\mathbb{E}\left[G^t\right] + \frac{\gamma}{\hat{\theta}}\hat{\beta}_1\widetilde{L}^2\mathbb{E}\left[R^t\right] \\
&\quad + \frac{\gamma}{\hat{\theta}}\widetilde{A}\hat{\beta}_2\mathbb{E}\left[\delta^{t+1}\right] + \frac{\gamma}{\hat{\theta}}\widetilde{C}\hat{\beta}_2 \\
&\leq \frac{\gamma\widetilde{A}\hat{\beta}_2}{\hat{\theta}}\mathbb{E}\left[\delta^{t+1}\right] + (1-\gamma\mu)\mathbb{E}\left[\delta^t\right] + \left(1 - \frac{\hat{\theta}}{2}\right)\mathbb{E}\left[\frac{\gamma}{\hat{\theta}}G^t\right] + \frac{\gamma}{\hat{\theta}}\widetilde{C}\hat{\beta}_2 \\
&\quad - \left(\frac{1}{2\gamma} - \frac{L}{2} - \frac{\gamma\hat{\beta}_1\widetilde{L}^2}{\hat{\theta}}\right)\mathbb{E}\left[R^t\right] \\
&\overset{(27)}{\leq} \frac{\gamma\widetilde{A}\hat{\beta}_2}{\hat{\theta}}\mathbb{E}\left[\delta^{t+1}\right] + (1-\gamma\mu)\mathbb{E}\left[\delta^t + \frac{\gamma}{\hat{\theta}}G^t\right] + \frac{\gamma}{\hat{\theta}}\widetilde{C}\hat{\beta}_2,
\end{aligned}$$

where $\hat{\theta} \overset{\text{def}}{=} 1 - (1-\alpha)(1+\rho)(1+\nu)$, $\hat{\beta}_1 \overset{\text{def}}{=} 2(1-\alpha)(1+\rho)\left(1+\frac{1}{\nu}\right)$, $\hat{\beta}_2 \overset{\text{def}}{=} 2(1-\alpha)(1+\rho)\left(1+\frac{1}{\nu}\right) + \left(1+\frac{1}{\rho}\right)$, and $\rho, \nu > 0$ are some positive numbers. Next, we rearrange the terms

$$\begin{aligned}
\left(1 - \frac{\gamma\widetilde{A}\hat{\beta}_2}{\hat{\theta}}\right)\mathbb{E}\left[\delta^{t+1} + \frac{\gamma}{\hat{\theta}}G^{t+1}\right] &\leq \mathbb{E}\left[\left(1 - \frac{\gamma\widetilde{A}\hat{\beta}_2}{\hat{\theta}}\right)\delta^{t+1} + \frac{\gamma}{\hat{\theta}}G^{t+1}\right] \\
&\leq (1-\gamma\mu)\mathbb{E}\left[\delta^t + \frac{\gamma}{\hat{\theta}}G^t\right] + \frac{\gamma}{\hat{\theta}}\widetilde{C}\hat{\beta}_2
\end{aligned}$$

and divide both sides of the inequality by $\left(1 - \frac{\gamma\widetilde{A}\hat{\beta}_2}{\hat{\theta}}\right)$:

$$\begin{aligned}
\mathbb{E}\left[\delta^{t+1} + \frac{\gamma}{\hat{\theta}}G^{t+1}\right] &\leq \frac{1-\gamma\mu}{1 - \frac{\gamma\widetilde{A}\hat{\beta}_2}{\hat{\theta}}}\mathbb{E}\left[\delta^t + \frac{\gamma}{\hat{\theta}}G^t\right] + \frac{\gamma}{\hat{\theta}\left(1 - \frac{\gamma\widetilde{A}\hat{\beta}_2}{\hat{\theta}}\right)}\widetilde{C}\hat{\beta}_2 \\
&\overset{(120)}{\leq} (1-\gamma\mu)\left(1 + \frac{2\gamma\widetilde{A}\hat{\beta}_2}{\hat{\theta}}\right)\mathbb{E}\left[\delta^t + \frac{\gamma}{\hat{\theta}}G^t\right] + \left(1 + \frac{2\gamma\widetilde{A}\hat{\beta}_2}{\hat{\theta}}\right)\frac{\gamma}{\hat{\theta}}\widetilde{C}\hat{\beta}_2.
\end{aligned}$$

Since $\frac{2\widetilde{A}\hat{\beta}_2}{\hat{\theta}} \le \frac{\mu}{2}$ and $\gamma \le \frac{2}{\mu}$, we have

$$
\mathbb{E}\left[\delta^{t+1} + \frac{\gamma}{\hat{\theta}}G^{t+1}\right] \le (1-\gamma\mu)\left(1+\frac{\gamma\mu}{2}\right)\mathbb{E}\left[\delta^t + \frac{\gamma}{\hat{\theta}}G^t\right] + \frac{2\gamma}{\hat{\theta}}\widetilde{C}\hat{\beta}_2
$$

$$
\overset{(121)}{\le} \left(1-\frac{\gamma\mu}{2}\right)\mathbb{E}\left[\delta^t + \frac{\gamma}{\hat{\theta}}G^t\right] + \frac{2\gamma}{\hat{\theta}}\widetilde{C}\hat{\beta}_2.
$$

Unrolling the recurrence, we get

$$
\mathbb{E}\left[\delta^T + \frac{\gamma}{\hat{\theta}}G^T\right] \le \left(1-\frac{\gamma\mu}{2}\right)^T \mathbb{E}\left[\delta^0 + \frac{\gamma}{\hat{\theta}}G^0\right] + \frac{2\gamma}{\hat{\theta}}\widetilde{C}\hat{\beta}_2 \sum_{t=0}^{T-1}\left(1-\frac{\gamma\mu}{2}\right)^t
$$

$$
\le \left(1-\frac{\gamma\mu}{2}\right)^T \mathbb{E}\left[\delta^0 + \frac{\gamma}{\hat{\theta}}G^0\right] + \frac{2\gamma}{\hat{\theta}}\widetilde{C}\hat{\beta}_2 \sum_{t=0}^{\infty}\left(1-\frac{\gamma\mu}{2}\right)^t
$$

$$
= \left(1-\frac{\gamma\mu}{2}\right)^T \mathbb{E}\left[\delta^0 + \frac{\gamma}{\hat{\theta}}G^0\right] + \frac{4}{\mu\hat{\theta}}\widetilde{C}\hat{\beta}_2
$$

that finishes the proof. $\qquad\square$

**Corollary 7.** *Let assumptions of Theorem 4 hold, $\rho = \alpha/2$, $\nu = \alpha/4$,*

$$
\gamma = \min\left\{\frac{1}{L+\widetilde{L}\sqrt{\frac{\hat{\beta}_1}{\hat{\theta}}}}, \frac{\hat{\theta}}{2\mu}\right\},
$$

$$
\tau_i = \left\lceil \max\left\{1, \frac{8(A_i+L_i(B_i-1))\hat{\beta}_2}{\mu\hat{\theta}}, \frac{64(A_i+L_i(B_i-1))\hat{\beta}_2}{\hat{\theta}\varepsilon\mu}\delta_i^{\inf}, \frac{32C_i\hat{\beta}_2}{\hat{\theta}\varepsilon\mu}\right\}\right\rceil,
$$

$$
T = \left\lceil \frac{2}{\gamma\mu}\ln\left(\frac{2\delta^0}{\varepsilon} + \mathbb{E}\left[\frac{2\gamma G^0}{\hat{\theta}\varepsilon}\right]\right)\right\rceil,
$$

*where $\delta_i^{\inf} = f^{\inf} - f_i^{\inf}$, $\delta^0 = f(x^0) - f^{\inf}$. Then, after $T$ iterations of* EF21-SGD *we have $\mathbb{E}\left[f(x^T) - f^{\inf}\right] \le \varepsilon$. It requires*

$$
T = \mathcal{O}\left(\frac{\widetilde{L}}{\mu\alpha}\ln\left(\frac{\delta^0}{\varepsilon} + \mathbb{E}\left[\frac{2G^0}{\widetilde{L}\varepsilon}\right]\right)\right)
$$

*iterations/communications rounds,*

$$
\#grad_i = \tau_i T
$$

$$
= \mathcal{O}\left(\left(\frac{\widetilde{L}}{\mu\alpha} + \frac{\widetilde{L}\left(\hat{A}_i(\varepsilon+\delta_i^{\inf})+C_i\right)}{\mu^2\alpha^3\varepsilon}\right)\ln\left(\frac{\delta^0}{\varepsilon} + \mathbb{E}\left[\frac{2G^0}{\widetilde{L}\varepsilon}\right]\right)\right)
$$

*stochastic oracle calls for worker $i$, and*

$$
\overline{\#grad} = \frac{1}{n}\sum_{i=1}^{n}\tau_i T
$$

$$
= \mathcal{O}\left(\left(\frac{\widetilde{L}}{\mu\alpha} + \frac{1}{n}\sum_{i=1}^{n}\frac{\widetilde{L}\left(\hat{A}_i(\varepsilon+\delta_i^{\inf})+C_i\right)}{\mu^2\alpha^3\varepsilon}\right)\ln\left(\frac{\delta^0}{\varepsilon} + \mathbb{E}\left[\frac{2G^0}{\widetilde{L}\varepsilon}\right]\right)\right)
$$

*stochastic oracle calls per worker on average, where $\hat{A}_i = A_i + L_i(B_i-1)$.*

*Proof.* The given choice of $\tau_i$ ensures that $\frac{2\widetilde{A}\hat{\beta}_2}{\hat{\theta}} \le \frac{\mu}{2}$ and $4\widetilde{C}\hat{\beta}_2/\mu\hat{\theta} \le \varepsilon/2$. Next, the choice of $T$ ensures that the right-hand side of (28) is smaller than $\varepsilon$. Finally, after simple computation we get the expression for $\tau_i T$. $\qquad\square$

**Corollary 8.** *Consider the setting described in Example 1. Let assumptions of Theorem 4 hold,* $\rho = \alpha/2$, $\nu = \alpha/4$,

$$\gamma = \min\left\{\frac{1}{L + \widetilde{L}\sqrt{\frac{\hat{\beta}_1}{\hat{\theta}}}}, \frac{\hat{\theta}}{2\mu}\right\}, \quad \tau_i = \left\lceil \max\left\{1, \frac{32 C_i \hat{\beta}_2}{\hat{\theta}\varepsilon\mu}\right\}\right\rceil, \quad T = \left\lceil\frac{2}{\gamma\mu}\ln\left(\frac{2\delta^0}{\varepsilon} + \mathbb{E}\left[\frac{2\gamma G^0}{\hat{\theta}\varepsilon}\right]\right)\right\rceil,$$

*where* $\delta^0 = f(x^0) - f^{\inf}$. *Then, after* $T$ *iterations of* EF21-SGD *we have* $\mathbb{E}\left[f(x^T) - f^{\inf}\right] \le \varepsilon$. *It requires*

$$T = \mathcal{O}\left(\frac{\widetilde{L}}{\mu\alpha}\ln\left(\frac{\delta^0}{\varepsilon} + \mathbb{E}\left[\frac{2G^0}{\widetilde{L}\varepsilon}\right]\right)\right)$$

*iterations/communications rounds,*

$$\begin{aligned}
\#grad_i &= \tau_i T \\
&= \mathcal{O}\left(\left(\frac{\widetilde{L}}{\mu\alpha} + \frac{\widetilde{L}\sigma_i^2}{\mu^2\alpha^3\varepsilon}\right)\ln\left(\frac{\delta^0}{\varepsilon} + \mathbb{E}\left[\frac{2G^0}{\widetilde{L}\varepsilon}\right]\right)\right)
\end{aligned}$$

*stochastic oracle calls for worker* $i$, *and*

$$\begin{aligned}
\overline{\#grad} &= \frac{1}{n}\sum_{i=1}^{n}\tau_i T \\
&= \mathcal{O}\left(\left(\frac{\widetilde{L}}{\mu\alpha} + \frac{\widetilde{L}\sigma^2}{\mu^2\alpha^3\varepsilon}\right)\ln\left(\frac{\delta^0}{\varepsilon} + \mathbb{E}\left[\frac{2G^0}{\widetilde{L}\varepsilon}\right]\right)\right)
\end{aligned}$$

*stochastic oracle calls per worker on average, where* $\sigma^2 = \frac{1}{n}\sum_{i=1}^{n}\sigma_i^2$.

**Corollary 9.** *Consider the setting described in Example 2. Let assumptions of Theorem 4 hold,* $\rho = \alpha/2$, $\nu = \alpha/4$,

$$\begin{aligned}
\gamma &= \min\left\{\frac{1}{L + \widetilde{L}\sqrt{\frac{\hat{\beta}_1}{\hat{\theta}}}}, \frac{\hat{\theta}}{2\mu}\right\}, \\
\tau_i &= \left\lceil\max\left\{1, \frac{8\overline{L}_i\hat{\beta}_2}{\mu\hat{\theta}}, \frac{64\overline{L}_i\hat{\beta}_2}{\hat{\theta}\varepsilon\mu}\delta_i^{\inf}, \frac{64\overline{L}_i\Delta_i^{\inf}\hat{\beta}_2}{\hat{\theta}\varepsilon\mu}\right\}\right\rceil, \\
T &= \left\lceil\frac{2}{\gamma\mu}\ln\left(\frac{2\delta^0}{\varepsilon} + \mathbb{E}\left[\frac{2\gamma G^0}{\hat{\theta}\varepsilon}\right]\right)\right\rceil,
\end{aligned}$$

*where* $\delta_i^{\inf} = f^{\inf} - f_i^{\inf}$, $\delta^0 = f(x^0) - f^{\inf}$, $\overline{L}_i = \frac{1}{m_i}\sum_{j=1}^{m_i} L_{ij}$, $\Delta_i^{\inf} = \frac{1}{m_i}\sum_{j=1}^{m_i}(f_i^{\inf} - f_{ij}^{\inf})$.
*Then, after* $T$ *iterations of* EF21-SGD *we have* $\mathbb{E}\left[f(x^T) - f^{\inf}\right] \le \varepsilon$. *It requires*

$$T = \mathcal{O}\left(\frac{\widetilde{L}}{\mu\alpha}\ln\left(\frac{\delta^0}{\varepsilon} + \mathbb{E}\left[\frac{2G^0}{\widetilde{L}\varepsilon}\right]\right)\right)$$

*iterations/communications rounds,*

$$\begin{aligned}
\#grad_i &= \tau_i T \\
&= \mathcal{O}\left(\left(\frac{\widetilde{L}}{\mu\alpha} + \frac{\widetilde{L}\overline{L}_i\left(\varepsilon + \delta_i^{\inf} + \Delta_i^{\inf}\right)}{\mu^2\alpha^3\varepsilon}\right)\ln\left(\frac{\delta^0}{\varepsilon} + \mathbb{E}\left[\frac{2G^0}{\widetilde{L}\varepsilon}\right]\right)\right)
\end{aligned}$$

*stochastic oracle calls for worker* $i$, *and*

$$\begin{aligned}
\overline{\#grad} &= \frac{1}{n}\sum_{i=1}^{n}\tau_i T \\
&= \mathcal{O}\left(\left(\frac{\widetilde{L}}{\mu\alpha} + \frac{1}{n}\sum_{i=1}^{n}\frac{\widetilde{L}\overline{L}_i\left(\varepsilon + \delta_i^{\inf} + \Delta_i^{\inf}\right)}{\mu^2\alpha^3\varepsilon}\right)\ln\left(\frac{\delta^0}{\varepsilon} + \mathbb{E}\left[\frac{2G^0}{\widetilde{L}\varepsilon}\right]\right)\right)
\end{aligned}$$

*stochastic oracle calls per worker on average.*

# E   VARIANCE REDUCTION

In this part, we modify the EF21 framework to better handle *finite-sum* problems with smooth summands. Unlike the *online/streaming case* where SGD has the optimal complexity (without additional assumption on the smoothness of stochastic trajectories) (Arjevani et al., 2019), in the *finite sum* regime, it is well-known that one can hope for convergence to the exact stationary point rather than its neighborhood. To achieve this, variance reduction techniques are instrumental. One approach is to apply a PAGE-estimator (Li et al., 2021) instead of a random minibatch applied in SGD. Note that PAGE has optimal complexity for nonconvex problems of the form (3). With Corollary 10, we illustrate that this $O\left(m + \sqrt{m}/\varepsilon^2\right)$ complexity is recovered for our Algorithm 3 when no compression is applied and $n = 1$.

We show how to combine PAGE estimator with EF21 mechanism and call the new method EF21-PAGE. At each step of EF21-PAGE, clients (nodes) either compute (with probability $p$) full gradients or use a recursive estimator $v_i^t + \frac{1}{\tau_i} \sum_{j \in I_i^t} \left(\nabla f_{ij}(x^{t+1}) - \nabla f_{ij}(x^t)\right)$ (with probability $1-p$). Then each client applies a Markov compressor/EF21-estimator and sends the result to the master node. Typically the number of data points $m$ is large, and $p < 1/m$. As a result, computation of full gradients rarely happens during optimization procedure, on average, only once in every $m$ iterations.

Notice that unlike VR-MARINA (Gorbunov et al., 2021), which is a state-of-the-art distributed optimization method designed specifically for unbiased compressors and which also uses PAGE-estimator, EF21-PAGE does not require the communication of full (not compressed) vectors at all. This is an important property of the algorithm since, in some distributed networks, and especially when $d$ is very large, as is the case in modern over-parameterized deep learning, full vector communication is prohibitive. However, unlike the rate of VR-MARINA, the rate of EF21-PAGE does not improve with the growth of $n$. This is not a flaw of our method, but rather an inevitable drawback of the distributed methods that use *biased* compressions only.

**Notations for this section.** In this section, we use the following additional notations $P_i^t \overset{\text{def}}{=} \|\nabla f_i(x^t) - v_i^t\|^2$, $P^t \overset{\text{def}}{=} \frac{1}{n} \sum_{i=1}^n P_i^t$, $V_i^t \overset{\text{def}}{=} \|v_i^t - g_i^t\|^2$, $V^t \overset{\text{def}}{=} \frac{1}{n} \sum_{i=1}^n V_i^t$, where $v_i^t$ is a PAGE estimator. Recall that $G^t \overset{\text{def}}{=} \frac{1}{n} \sum_{i=1}^n G_i^t$, $G_i^t \overset{\text{def}}{=} \|\nabla f_i(x^t) - g_i^t\|^2$.

The main idea of the analysis in this section is to split the error in two parts $G_i^t \leq 2P_i^t + 2V_i^t$, and bound them separetely.

---

**Algorithm 3** EF21-PAGE

1: **Input:** starting point $x^0 \in \mathbb{R}^d$; $g_i^0, v_i^0 \in \mathbb{R}^d$ for $i = 1, \ldots, n$ (known by nodes); $g^0 = \frac{1}{n} \sum_{i=1}^n g_i^0$
   (known by master); learning rate $\gamma > 0$; probabilities $p_i \in (0,1]$; batch-sizes $1 \leq \tau_i \leq m_i$
2: **for** $t = 0, 1, 2, \ldots, T - 1$ **do**
3:     Master computes $x^{t+1} = x^t - \gamma g^t$
4:     **for all nodes** $i = 1, \ldots, n$ **in parallel do**
5:         Sample $b_i^t \sim \text{Be}(p_i)$
6:         If $b_i^t = 0$, sample a minibatch of data samples $I_i^t$ with $|I_i^t| = \tau_i$
7:         $v_i^{t+1} = \begin{cases} \nabla f_i(x^{t+1}) & \text{if } b_i^t = 1, \\ v_i^t + \frac{1}{\tau_i} \sum\limits_{j \in I_i^t} \left(\nabla f_{ij}(x^{t+1}) - \nabla f_{ij}(x^t)\right) & \text{if } b_i^t = 0 \end{cases}$
8:         Compress $c_i^t = \mathcal{C}(v_i^{t+1} - g_i^t)$ and send $c_i^t$ to the master
9:         Update local state $g_i^{t+1} = g_i^t + c_i^t$
10:    **end for**
11:    Master computes $g^{t+1} = \frac{1}{n} \sum_{i=1}^n g_i^{t+1}$ via $g^{t+1} = g^t + \frac{1}{n} \sum_{i=1}^n c_i^t$
12: **end for**
13: **Output:** $\hat{x}_T$ chosen uniformly from $\{x^t\}_{t \in [T]}$

---

**Lemma 3.** *Let Assumption 3 hold, and let $v_i^{t+1}$ be a PAGE estimator, i. e. for $b_i^t \sim \text{Be}(p_i)$*

$$v_i^{t+1} = \begin{cases} \nabla f_i(x^{t+1}) & \text{if} \quad b_i^t = 1, \\ v_i^t + \frac{1}{\tau_i} \sum\limits_{j \in I_i^t} \left(\nabla f_{ij}(x^{t+1}) - \nabla f_{ij}(x^t)\right) & \text{if} \quad b_i^t = 0, \end{cases} \tag{29}$$

*for all $i = 1, \ldots, n$, $t \geq 0$. Then*

$$\mathbb{E}\left[P^{t+1}\right] \leq (1 - p_{\min})\mathbb{E}\left[P^t\right] + \widetilde{\mathcal{L}}^2\mathbb{E}\left[\left\|x^{t+1} - x^t\right\|^2\right], \tag{30}$$

*where $\widetilde{\mathcal{L}} = \frac{1}{n}\sum_{i=1}^n \frac{(1-p_i)\mathcal{L}_i^2}{\tau_i}$, $p_{\min} = \min_{i=1,\ldots,n} p_i$.*

*Proof.*

$$
\begin{aligned}
\mathbb{E}\left[P_i^{t+1}\right] &= \mathbb{E}\left[\left\|v_i^{t+1} - \nabla f_i(x^{t+1})\right\|^2\right] \\
&= (1 - p_i)\mathbb{E}\left[\left\|v_i^t + \frac{1}{\tau_i}\sum_{j \in I_i^t}(\nabla f_{ij}(x^{t+1}) - \nabla f_{ij}(x^t)) - \nabla f_i(x^{t+1})\right\|^2\right] \\
&= (1 - p_i)\mathbb{E}\left[\left\|v_i^t - \nabla f_i(x^t) + \widetilde{\Delta}_i^t - \nabla f_i(x^{t+1}) + \nabla f_i(x^t)\right\|^2\right] \\
&= (1 - p_i)\mathbb{E}\left[\left\|v_i^t - \nabla f_i(x^t) + \widetilde{\Delta}_i^t - \Delta_i^t\right\|^2\right] \\
&\stackrel{(i)}{=} (1 - p_i)\mathbb{E}\left[\left\|v_i^t - \nabla f_i(x^t)\right\|^2\right] + (1 - p_i)\mathbb{E}\left[\left\|\widetilde{\Delta}_i^t - \Delta_i^t\right\|^2\right] \\
&\stackrel{(ii)}{\leq} (1 - p_i)\mathbb{E}\left[P_i^t\right] + \frac{(1-p_i)\mathcal{L}_i^2}{\tau_i}\mathbb{E}\left[\left\|x^{t+1} - x^t\right\|^2\right] \\
&\leq (1 - p_{\min})\mathbb{E}\left[P_i^t\right] + \frac{(1-p_i)\mathcal{L}_i^2}{\tau_i}\mathbb{E}\left[\left\|x^{t+1} - x^t\right\|^2\right], \tag{31}
\end{aligned}
$$

where equality $(i)$ holds because $\mathbb{E}\left[\widetilde{\Delta}_i^t - \Delta_i^t \mid x^t, x^{t+1}, v_i^t\right] = 0$, and $(ii)$ holds by Assumption 3. It remains to average the above inequality over $i = 1, \ldots, n$. $\qquad\square$

**Lemma 4.** *Let Assumptions 1 and 3 hold, let $v_i^{t+1}$ be a PAGE estimator, i. e. for $b_i^t \sim \mathrm{Be}(p_i)$ and for all $i = 1, \ldots, n$, $t \geq 0$*

$$v_i^{t+1} = \begin{cases} \nabla f_i(x^{t+1}) & \text{if} \quad b_i^t = 1, \\ v_i^t + \frac{1}{\tau_i}\sum_{j \in I_i^t}\left(\nabla f_{ij}(x^{t+1}) - \nabla f_{ij}(x^t)\right) & \text{if} \quad b_i^t = 0, \end{cases} \tag{32}$$

*and let $g_i^{t+1}$ be an EF21 estimator, i. e.*

$$g_i^{t+1} = g_i^t + \mathcal{C}(v_i^{t+1} - g_i^t), \quad g_i^0 = \mathcal{C}\left(v_i^0\right) \tag{33}$$

*for all $i = 1, \ldots, n$, $t \geq 0$. Then*

$$\mathbb{E}\left[V^{t+1}\right] \leq (1 - \theta)\mathbb{E}\left[V^t\right] + 2\beta p_{\max}\mathbb{E}\left[P^t\right] + \beta\left(2\widetilde{L}^2 + \widetilde{\mathcal{L}}^2\right)\mathbb{E}\left[\left\|x^{t+1} - x^t\right\|^2\right], \tag{34}$$

*where $\widetilde{\mathcal{L}} = \frac{1}{n}\sum_{i=1}^n \frac{(1-p_i)\mathcal{L}_i^2}{\tau_i}$, $p_{\max} = \max_{i=1,\ldots,n} p_i$, $\theta = 1 - (1-\alpha)(1+s)$, $\beta = (1-\alpha)\left(1 + s^{-1}\right)$ for any $s > 0$.*

*Proof.* Following the steps in proof of Lemma 1, but with $\nabla f_i(x^{t+1})$ and $\nabla f_i(x^t)$ being substituted by their estimators $v_i^{t+1}$ and $v_i^t$, we end up with an analogue of (15)

$$\mathbb{E}\left[\left\|g_i^{t+1} - v_i^{t+1}\right\|^2\right] \leq (1 - \theta)\mathbb{E}\left[\left\|g_i^t - v_i^t\right\|^2\right] + \beta\mathbb{E}\left[\left\|v_i^{t+1} - v_i^t\right\|^2\right], \tag{35}$$

where $\theta = 1 - (1-\alpha)(1+s)$, $\beta = (1-\alpha)\left(1 + s^{-1}\right)$ for any $s > 0$. Then

$$
\begin{aligned}
\mathbb{E}\left[V_i^t\right] \;&=\; \mathbb{E}\left[\left\|g_i^{t+1} - v_i^{t+1}\right\|^2\right] \\[4pt]
&\overset{(35)}{\leq}\; (1-\theta)\mathbb{E}\left[\left\|g_i^t - v_i^t\right\|^2\right] + \beta\mathbb{E}\left[\left\|v_i^{t+1} - v_i^t\right\|^2\right] \\[4pt]
&=\; (1-\theta)\mathbb{E}\left[\left\|g_i^t - v_i^t\right\|^2\right] + \beta\mathbb{E}\left[\mathbb{E}\left[\left\|v_i^{t+1} - v_i^t\right\|^2 \mid v_i^t, x^t, x^{t+1}\right]\right] \\[4pt]
&\overset{(i)}{=}\; (1-\theta)\mathbb{E}\left[V_i^t\right] + \beta p_i \mathbb{E}\left[\left\|v_i^t - \nabla f_i(x^{t+1})\right\|^2\right] \\[6pt]
&\qquad + \beta(1-p_i)\mathbb{E}\left[\left\|\frac{1}{\tau_i}\sum_{j\in I_i^t}\left(\nabla f_{ij}(x^{t+1}) - \nabla f_{ij}(x^t)\right)\right\|^2\right] \\[6pt]
&=\; (1-\theta)\mathbb{E}\left[V_i^t\right] + \beta p_i\mathbb{E}\left[\left\|v_i^t - \nabla f_i(x^{t+1})\right\|^2\right] + \beta(1-p_i)\mathbb{E}\left[\left\|\widetilde{\Delta}_i^t\right\|^2\right] \\[4pt]
&\overset{(ii)}{=}\; (1-\theta)\mathbb{E}\left[V_i^t\right] + 2\beta p_i\mathbb{E}\left[\left\|v_i^t - \nabla f_i(x^t)\right\|^2\right] \\[6pt]
&\qquad + 2\beta p_i\mathbb{E}\left[\left\|\nabla f_i(x^{t+1}) - \nabla f_i(x^t)\right\|^2\right] + \beta(1-p_i)\mathbb{E}\left[\left\|\widetilde{\Delta}_i^t\right\|^2\right] \\[4pt]
&=\; (1-\theta)\mathbb{E}\left[V_i^t\right] + 2\beta p_i\mathbb{E}\left[P_i^t\right] + 2\beta p_i\mathbb{E}\left[\left\|\Delta_i^t\right\|^2\right] + \beta(1-p_i)\mathbb{E}\left[\left\|\widetilde{\Delta}_i^t\right\|^2\right] \\[4pt]
&\overset{(iii)}{=}\; (1-\theta)\mathbb{E}\left[V_i^t\right] + 2\beta p_i\mathbb{E}\left[P_i^t\right] + \beta(2p_i + 1 - p_i)\mathbb{E}\left[\left\|\Delta_i^t\right\|^2\right] \\[6pt]
&\qquad + \beta(1-p_i)\mathbb{E}\left[\left\|\widetilde{\Delta}_i^t - \Delta_i^t\right\|^2\right] \\[4pt]
&\overset{(iv)}{\leq}\; (1-\theta)\mathbb{E}\left[V_i^t\right] + 2\beta p_i\mathbb{E}\left[P_i^t\right] + \beta(1+p_i)L_i^2\mathbb{E}\left[\left\|x^{t+1} - x^t\right\|^2\right] \\[6pt]
&\qquad + \beta\frac{(1-p_i)\mathcal{L}_i^2}{\tau_i}\mathbb{E}\left[\left\|x^{t+1} - x^t\right\|^2\right] \\[4pt]
&\leq\; (1-\theta)\mathbb{E}\left[V_i^t\right] + 2\beta p_{\max}\mathbb{E}\left[P_i^t\right] + \beta\left(2L_i^2 + \frac{(1-p_i)\mathcal{L}_i^2}{\tau_i}\right)\mathbb{E}\left[\left\|x^{t+1} - x^t\right\|^2\right],
\end{aligned}
$$

where in $(i)$ we use the definition of PAGE estimator (32), $(ii)$ applies (119) with $s = 1$, $(iii)$ is due to bias-variance decomposition (123), $(iv)$ makes use of Assumptions 1 and 3, and the last step is due to $p_i \leq 1$, $p_i \leq p_{\max}$ .

It remains to average the above inequality over $i = 1, \ldots, n$.

$\square$

### E.1   CONVERGENCE FOR GENERAL NON-CONVEX FUNCTIONS

**Theorem 5.** *Let Assumptions 1 and 3 hold, and let the stepsize in Algorithm 3 be set as*

$$
0 < \gamma \leq \left(L + \sqrt{\frac{4\beta}{\theta}\widetilde{L}^2 + 2\left(\frac{3\beta}{\theta}\frac{p_{\max}}{p_{\min}} + \frac{1}{p_{\min}}\right)\widetilde{\mathcal{L}}^2}\right)^{-1}. \tag{36}
$$

*Fix $T \geq 1$ and let $\hat{x}^T$ be chosen from the iterates $x^0, x^1, \ldots, x^{T-1}$ uniformly at random. Then*

$$
\mathbb{E}\left[\left\|\nabla f(\hat{x}^T)\right\|^2\right] \leq \frac{2\Psi^0}{\gamma T}, \tag{37}
$$

*where $\Psi^t \overset{\text{def}}{=} f(x^t) - f^{\inf} + \frac{\gamma}{\theta}V^t + \frac{\gamma}{p_{\min}}\left(1 + \frac{2\beta p_{\min}}{\theta}\right)P^t$, $p_{\max} = \max_{i=1,\ldots,n} p_i$, $p_{\min} = \min_{i=1,\ldots,n} p_i$, $\widetilde{L} = \sqrt{\frac{1}{n}\sum_{i=1}^n L_i^2}$, $\theta = 1 - (1-\alpha)(1+s)$, $\beta = (1-\alpha)\left(1 + s^{-1}\right)$ for any $s > 0$.*

*Proof.* We apply Lemma 16 and split the error $\|g_i^t - \nabla f_i(x^t)\|^2$ in two parts

$$
\begin{aligned}
f(x^{t+1}) &\leq f(x^t) - \frac{\gamma}{2}\|\nabla f(x^t)\|^2 - \left(\frac{1}{2\gamma} - \frac{L}{2}\right)R^t + \frac{\gamma}{2}\|g^t - \nabla f(x^t)\|^2 \\
&\leq f(x^t) - \frac{\gamma}{2}\|\nabla f(x^t)\|^2 - \left(\frac{1}{2\gamma} - \frac{L}{2}\right)R^t \\
&\quad + \gamma\|g^t - v^t\|^2 + \gamma\mathbb{E}\left[\|v^t - \nabla f(x^t)\|^2\right] \\
&\leq f(x^t) - \frac{\gamma}{2}\|\nabla f(x^t)\|^2 - \left(\frac{1}{2\gamma} - \frac{L}{2}\right)R^t \\
&\quad + \gamma\frac{1}{n}\sum_{i=1}^n\|g_i^t - v_i^t\|^2 + \gamma\frac{1}{n}\sum_{i=1}^n\|v_i^t - \nabla f_i(x^t)\|^2 \\
&= f(x^t) - \frac{\gamma}{2}\|\nabla f(x^t)\|^2 - \left(\frac{1}{2\gamma} - \frac{L}{2}\right)R^t + \gamma V^t + \gamma P^t, \qquad (38)
\end{aligned}
$$

where we used notation $R^t = \|\gamma g^t\|^2 = \|x^{t+1} - x^t\|^2$, and applied (118) and (119).

Subtracting $f^{\text{inf}}$ from both sides of the above inequality, taking expectation and using the notation $\delta^t = f(x^{t+1}) - f^{\text{inf}}$, we get

$$
\mathbb{E}\left[\delta^{t+1}\right] \leq \mathbb{E}\left[\delta^t\right] - \frac{\gamma}{2}\mathbb{E}\left[\|\nabla f(x^t)\|^2\right] - \left(\frac{1}{2\gamma} - \frac{L}{2}\right)\mathbb{E}\left[R^t\right] + \gamma\mathbb{E}\left[V^t\right] + \gamma\mathbb{E}\left[P^t\right]. \quad (39)
$$

Further, Lemma 3 and 4 provide the recursive bounds for the last two terms of (39)

$$
\mathbb{E}\left[P^{t+1}\right] \leq (1 - p_{\min})\mathbb{E}\left[P^t\right] + \widetilde{\mathcal{L}}^2\mathbb{E}\left[R_t\right], \qquad (40)
$$

$$
\mathbb{E}\left[V^{t+1}\right] \leq (1 - \theta)\mathbb{E}\left[V^t\right] + \beta\left(2\widetilde{L}^2 + \widetilde{\mathcal{L}}^2\right)\mathbb{E}\left[R_t\right] + 2\beta p_{\max}\mathbb{E}\left[P^t\right]. \qquad (41)
$$

Adding (39) with a $\frac{\gamma}{\theta}$ multiple of (41) we obtain

$$
\begin{aligned}
\mathbb{E}\left[\delta^{t+1}\right] + \frac{\gamma}{\theta}\mathbb{E}\left[V^{t+1}\right] &\leq \mathbb{E}\left[\delta^t\right] - \frac{\gamma}{2}\mathbb{E}\left[\|\nabla f\left(x^t\right)\|^2\right] - \left(\frac{1}{2\gamma} - \frac{L}{2}\right)\mathbb{E}\left[R^t\right] + \gamma\mathbb{E}\left[V^t\right] \\
&\quad + \gamma\mathbb{E}\left[P^t\right] + \frac{\gamma}{\theta}\left((1 - \theta)\mathbb{E}\left[V^t\right] + Ar^t + C\mathbb{E}\left[P^t\right]\right) \\
&\leq \delta^t + \frac{\gamma}{\theta}\mathbb{E}\left[V^t\right] - \frac{\gamma}{2}\mathbb{E}\left[\|\nabla f\left(x^t\right)\|^2\right] - \left(\frac{1}{2\gamma} - \frac{L}{2} - \frac{\gamma A}{\theta}\right)\mathbb{E}\left[R^t\right] \\
&\quad + \gamma\left(1 + \frac{C}{\theta}\right)\mathbb{E}\left[P^t\right],
\end{aligned}
$$

where we denote $A \stackrel{\text{def}}{=} \beta\left(2\widetilde{L}^2 + \widetilde{\mathcal{L}}^2\right), C \stackrel{\text{def}}{=} 2\beta p_{\max}$.

Then adding the above inequality with a $\frac{\gamma}{p_{\min}}\left(1 + \frac{C}{\theta}\right)$ multiple of (40), we get

$$
\begin{aligned}
\mathbb{E}\left[\Phi^{t+1}\right] &= \mathbb{E}\left[\delta^{t+1}\right] + \frac{\gamma}{\theta}\mathbb{E}\left[V^{t+1}\right] + \frac{\gamma}{p_{\min}}\left(1+\frac{C}{\theta}\right)\mathbb{E}\left[P^{t+1}\right] \\
&\leq \delta^t + \frac{\gamma}{\theta}\mathbb{E}\left[V^t\right] - \frac{\gamma}{2}\mathbb{E}\left[\left\|\nabla f\left(x^t\right)\right\|^2\right] - \left(\frac{1}{2\gamma} - \frac{L}{2} - \frac{\gamma A}{\theta}\right)\mathbb{E}\left[R^t\right] \\
&\quad + \gamma\left(1+\frac{C}{\theta}\right)\mathbb{E}\left[P^t\right] \\
&\quad + \frac{\gamma}{p_{\min}}\left(1+\frac{C}{\theta}\right)\left((1-p_{\min})\mathbb{E}\left[P^t\right] + \widetilde{\mathcal{L}}^2\mathbb{E}\left[R^t\right]\right) \\
&\leq \mathbb{E}\left[\delta^t\right] + \frac{\gamma}{\theta}\mathbb{E}\left[V^t\right] + \frac{\gamma}{p_{\min}}\left(1+\frac{C}{\theta}\right)\mathbb{E}\left[P^t\right] - \frac{\gamma}{2}\mathbb{E}\left[\left\|\nabla f\left(x^t\right)\right\|^2\right] \\
&\quad - \left(\frac{1}{2\gamma} - \frac{L}{2} - \frac{\gamma A}{\theta} - \frac{\gamma}{p_{\min}}\left(1+\frac{C}{\theta}\right)\widetilde{\mathcal{L}}^2\right)\mathbb{E}\left[R^t\right] \\
&= \mathbb{E}\left[\Phi^t\right] - \frac{\gamma}{2}\mathbb{E}\left[\left\|\nabla f\left(x^t\right)\right\|^2\right] \\
&\quad - \left(\frac{1}{2\gamma} - \frac{L}{2} - \frac{\gamma A}{\theta} - \frac{\gamma}{p_{\min}}\left(1+\frac{C}{\theta}\right)\widetilde{\mathcal{L}}^2\right)\mathbb{E}\left[R^t\right]. \qquad (42)
\end{aligned}
$$

The coefficient in front of $\mathbb{E}\left[R^t\right]$ simplifies after substitution by $A$ and $C$

$$
\frac{\gamma A}{\theta} + \frac{\gamma}{p_{\min}}\left(1+\frac{C}{\theta}\right)\widetilde{\mathcal{L}}^2 \leq \frac{2\beta}{\theta}\widetilde{L}^2 + \left(\frac{3\beta}{\theta}\frac{p_{\max}}{p_{\min}} + \frac{1}{p_{\min}}\right)\widetilde{\mathcal{L}}^2.
$$

Thus by Lemma 15 and the stepsize choice

$$
0 < \gamma \leq \left(L + \sqrt{\frac{4\beta}{\theta}\widetilde{L}^2 + 2\left(\frac{3\beta}{\theta}\frac{p_{\max}}{p_{\min}} + \frac{1}{p_{\min}}\right)\widetilde{\mathcal{L}}^2}\right)^{-1} \qquad (43)
$$

the last term in (42) is not positive. By summing up inequalities for $t = 0, \ldots, T-1$, we get

$$
0 \leq \mathbb{E}\left[\Phi^T\right] \leq \mathbb{E}\left[\Phi^0\right] - \frac{\gamma}{2}\sum_{t=0}^{T-1}\mathbb{E}\left[\left\|\nabla f(x^t)\right\|^2\right].
$$

Multiplying both sides by $\frac{2}{\gamma T}$ and rearranging we get

$$
\sum_{t=0}^{T-1}\frac{1}{T}\mathbb{E}\left[\left\|\nabla f(x^t)\right\|^2\right] \leq \frac{2\mathbb{E}\left[\Phi^0\right]}{\gamma T}.
$$

It remains to notice that the left hand side can be interpreted as $\mathbb{E}\left[\left\|\nabla f(\hat{x}^T)\right\|^2\right]$, where $\hat{x}^T$ is chosen from $x^0, x^1, \ldots, x^{T-1}$ uniformly at random.

$\square$

**Corollary 10.** *Let assumptions of Theorem 5 hold,*

$$
\begin{aligned}
v_i^0 &= g_i^0 = \nabla f_i(x^0), \quad i = 1, \ldots, n, \\
\gamma &= \left(L + \sqrt{\frac{4\beta}{\theta}\widetilde{L}^2 + 2\left(\frac{3\beta}{\theta}\frac{p_{\max}}{p_{\min}} + \frac{1}{p_{\min}}\right)\widetilde{\mathcal{L}}^2}\right)^{-1}, \\
p_i &= \frac{\tau_i}{\tau_i + m_i}, \quad i = 1, \ldots, n.
\end{aligned}
$$

*Then, after $T$ iterations/communication rounds of* EF21-PAGE *we have* $\mathbb{E}\left[\left\|\nabla f(\hat{x}^T)\right\|^2\right] \leq \varepsilon^2$. *It requires*

$$T = \mathcal{O}\left(\frac{(\widetilde{L} + \widetilde{\mathcal{L}})\delta^0}{\alpha\varepsilon^2}\sqrt{\frac{p_{\max}}{p_{\min}}} + \frac{\sqrt{m_{\max}}\widetilde{\mathcal{L}}\delta^0}{\varepsilon^2}\right)$$

*iterations/communications rounds,*

$$\#grad_i = \mathcal{O}\left(m_i + \frac{\tau_i(\widetilde{L} + \widetilde{\mathcal{L}})\delta^0}{\alpha\varepsilon^2}\sqrt{\frac{p_{\max}}{p_{\min}}} + \frac{\tau_i\sqrt{m_{\max}}\widetilde{\mathcal{L}}\delta^0}{\varepsilon^2}\right)$$

*stochastic oracle calls for worker $i$, and*

$$\overline{\#grad} = \mathcal{O}\left(m + \frac{\tau(\widetilde{L} + \widetilde{\mathcal{L}})\delta^0}{\alpha\varepsilon^2}\sqrt{\frac{p_{\max}}{p_{\min}}} + \frac{\tau\sqrt{m_{\max}}\widetilde{\mathcal{L}}\delta^0}{\varepsilon^2}\right)$$

*stochastic oracle calls per worker on average, where $\tau = \frac{1}{n}\sum_{i=1}^n \tau_i$, $m = \frac{1}{n}\sum_{i=1}^n m_i$, $m_{\max} = \max_{i=1,\dots,n} m_i$, $p_{\max} = \max_{i=1,\dots,n} p_i$, $p_{\min} = \min_{i=1,\dots,n} p_i$.*

*Proof.* Notice that by Lemma 17 we have

$$
\begin{aligned}
L + \sqrt{\frac{4\beta}{\theta}\widetilde{L}^2 + 2\left(\frac{3\beta}{\theta}\frac{p_{\max}}{p_{\min}} + \frac{1}{p_{\min}}\right)\widetilde{\mathcal{L}}^2} &\leq L + \sqrt{\frac{16}{\alpha^2}\widetilde{L}^2 + 2\left(\frac{12}{\alpha^2}\frac{p_{\max}}{p_{\min}} + \frac{1}{p_{\min}}\right)\widetilde{\mathcal{L}}^2} \\
&\leq L + \frac{4}{\alpha}\widetilde{L} + \sqrt{\frac{24}{\alpha^2}\frac{p_{\max}}{p_{\min}} + \frac{2}{p_{\min}}}\widetilde{\mathcal{L}} \\
&\leq L + \frac{4}{\alpha}\widetilde{L} + \frac{\sqrt{24}}{\alpha}\sqrt{\frac{p_{\max}}{p_{\min}}}\widetilde{\mathcal{L}} + \frac{\sqrt{2}}{\sqrt{p_{\min}}}\widetilde{\mathcal{L}} \\
&\leq \frac{5}{\alpha}\widetilde{L} + \frac{\sqrt{24}}{\alpha}\sqrt{\frac{p_{\max}}{p_{\min}}}\widetilde{\mathcal{L}} + \frac{\sqrt{2}}{\sqrt{p_{\min}}}\widetilde{\mathcal{L}} \\
&\leq \frac{5}{\alpha}\sqrt{\frac{p_{\max}}{p_{\min}}}\left(\widetilde{L} + \widetilde{\mathcal{L}}\right) + \frac{\sqrt{2}}{\sqrt{p_{\min}}}\widetilde{\mathcal{L}} \\
&\leq \frac{5}{\alpha}\sqrt{\frac{p_{\max}}{p_{\min}}}\left(\widetilde{L} + \widetilde{\mathcal{L}}\right) + 2\sqrt{m_{\max}}\widetilde{\mathcal{L}},
\end{aligned}
$$

where we used $L \leq \widetilde{L}$, $p_{\min} \leq p_{\max}$, and the fact that $\sqrt{a+b} \leq \sqrt{a} + \sqrt{b}$ for $a, b \geq 0$.

Then the number of communication rounds

$$
\begin{aligned}
T &\leq \frac{2\delta^0}{\gamma\varepsilon^2} \\
&\leq \frac{2\delta^0}{\varepsilon^2}\left(\frac{5}{\alpha}\sqrt{\frac{p_{\max}}{p_{\min}}}\left(\widetilde{L} + \widetilde{\mathcal{L}}\right) + 2\sqrt{m_{\max}}\widetilde{\mathcal{L}}\right) \\
&= \mathcal{O}\left(\frac{(\widetilde{L} + \widetilde{\mathcal{L}})\delta^0}{\alpha\varepsilon^2}\sqrt{\frac{p_{\max}}{p_{\min}}} + \frac{\sqrt{m_{\max}}\widetilde{\mathcal{L}}\delta^0}{\varepsilon^2}\right).
\end{aligned}
$$

At each worker, we have

$$
\begin{aligned}
\#grad_i &= m_i + T\left(p_i m_i + (1 - p_i)\tau_i\right) \\
&= m_i + \frac{2m_i\tau_i}{\tau_i + m_i}T \\
&\leq m_i + 2\tau_i T.
\end{aligned}
$$

Averaging over $i = 1,\dots,n$, we get

$$
\begin{aligned}
\overline{\#\text{grad}} &\leq m + 2\tau T \\
&= \mathcal{O}\left( m + \frac{\tau(\widetilde{L} + \widetilde{\mathcal{L}})\delta^0}{\alpha\varepsilon^2}\sqrt{\frac{p_{\max}}{p_{\min}}} + \frac{\tau\sqrt{m_{\max}}\widetilde{\mathcal{L}}\delta^0}{\varepsilon^2}\right).
\end{aligned}
$$

$\square$

### E.2 CONVERGENCE UNDER POLYAK-ŁOJASIEWICZ CONDITION

**Theorem 6.** *Let Assumptions 1 and 4 hold, and let the stepsize in Algorithm 3 be set as*

$$
0 < \gamma \leq \min\left\{ \gamma_0, \frac{\theta}{2\mu}, \frac{p_{\min}}{2\mu} \right\}, \tag{44}
$$

*where* $\gamma_0 \overset{def}{=} 0 < \gamma \leq \left( L + \sqrt{\frac{8\beta}{\theta}\widetilde{L}^2 + 4\left(\frac{5\beta}{\theta}\frac{p_{\max}}{p_{\min}} + \frac{1}{p_{\min}}\right)\widetilde{\mathcal{L}}^2}\right)^{-1}$, $\widetilde{L} = \sqrt{\frac{1}{n}\sum_{i=1}^n L_i^2}$, $\theta = 1 - (1-\alpha)(1+s)$, $\beta = (1-\alpha)\left(1 + s^{-1}\right)$ *for any* $s > 0$.
*Let* $\Psi^t \overset{def}{=} f(x^t) - f(x^\star) + \frac{2\gamma}{\theta}V^t + \frac{2\gamma}{p_{\min}}\left(1 + \frac{4\beta p_{\max}}{\theta}\right)P^t$. *Then for any* $T \geq 0$, *we have*

$$
\mathbb{E}\left[\Psi^T\right] \leq (1 - \gamma\mu)^T\mathbb{E}\left[\Psi^0\right]. \tag{45}
$$

*Proof.* Similarly to the proof of Theorem 5 the inequalities (39), (40), (41) hold with $\delta^t = f(x^t) - f(x^\star)$.

Adding (39) with a $\frac{2\gamma}{\theta}$ multiple of (41) we obtain

$$
\begin{aligned}
\mathbb{E}\left[\delta^{t+1}\right] + \frac{2\gamma}{\theta}\mathbb{E}\left[V^{t+1}\right] &\leq \mathbb{E}\left[\delta^t\right] - \frac{\gamma}{2}\mathbb{E}\left[\left\|\nabla f\left(x^t\right)\right\|^2\right] - \left(\frac{1}{2\gamma} - \frac{L}{2}\right)\mathbb{E}\left[R^t\right] + \gamma\mathbb{E}\left[V^t\right] \\
&\quad + \gamma\mathbb{E}\left[P^t\right] + \frac{2\gamma}{\theta}\left((1-\theta)\mathbb{E}\left[V^t\right] + Ar^t + C\mathbb{E}\left[P^t\right]\right) \\
&\leq \delta^t + \frac{2\gamma}{\theta}\mathbb{E}\left[V^t\right]\left(1 - \frac{\theta}{2}\right) \\
&\quad - \frac{\gamma}{2}\mathbb{E}\left[\left\|\nabla f\left(x^t\right)\right\|^2\right] - \left(\frac{1}{2\gamma} - \frac{L}{2} - \frac{2\gamma A}{\theta}\right)\mathbb{E}\left[R^t\right] \\
&\quad + \gamma\left(1 + \frac{2C}{\theta}\right)\mathbb{E}\left[P^t\right],
\end{aligned}
$$

where $A \overset{def}{=} \beta\left(2\widetilde{L}^2 + \widetilde{\mathcal{L}}^2\right)$, $C \overset{def}{=} 2\beta p_{\max}$.

Then adding the above inequality with a $\frac{2\gamma}{p_{\min}}\left(1 + \frac{2C}{\theta}\right)$ multiple of (40), we get

$$
\begin{aligned}
\mathbb{E}\left[\Psi^{t+1}\right] &= \mathbb{E}\left[\delta^{t+1}\right] + \frac{2\gamma}{\theta}\mathbb{E}\left[V^{t+1}\right] + \frac{2\gamma}{p_{\min}}\left(1 + \frac{2C}{\theta}\right)\mathbb{E}\left[P^{t+1}\right] \\
&\leq \delta^t + \frac{\gamma}{\theta}\mathbb{E}\left[V^t\right]\left(1 - \frac{\theta}{2}\right) - \frac{\gamma}{2}\mathbb{E}\left[\left\|\nabla f\left(x^t\right)\right\|^2\right] - \left(\frac{1}{2\gamma} - \frac{L}{2} - \frac{2\gamma A}{\theta}\right)\mathbb{E}\left[R^t\right] \\
&\quad + \gamma\left(1 + \frac{2C}{\theta}\right)\mathbb{E}\left[P^t\right] \\
&\quad + \frac{2\gamma}{p_{\min}}\left(1 + \frac{2C}{\theta}\right)\left((1 - p_{\min})\mathbb{E}\left[P^t\right] + \widetilde{\mathcal{L}}^2\mathbb{E}\left[R^t\right]\right) \\
&\leq \mathbb{E}\left[\delta^t\right] + \frac{2\gamma}{\theta}\mathbb{E}\left[V^t\right]\left(1 - \frac{\theta}{2}\right) + \frac{2\gamma}{p_{\min}}\left(1 + \frac{2C}{\theta}\right)\mathbb{E}\left[P^t\right]\left(1 - \frac{p_{\min}}{2}\right) \\
&\quad - \frac{\gamma}{2}\mathbb{E}\left[\left\|\nabla f\left(x^t\right)\right\|^2\right] - \left(\frac{1}{2\gamma} - \frac{L}{2} - \frac{2\gamma A}{\theta} - \frac{2\gamma}{p_{\min}}\left(1 + \frac{2C}{\theta}\right)\widetilde{\mathcal{L}}^2\right)\mathbb{E}\left[R^t\right].
\end{aligned}
$$

(46)

PL inequality implies that $\delta^t - \frac{\gamma}{2}\|\nabla f(x^t)\|^2 \leq (1 - \gamma\mu)\delta^t$. In view of the above inequality and our assumption on the stepsize ( $\gamma \leq \frac{\theta}{2\mu}, \gamma \leq \frac{p_{\min}}{2\mu}$ ), we get

$$
\mathbb{E}\left[\Psi^{t+1}\right] \leq (1 - \gamma\mu)\mathbb{E}\left[\Psi^t\right] - \left(\frac{1}{2\gamma} - \frac{L}{2} - \frac{2\gamma A}{\theta} - \frac{2\gamma}{p_{\min}}\left(1 + \frac{2C}{\theta}\right)\widetilde{\mathcal{L}}^2\right)\mathbb{E}\left[R^t\right].
$$

The coefficient in front of $\mathbb{E}\left[R^t\right]$ simplifies after substitution by $A$ and $C$

$$
\begin{aligned}
\frac{2\gamma A}{\theta} + \frac{2\gamma}{p_{\min}}\left(1 + \frac{2C}{\theta}\right)\widetilde{\mathcal{L}}^2 &= \frac{4\beta}{\theta}\widetilde{L}^2 + \left(\frac{2\beta}{\theta} + \frac{2}{p_{\min}} + \frac{8\beta}{\theta}\frac{p_{\max}}{p_{\min}}\right)\widetilde{\mathcal{L}}^2 \\
&\leq \frac{4\beta}{\theta}\widetilde{L}^2 + 2\left(\frac{5\beta}{\theta}\frac{p_{\max}}{p_{\min}} + \frac{1}{p_{\min}}\right)\widetilde{\mathcal{L}}^2.
\end{aligned}
$$

Thus by Lemma 15 and the stepsize choice

$$
0 < \gamma \leq \left(L + \sqrt{\frac{8\beta}{\theta}\widetilde{L}^2 + 4\left(\frac{5\beta}{\theta}\frac{p_{\max}}{p_{\min}} + \frac{1}{p_{\min}}\right)\widetilde{\mathcal{L}}^2}\right)^{-1}
$$

(47)

the last term in (46) is not positive.

$$
\mathbb{E}\left[\Psi^{t+1}\right] \leq (1 - \gamma\mu)\mathbb{E}\left[\Psi^t\right].
$$

It remains to unroll the recurrence.

$\qquad\qquad\qquad\qquad\qquad\qquad\qquad\qquad\qquad\qquad\qquad\qquad\qquad\qquad\qquad\qquad\square$

**Corollary 11.** *Let assumptions of Theorem 6 hold,*

$$
\begin{aligned}
v_i^0 &= g_i^0 = \nabla f_i(x^0), \quad i = 1, \ldots, n, \\
\gamma &= \min\left\{\gamma_0, \frac{\theta}{2\mu}, \frac{p_{\min}}{2\mu}\right\}, \qquad \gamma_0 = \left(L + \sqrt{\frac{8\beta}{\theta}\widetilde{L}^2 + 4\left(\frac{5\beta}{\theta}\frac{p_{\max}}{p_{\min}} + \frac{1}{p_{\min}}\right)\widetilde{\mathcal{L}}^2}\right)^{-1}, \\
p_i &= \frac{\tau_i}{\tau_i + m_i}, \quad i = 1, \ldots, n.
\end{aligned}
$$

*Then, after $T$ iterations of* EF21-PAGE *we have $\mathbb{E}\left[f(x^T) - f^{\inf}\right] \leq \varepsilon$. It requires*

$$
T = \mathcal{O}\left(\frac{1}{\mu}\left(\frac{\widetilde{L} + \widetilde{\mathcal{L}}}{\alpha}\sqrt{\frac{p_{\max}}{p_{\min}}} + \sqrt{m_{\max}}\widetilde{\mathcal{L}}\right)\ln\left(\frac{\delta^0}{\varepsilon}\right)\right)
$$

*iterations/communications rounds,*

$$\#grad_i \;=\; \mathcal{O}\left(m_i + \frac{\tau_i}{\mu}\left(\frac{\widetilde{L} + \widetilde{\mathcal{L}}}{\alpha}\sqrt{\frac{p_{\max}}{p_{\min}}} + \sqrt{m_{\max}}\widetilde{\mathcal{L}}\right)\ln\left(\frac{\delta^0}{\varepsilon}\right)\right)$$

*stochastic oracle calls for worker i, and*

$$\overline{\#grad} \;=\; \mathcal{O}\left(m + \frac{\tau}{\mu}\left(\frac{\widetilde{L} + \widetilde{\mathcal{L}}}{\alpha}\sqrt{\frac{p_{\max}}{p_{\min}}} + \sqrt{m_{\max}}\widetilde{\mathcal{L}}\right)\ln\left(\frac{\delta^0}{\varepsilon}\right)\right)$$

*stochastic oracle calls per worker on average, where* $\tau = \frac{1}{n}\sum_{i=1}^{n}\tau_i$, $m = \frac{1}{n}\sum_{i=1}^{n}m_i$, $m_{\max} = \max_{i=1,\ldots,n} m_i$, $p_{\max} = \max_{i=1,\ldots,n} p_i$, $p_{\min} = \min_{i=1,\ldots,n} p_i$.

# F    PARTIAL PARTICIPATION

In this section, we provide an option for partial participation of the clients – a feature important in federated learning. Most of the works in compressed distributed optimization deal with full worker participation, i.e., the case when all clients are involved in computation and communication at every iteration. However, in the practice of federated learning, only a subset of clients are allowed to participate at each training round. This limitation comes mainly due to the following two reasons. First, clients (e.g., mobile devices) may wish to join or leave the network randomly. Second, it is often prohibitive to wait for all available clients since stragglers can significantly slow down the training process. Although many existing works (Gorbunov et al., 2021; Horváth & Richtárik, 2021; Philippenko & Dieuleveut, 2020; Karimireddy et al., 2020; Yang et al., 2021; Cho et al., 2020) allow for partial participation, they assume either unbiased compressors or no compression at all. We provide a simple analysis of partial participation, which works with *biased compressors* and builds upon the EF21 mechanism.

The modified method (Algorithm 4) is called EF21-PP . At each iteration of EF21-PP , the master samples a subset $S_t$ of clients (nodes), which are required to perform computation. Note, that all other clients (nodes) $i \notin S_t$ participate neither in the computation nor in communication at iteration $t$.

We allow for an arbitrary sampling strategy of a subset $S_t$ at the master node. The only requirement is that $\mathbf{Prob}\,(i \in S_t) = p_i > 0$ for all $i = 1, \ldots, n$, which is often referred to as a *proper arbitrary* sampling.[9] Clearly, many poplular sampling procedures fell into this setting, for instance, independent sampling with/without replacement, $\tau$-nice sampling. We do not discuss particular sampling strategies here, more on samplings can be found in (Qu & Richtárik, 2014).

---

**Algorithm 4** EF21-PP (EF21 with partial participation)

1: **Input:** starting point $x^0 \in \mathbb{R}^d$; $g_i^0 \in \mathbb{R}^d$ for $i = 1, \ldots, n$ (known by nodes); $g^0 = \frac{1}{n}\sum_{i=1}^n g_i^0$ (known by master); learning rate $\gamma > 0$
2: **for** $t = 0, 1, 2, \ldots, T - 1$ **do**
3:     Master computes $x^{t+1} = x^t - \gamma g^t$
4:     Master samples a subset $S_t$ of nodes ($|S_t| \leq n$) such that $\mathbf{Prob}\,(i \in S_t) = p_i$
5:     Master broadcasts $x^{t+1}$ to the nodes with $i \in S_t$
6:     **for all nodes** $i = 1, \ldots, n$ **in parallel do**
7:         **if** $i \in S_t$ **then**
8:             Compress $c_i^t = \mathcal{C}(\nabla f_i(x^{t+1}) - g_i^t)$ and send $c_i^t$ to the master
9:             Update local state $g_i^{t+1} = g_i^t + c_i^t$
10:        **end if**
11:        **if** $i \notin S_t$ **then**
12:            Do not change local state $g_i^{t+1} = g_i^t$
13:        **end if**
14:    **end for**
15:    Master updates $g_i^{t+1} = g_i^t$, $c_i^t = 0$ for $i \notin S_t$
16:    Master computes $g^{t+1} = \frac{1}{n}\sum_{i=1}^n g_i^{t+1}$ via $g^{t+1} = g^t + \frac{1}{n}\sum_{i=1}^n c_i^t$
17: **end for**

---

**Lemma 5.** *Then for Algorithm 4 holds*

$$\mathbb{E}\left[G^{t+1}\right] \leq (1 - \theta_p)\mathbb{E}\left[G^t\right] + B\mathbb{E}\left[\left\|x^{t+1} - x^t\right\|^2\right] \tag{48}$$

*with* $\theta_p \stackrel{def}{=} \rho p_{min} + \theta p_{max} - \rho - (p_{max} - p_{min})$, $B \stackrel{def}{=} \frac{1}{n}\sum_{i=1}^n \left(\beta p_i + \left(1 + \rho^{-1}\right)(1 - p_i)\right) L_i^2$, $p_{max} \stackrel{def}{=} \max_{1 \leq i \leq n} p_i$, $p_{min} \stackrel{def}{=} \min_{1 \leq i \leq n} p_i$, $\theta = 1 - (1 + s)(1 - \alpha)$, $\beta = \left(1 + \frac{1}{s}\right)(1 - \alpha)$ *and small enough* $\rho, s > 0$.

*Proof.* By (13) in Lemma 1, we have for all $i \in S_t$

$$\mathbb{E}\left[G_i^{t+1} \mid i \in S_t\right] \leq (1 - \theta)\mathbb{E}\left[G_i^t\right] + \beta L_i^2 \mathbb{E}\left[\left\|x^{t+1} - x^t\right\|^2 \mid i \in S_t\right] \tag{49}$$

---

[9]It is natural to focus on *proper* samplings only since otherwise there is a node $i$, which never communicaties. This would be a critical issue when trying to minimize (1) as we do not assume any similarity between $f_i(\cdot)$.

with $\theta = 1 - (1+s)(1-\alpha)$, $\beta = \left(1 + \frac{1}{s}\right)(1-\alpha)$ and arbitrary $s > 0$.

Define $W^t \stackrel{\text{def}}{=} \{g_1^t, \ldots, g_n^t, x^t, x^{t+1}\}$ and let $i \notin S_t$, then

$$
\begin{aligned}
\mathbb{E}\left[G_i^{t+1} \mid i \notin S_t\right] &= \mathbb{E}\left[\mathbb{E}\left[G_i^{t+1} \mid W^t\right] \mid i \notin S_t\right] \\
&= \mathbb{E}\left[\mathbb{E}\left[\left\|g_i^{t+1} - \nabla f_i(x^{t+1})\right\|^2 \mid W^t\right] \mid i \notin S_t\right] \\
&\leq (1+\rho)\mathbb{E}\left[\mathbb{E}\left[\left\|g_i^t - \nabla f_i(x^t)\right\|^2 \mid W^t\right] \mid i \notin S_t\right] \\
&\quad + \left(1+\rho^{-1}\right)\mathbb{E}\left[\mathbb{E}\left[\left\|\nabla f_i(x^{t+1}) - \nabla f_i(x^t)\right\|^2 \mid W^t\right] \mid i \notin S_t\right] \\
&\leq (1+\rho)\mathbb{E}\left[G_i^t\right] \\
&\quad + \left(1+\rho^{-1}\right)\mathbb{E}\left[\left\|\nabla f_i(x^{t+1}) - \nabla f_i(x^t)\right\|^2 \mid i \notin S_t\right] \\
&\leq (1+\rho)\mathbb{E}\left[G_i^t\right] + \left(1+\rho^{-1}\right)L_i^2\mathbb{E}\left[\left\|x^{t+1} - x^t\right\|^2\right]. \quad (50)
\end{aligned}
$$

Combining (49) and (50), we get

$$
\begin{aligned}
\mathbb{E}\left[G^{t+1}\right] &= \frac{1}{n}\sum_{i=1}^n \mathbb{E}\left[G_i^{t+1}\right] \\
&= \frac{1}{n}\sum_{i=1}^n p_i \mathbb{E}\left[G_i^{t+1} \mid i \in S_t\right] + \frac{1}{n}\sum_{i=1}^n (1-p_i)\mathbb{E}\left[G_i^{t+1} \mid i \notin S_t\right] \\
&\stackrel{(49),(50)}{\leq} (1-\theta)\frac{1}{n}\sum_{i=1}^n p_i \mathbb{E}\left[G_i^t\right] + \beta\left(\frac{1}{n}\sum_{i=1}^n p_i L_i^2\right)\mathbb{E}\left[\left\|x^{t+1} - x^t\right\|^2\right] \\
&\quad + (1+\rho)\frac{1}{n}\sum_{i=1}^n (1-p_i)\mathbb{E}\left[G_i^t\right] \\
&\quad + \left(1+\rho^{-1}\right)\left(\frac{1}{n}\sum_{i=1}^n (1-p_i)L_i^2\right)\mathbb{E}\left[\left\|x^{t+1} - x^t\right\|^2\right] \\
&\stackrel{(i)}{\leq} (1-\theta)p_{max}\frac{1}{n}\sum_{i=1}^n \mathbb{E}\left[G_i^t\right] + \beta\left(\frac{1}{n}\sum_{i=1}^n p_i L_i^2\right)\mathbb{E}\left[\left\|x^{t+1} - x^t\right\|^2\right] \\
&\quad + (1+\rho)(1-p_{min})\frac{1}{n}\sum_{i=1}^n \mathbb{E}\left[G_i^t\right] \\
&\quad + \left(1+\rho^{-1}\right)\left(\frac{1}{n}\sum_{i=1}^n (1-p_i)L_i^2\right)\mathbb{E}\left[\left\|x^{t+1} - x^t\right\|^2\right] \\
&= \left((1-\theta)p_{max} + (1+\rho)(1-p_{min})\right)\mathbb{E}\left[G^t\right] \\
&\quad + \left(\frac{1}{n}\sum_{i=1}^n \left(\beta p_i + \left(1+\rho^{-1}\right)(1-p_i)\right)L_i^2\right)\mathbb{E}\left[\left\|x^{t+1} - x^t\right\|^2\right] \\
&= \left(1 - \left(\rho p_{min} + \theta p_{max} - \rho - (p_{max} - p_{min})\right)\right)\mathbb{E}\left[G^t\right] \\
&\quad + \left(\frac{1}{n}\sum_{i=1}^n \left(\beta p_i + \left(1+\rho^{-1}\right)(1-p_i)\right)L_i^2\right)\mathbb{E}\left[\left\|x^{t+1} - x^t\right\|^2\right]. \\
&= (1-\theta_p)\mathbb{E}\left[G^t\right] + B\mathbb{E}\left[\left\|x^{t+1} - x^t\right\|^2\right],
\end{aligned}
$$

$\square$

**Lemma 6.** *[To simplify the rates for partial participation] Let $B$ and $\theta_p$ be defined as in Theorems 7 and Theorems 8, and let $p_i = p > 0$ for all $i = 1, \ldots, n$. Then there exist $\rho, s > 0$ such that*

$$\theta_p \geq \frac{p\alpha}{2}, \tag{51}$$

$$0 < \frac{B}{\theta_p} \leq \left(\frac{4\widetilde{L}}{p\alpha}\right)^2. \tag{52}$$

*Proof.* Under the assumption that $p_i = p$ for all $i = 1, \ldots, n$, the constants simplify to

$$\theta_p = \rho p + \theta p - \rho,$$

$$B = \left(\beta p + \left(1 + \rho^{-1}\right)(1 - p)\right)\widetilde{L}^2,$$

$$p_{max} = p_{min} = p.$$

*Case I:* let $\alpha = 1, p = 1$, then the result holds trivially.

*Case II:* let $0 < \alpha < 1, p = 1$, then $B = \beta\widetilde{L}^2$, $\theta_p = \theta = 1 - \sqrt{1 - \alpha} \geq \frac{\alpha}{2}$ and (52) follows by Lemma 17.

*Case III:* let $\alpha = 1$, and $0 < p < 1$, then $\theta = 1$, $\beta = 0$, $B = \left(1 + \rho^{-1}\right)(1 - p)\widetilde{L}^2$, $\theta_p = p - \rho(1 - p)$. Then the choice $\rho = \frac{p\alpha}{2(1-p)}$ simplifies

$$\theta_p = \frac{p}{2},$$

$$\frac{B}{\theta_p} = \frac{\left(1 + \rho^{-1}\right)(1 - p)\widetilde{L}^2}{p - \rho(1 - p)} = \frac{2(1 - p)\widetilde{L}^2}{p}\left(\frac{2}{p} - 1\right) \leq \frac{4\widetilde{L}^2}{p^2}.$$

*Case IV:* let $0 < \alpha < 1$, and $0 < p < 1$. Then the choice of constants $\theta = 1 - (1 - \alpha)(1 + s)$, $\beta = (1 - \alpha)\left(1 + \frac{1}{s}\right)$, $\rho = \frac{p\alpha}{4(1-p)}$, $s = \frac{\alpha}{4(1-\alpha)}$ yields

$$\begin{aligned}
p\rho + \theta p - \rho &= p(\rho + 1 - (1 - \alpha)(1 + s)) - \rho \\
&= p\alpha - p(1 - \alpha)s - (1 - p)\rho \\
&= \frac{1}{2}p\alpha. \tag{53}
\end{aligned}$$

Also

$$1 + \frac{1}{s} = \frac{4 - 3\alpha}{\alpha} \leq \frac{4}{\alpha}, \quad 1 + \frac{1}{\rho} = \frac{4(1 - p) + \alpha p}{p\alpha} = \frac{4 - p(4 - \alpha)}{p\alpha} \leq \frac{4}{p\alpha}.$$

Thus

$$\begin{aligned}
\frac{B}{\theta_p} = \frac{p\beta + (1 - p)\left(1 + \frac{1}{\rho}\right)}{p(\rho + \theta) - \rho}\widetilde{L}^2 &= \frac{p(1 - \alpha)\left(1 + \frac{1}{s}\right) + (1 - p)\left(1 + \frac{1}{\rho}\right)}{\frac{1}{2}p\alpha}\widetilde{L}^2 \\
&\leq \frac{p(1 - \alpha)\frac{4}{\alpha} + (1 - p)\frac{4}{p\alpha}}{\frac{1}{2}p\alpha}\widetilde{L}^2 \\
&\leq \frac{\frac{4}{\alpha} + \frac{4}{p\alpha}}{\frac{1}{2}p\alpha}\widetilde{L}^2 \\
&\leq \frac{\frac{8}{p\alpha}}{\frac{1}{2}p\alpha}\widetilde{L}^2 \\
&\leq \frac{16\widetilde{L}^2}{p^2\alpha^2}. \tag{54}
\end{aligned}$$

$\square$

### F.1 Convergence for General Non-Convex Functions

**Theorem 7.** *Let Assumption 1 hold, and let the stepsize in Algorithm 4 be set as*

$$0 < \gamma \leq \left( L + \sqrt{\frac{B}{\theta_p}} \right)^{-1}. \tag{55}$$

*Fix $T \geq 1$ and let $\hat{x}^T$ be chosen from the iterates $x^0, x^1, \ldots, x^{T-1}$ uniformly at random. Then*

$$\mathbb{E}\left[\left\|\nabla f(\hat{x}^T)\right\|^2\right] \leq \frac{2\left(f(x^0) - f^{\inf}\right)}{\gamma T} + \frac{\mathbb{E}\left[G^0\right]}{\theta_p T} \tag{56}$$

*with $\theta_p = \rho p_{min} + \theta p_{max} - \rho - (p_{max} - p_{min})$, $B = \frac{1}{n}\sum_{i=1}^{n}\left(\beta p_i + \left(1 + \rho^{-1}\right)(1 - p_i)\right)L_i^2$, $p_{max} = \max_{1 \leq i \leq n} p_i$, $p_{min} = \min_{1 \leq i \leq n} p_i$, $\theta = 1 - (1 + s)(1 - \alpha)$, $\beta = \left(1 + \frac{1}{s}\right)(1 - \alpha)$ and $\rho, s > 0$.*

*Proof.* By (20), we have

$$\mathbb{E}\left[\delta^{t+1}\right] \leq \mathbb{E}\left[\delta^t\right] - \frac{\gamma}{2}\mathbb{E}\left[\left\|\nabla f(x^t)\right\|^2\right] - \left(\frac{1}{2\gamma} - \frac{L}{2}\right)\mathbb{E}\left[R^t\right] + \frac{\gamma}{2}\mathbb{E}\left[G^t\right]. \tag{57}$$

Lemma 5 states that

$$\mathbb{E}\left[G^{t+1}\right] \leq (1 - \theta_p)\mathbb{E}\left[G^t\right] + B\mathbb{E}\left[R^t\right] \tag{58}$$

with $\theta_p = \rho p_{min} + \theta p_{max} - \rho - (p_{max} - p_{min})$, $B = \frac{1}{n}\sum_{i=1}^{n}\left(\beta p_i + \left(1 + \rho^{-1}\right)(1 - p_i)\right)L_i^2$, $p_{max} = \max_{1 \leq i \leq n} p_i$, $p_{min} = \min_{1 \leq i \leq n} p_i$, $\theta = 1 - (1 + s)(1 - \alpha)$, $\beta = \left(1 + \frac{1}{s}\right)(1 - \alpha)$ and small enough $\rho, s > 0$.

Adding (57) with a $\frac{\gamma}{2\theta_2}$ multiple of (58) and rearranging terms in the right hand side, we have

$$\mathbb{E}\left[\delta^{t+1}\right] + \frac{\gamma}{2\theta_p}\mathbb{E}\left[G^{t+1}\right] \leq \mathbb{E}\left[\delta^t\right] + \frac{\gamma}{2\theta_p}\mathbb{E}\left[G^t\right]$$

$$- \frac{\gamma}{2}\mathbb{E}\left[\left\|\nabla f(x^t)\right\|^2\right] - \left(\frac{1}{2\gamma} - \frac{L}{2} - \frac{\gamma B}{2\theta}\right)\mathbb{E}\left[R^t\right]$$

$$\leq \mathbb{E}\left[\delta^t\right] + \frac{\gamma}{2\theta_p}\mathbb{E}\left[G^t\right] - \frac{\gamma}{2}\mathbb{E}\left[\left\|\nabla f(x^t)\right\|^2\right].$$

The last inequality follows from the bound $\gamma^2 \frac{B}{\theta_p} + L\gamma \leq 1$, which holds because of Lemma 15 and our assumption on the stepsize. By summing up inequalities for $t = 0, \ldots, T-1$, we get

$$0 \leq \mathbb{E}\left[\delta^T + \frac{\gamma}{2\theta}G^T\right] \leq \delta^0 + \frac{\gamma}{2\theta}\mathbb{E}\left[G^0\right] - \frac{\gamma}{2}\sum_{t=0}^{T-1}\mathbb{E}\left[\left\|\nabla f(x^t)\right\|^2\right].$$

Multiplying both sides by $\frac{2}{\gamma T}$, after rearranging we get

$$\sum_{t=0}^{T-1}\frac{1}{T}\mathbb{E}\left[\left\|\nabla f(x^t)\right\|^2\right] \leq \frac{2\delta^0}{\gamma T} + \frac{\mathbb{E}\left[G^0\right]}{\theta T}.$$

It remains to notice that the left hand side can be interpreted as $\mathbb{E}\left[\left\|\nabla f(\hat{x}^T)\right\|^2\right]$, where $\hat{x}^T$ is chosen from $x^0, x^1, \ldots, x^{T-1}$ uniformly at random.

$\square$

**Corollary 12.** *Let assumptions of Theorem 7 hold,*

$$g_i^0 = \nabla f_i(x^0), \qquad i = 1, \ldots, n,$$

$$\gamma = \left(L + \sqrt{\frac{B}{\theta_p}}\right)^{-1},$$

$$p_i = p, \qquad i = 1, \ldots, n,$$

where $B$ and $\theta_p$ are given in Theorem 7. Then, after $T$ iterations/communication rounds of EF21-PP we have $\mathbb{E}\left[\left\|\nabla f(\hat{x}^T)\right\|^2\right] \leq \varepsilon^2$. It requires

$$T = \#grad = \mathcal{O}\left(\frac{\widetilde{L}\delta^0}{p\alpha\varepsilon^2}\right) \tag{59}$$

iterations/communications rounds/gradint computations at each node.

*Proof.* Let $g_i^0 = \nabla f_i(x^0)$, $i = 1, \ldots, n$ , then $G^0 = 0$ and by Theorem 7

$$
\begin{aligned}
\#\text{grad} &= T \overset{(i)}{\leq} \frac{2\delta^0}{\gamma\varepsilon^2} \overset{(ii)}{\leq} \frac{2\delta^0}{\varepsilon^2}\left(L + \widetilde{L}\sqrt{\frac{B}{\theta_p}}\right) \overset{(iii)}{\leq} \frac{2\delta^0}{\varepsilon^2}\left(L + \frac{4\widetilde{L}}{p\alpha}\right) \\
&\leq \frac{2\delta^0}{\varepsilon^2}\left(L + \frac{4\widetilde{L}}{p\alpha}\right) \overset{(iv)}{\leq} \frac{2\delta^0}{\varepsilon^2}\left(\frac{\widetilde{L}}{p\alpha} + \frac{4\widetilde{L}}{p\alpha}\right) = \frac{5\widetilde{L}\delta^0}{p\alpha\varepsilon^2},
\end{aligned}
$$

where $(i)$ is due to the rate (56) given by Theorem 7. In two $(ii)$ we use the largest possible stepsize (55), in $(iii)$ we utilize Lemma 6, and $(iv)$ follows by the inequalities $\alpha \leq 1$, $p \leq 1$ and $L \leq \widetilde{L}$. $\quad\square$

### F.2 CONVERGENCE UNDER POLYAK-ŁOJASIEWICZ CONDITION

**Theorem 8.** *Let Assumptions 1 and 4 hold, and let the stepsize in Algorithm 4 be set as*

$$0 < \gamma \leq \min\left\{\left(L + \sqrt{\frac{2B}{\theta_p}}\right)^{-1}, \frac{\theta_p}{2\mu}\right\}. \tag{60}$$

*Let $\Psi^t \overset{def}{=} f(x^t) - f(x^\star) + \frac{\gamma}{\theta_p}G^t$. Then for any $T \geq 0$, we have*

$$\mathbb{E}\left[\Psi^T\right] \leq (1 - \gamma\mu)^T \mathbb{E}\left[\Psi^0\right] \tag{61}$$

*with $\theta_p = \rho p_{min} + \theta p_{max} - \rho - (p_{max} - p_{min})$, $B = \frac{1}{n}\sum_{i=1}^n \left(\beta p_i + \left(1 + \rho^{-1}\right)(1 - p_i)\right)L_i^2$, $p_{max} = \max_{1 \leq i \leq n} p_i$, $p_{min} = \min_{1 \leq i \leq n} p_i$, $\theta = 1 - (1 + s)(1 - \alpha)$, $\beta = \left(1 + \frac{1}{s}\right)(1 - \alpha)$ and $\rho, s > 0$.*

*Proof.* Following the same steps as in the proof of Theorem 2, but using (58), and assumption on the stepsize (60), we obtain the result. $\quad\square$

**Corollary 13.** *Let assumptions of Theorem 8 hold,*

$$
\begin{aligned}
g_i^0 &= \nabla f_i(x^0), \qquad i = 1, \ldots, n, \\
\gamma &= \min\left\{\left(L + \sqrt{\frac{2B}{\theta_p}}\right)^{-1}, \frac{\theta_p}{2\mu}\right\}, \\
p_i &= p, \qquad i = 1, \ldots, n,
\end{aligned}
$$

*where $B$ and $\theta_p$ are given in Theorem 8. Then, after $T$ iterations/communication rounds of EF21-PP we have $\mathbb{E}\left[f(x^T) - f(x^\star)\right] \leq \varepsilon$. It requires*

$$T = \#grad = \mathcal{O}\left(\frac{\widetilde{L}}{p\alpha\mu}\log\left(\frac{\delta^0}{\varepsilon}\right)\right) \tag{62}$$

*iterations/communications rounds/gradint computations at each node.*

*Proof.* The proof is the same as for Corollary 3. The only difference is that Lemma 6 is needed to upper bound the quantities $1/\theta_p$ and $B/\theta_p$, which appear in Theorem 8. $\quad\square$

## G  BIDIRECTIONAL COMPRESSION

In the majority of applications, the uplink (**C**lient $\rightarrow$ **S**erver) communication is the bottleneck. However, in some settings the downlink (**S**erver $\rightarrow$ **C**lient) communication can also slowdown training. Tang et al. (2020) construct a mechanism which allows bidirectional biased compression. Their method builds upon the original EF meachanism and they prove $\mathcal{O}\left(\frac{1}{T^{2/3}}\right)$ rate for general nonconvex objectives. However, the main defficiency of this approach is that it requires an additional assumption of *bounded magnitude of error* (there exists $\Delta > 0$ such that $\mathbb{E}\left[\|\mathcal{C}(x) - x\|^2\right] \leq \Delta$ for all $x$). In this section, we lift this limitation and propose a new method EF21-BC (Algorithm 5), which enjoys the desirable $\mathcal{O}\left(\frac{1}{T}\right)$, and does not rely on additional assumptions.

---

**Algorithm 5** EF21-BC (EF21 with bidirectional biased compression)

---

1: **Input:** starting point $x^0 \in \mathbb{R}^d$; $g^0$, $b^0$, $\widetilde{g}_i^0 \in \mathbb{R}^d$ for $i = 1, \ldots, n$ (known by nodes); $\widetilde{g}^0 = \frac{1}{n}\sum_{i=1}^n \widetilde{g}_i^0$ (known by master) ; learning rate $\gamma > 0$
2: **for** $t = 0,1,2,\ldots,T-1$ **do**
3:      Master updates $x^{t+1} = x^t - \gamma g^t$
4:      **for all nodes** $i = 1, \ldots, n$ **in parallel do**
5:          Update $x^{t+1} = x^t - \gamma g^t$, $g^{t+1} = g^t + b^t$,
6:          compress $c_i^t = \mathcal{C}_w(\nabla f_i(x^{t+1}) - \widetilde{g}_i^t)$, send $c_i^t$ to the master, and
7:          update local state $\widetilde{g}_i^{t+1} = \widetilde{g}_i^t + c_i^t$
8:      **end for**
9:      Master computes $\widetilde{g}^{t+1} = \frac{1}{n}\sum_{i=1}^n \widetilde{g}_i^{t+1}$ via $\widetilde{g}^{t+1} = \widetilde{g}^t + \frac{1}{n}\sum_{i=1}^n c_i^t$,
10:      compreses $b^{t+1} = \mathcal{C}_M(\widetilde{g}^{t+1} - g^t)$, broadcast $b^{t+1}$ to workers ,
11:      and updates $g^{t+1} = g^t + b^{t+1}$
12: **end for**

---

Note that $\mathcal{C}_M$ and $\mathcal{C}_w$ stand for contractive compressors of the type 1 of master and workers respectively. In general, different $\alpha_M$ and $\alpha_w$ are accepted.

**Notations for this section:** $P_i^t \overset{\text{def}}{=} \|\widetilde{g}_i^t - \nabla f_i(x^t)\|^2$, $P^t \overset{\text{def}}{=} \frac{1}{n}\sum_{i=1}^n P_i^t$.

**Lemma 7.** *Let Assumption 1 hold, $\mathcal{C}_w$ be a contractive compressor, and $\widetilde{g}_i^{t+1}$ be an EF21 estimator of $\nabla f_i(x^{t+1})$, i. e.*

$$\widetilde{g}_i^{t+1} = \widetilde{g}_i^t + \mathcal{C}_w(\nabla f_i(x^{t+1}) - \widetilde{g}_i^t) \tag{63}$$

*for arbitrary $\widetilde{g}_i^0$ and all all $i = 1, \ldots, n$, $t \geq 0$. Then*

$$\mathbb{E}\left[P^{t+1}\right] \leq (1 - \theta_w)\mathbb{E}\left[P^t\right] + \beta_w \widetilde{L}^2 \mathbb{E}\left[R^t\right], \tag{64}$$

*where $\theta_w \overset{\text{def}}{=} 1 - (1 - \alpha_w)(1 + s)$,    $\beta_w \overset{\text{def}}{=} (1 - \alpha_w)\left(1 + s^{-1}\right)$   for any $s > 0$.*

*Proof.* The proof is the same as for Lemma 1. $\qquad\square$

**Lemma 8.** *Let Assumption 1 hold, $\mathcal{C}_M$, $\mathcal{C}_w$ be contractive compressors. Let $\widetilde{g}_i^{t+1}$ be an EF21 estimator of $\nabla f_i(x^{t+1})$, i. e.*

$$\widetilde{g}_i^{t+1} = \widetilde{g}_i^t + \mathcal{C}_w(\nabla f_i(x^{t+1}) - \widetilde{g}_i^t), \tag{65}$$

*and let $g^{t+1}$ be an EF21 estimator of $\widetilde{g}^{t+1} = \frac{1}{n}\sum_{i=1}^n \widetilde{g}_i^{t+1}$, i. e.*

$$g^{t+1} = g^t + \mathcal{C}_M(\widetilde{g}^{t+1} - g^t) \tag{66}$$

*for arbitrary $g^0$, $\widetilde{g}_i^0$ and all $i = 1, \ldots, n$, $t \geq 0$. Then*

$$\mathbb{E}\left[\|g^{t+1} - \widetilde{g}^{t+1}\|^2\right] \leq (1 - \theta_M)\mathbb{E}\left[\|g^t - \widetilde{g}^t\|^2\right] + 8\beta_M \mathbb{E}\left[P^t\right] + 8\beta_M \widetilde{L}^2 \mathbb{E}\left[R^t\right], \tag{67}$$

*where $g^t = \frac{1}{n}\sum_{i=1}^n g_i^t$, $\widetilde{g}^t = \frac{1}{n}\sum_{i=1}^n \widetilde{g}_i^t$, $\theta_M = 1 - (1 - \alpha_M)(1 + \rho)$, $\beta_M = (1 - \alpha_M)\left(1 + \rho^{-1}\right)$ for any $\rho > 0$.*

*Proof.* Similarly to the proof of Lemma 1, define $W^t \overset{\text{def}}{=} \{g_1^t, \ldots, g_n^t, x^t, x^{t+1}\}$ and

$$
\begin{aligned}
\mathbb{E}\left[\left\|g^{t+1} - \widetilde{g}^{t+1}\right\|^2\right] &= \mathbb{E}\left[\mathbb{E}\left[\left\|g^{t+1} - \widetilde{g}^{t+1}\right\|^2 \mid W^t\right]\right] \\
&= \mathbb{E}\left[\mathbb{E}\left[\left\|g^t + \mathcal{C}_M(\widetilde{g}^{t+1} - g^t) - \widetilde{g}^{t+1}\right\|^2 \mid W^t\right]\right] \\
&\overset{(8)}{\leq} (1 - \alpha_M)\mathbb{E}\left[\left\|\widetilde{g}^{t+1} - g^t\right\|^2\right] \\
&\overset{(i)}{\leq} (1 - \alpha_M)(1 + \rho)\mathbb{E}\left[\left\|\widetilde{g}^t - g^t\right\|^2\right] \\
&\qquad + (1 - \alpha_M)\left(1 + \rho^{-1}\right)\left\|\widetilde{g}^{t+1} - \widetilde{g}^t\right\|^2 \\
&= (1 - \theta_M)\mathbb{E}\left[\left\|g^t - \widetilde{g}^t\right\|^2\right] + \beta_M\left\|\widetilde{g}^{t+1} - \widetilde{g}^t\right\|^2,
\end{aligned}
\tag{68}
$$

where $(i)$ follows by Young's inequality (118), and in $(ii)$ we use the definition of $\theta_M$ and $\beta_M$.

Further we bound the last term in (68). Recall that

$$
\widetilde{g}^{t+1} = \widetilde{g}^t + \frac{1}{n}\sum_{i=1}^{n} c_i^t.
\tag{69}
$$

where $c_i^t = \mathcal{C}_w(\nabla f_i(x^{t+1}) - \widetilde{g}_i^t)$ and $\widetilde{g}^t = \frac{1}{n}\sum_{i=1}^{n}\widetilde{g}_i^t$. Then

$$
\begin{aligned}
\mathbb{E}\left[\left\|\widetilde{g}^{t+1} - \widetilde{g}^t\right\|^2\right] &\overset{(69)}{=} \mathbb{E}\left[\left\|\widetilde{g}^t + \frac{1}{n}\sum_{i=1}^{n} c_i^t - \widetilde{g}^t\right\|^2\right] \\
&= \mathbb{E}\left[\left\|\frac{1}{n}\sum_{i=1}^{n} c_i^t\right\|^2\right] \\
&\overset{(i)}{\leq} \frac{1}{n}\sum_{i=1}^{n}\mathbb{E}\left[\left\|c_i^t\right\|^2\right] \\
&= \frac{1}{n}\sum_{i=1}^{n}\mathbb{E}\left[\left\|c_i^t - \left(\nabla f_i(x^{t+1}) - \widetilde{g}_i^t\right) + \left(\nabla f_i(x^{t+1}) - \widetilde{g}_i^t\right)\right\|^2\right] \\
&\overset{(118)}{\leq} 2\frac{1}{n}\sum_{i=1}^{n}\mathbb{E}\left[\mathbb{E}\left[\left\|\mathcal{C}_w\left(\nabla f_i(x^{t+1}) - \widetilde{g}_i^t\right) - \left(\nabla f_i(x^{t+1}) - \widetilde{g}_i^t\right)\right\|^2 \mid W^t\right]\right] \\
&\qquad + 2\frac{1}{n}\sum_{i=1}^{n}\mathbb{E}\left[\left\|\nabla f_i(x^{t+1}) - \widetilde{g}_i^t\right\|^2\right] \\
&\overset{(8)}{\leq} 2(1 - \alpha_w)\frac{1}{n}\sum_{i=1}^{n}\mathbb{E}\left[\left\|\nabla f_i(x^{t+1}) - \widetilde{g}_i^t\right\|^2\right] + 2\frac{1}{n}\sum_{i=1}^{n}\mathbb{E}\left[\left\|\nabla f_i(x^{t+1}) - \widetilde{g}_i^t\right\|^2\right] \\
&= 2(2 - \alpha_w)\frac{1}{n}\sum_{i=1}^{n}\mathbb{E}\left[\left\|\nabla f_i(x^{t+1}) - \widetilde{g}_i^t\right\|^2\right] \\
&\overset{(ii)}{<} 4\frac{1}{n}\sum_{i=1}^{n}\mathbb{E}\left[\left\|\nabla f_i(x^{t+1}) - \widetilde{g}_i^t\right\|^2\right] \\
&= 4\frac{1}{n}\sum_{i=1}^{n}\mathbb{E}\left[\left\|\nabla f_i(x^{t+1}) - \nabla f_i(x^t) - \left(\widetilde{g}_i^t - \nabla f_i(x^t)\right)\right\|^2\right] \\
&\overset{(118)}{\leq} 8\frac{1}{n}\sum_{i=1}^{n}\left\|\widetilde{g}_i^t - \nabla f_i(x^t)\right\|^2 + 8\frac{1}{n}\sum_{i=1}^{n}\mathbb{E}\left[\left\|\nabla f_i(x^{t+1}) - \nabla f_i(x^t)\right\|^2\right] \\
&\overset{(iii)}{\leq} 8\frac{1}{n}\sum_{i=1}^{n}\mathbb{E}\left[\left\|\widetilde{g}_i^t - \nabla f_i(x^t)\right\|^2\right] + 8\widetilde{L}^2\mathbb{E}\left[\left\|x^{t+1} - x^t\right\|^2\right] \\
&= 8\mathbb{E}\left[P^t\right] + 8\widetilde{L}^2\mathbb{E}\left[R^t\right],
\end{aligned}
\tag{70}
$$

where in $(i)$ we use (119), $(ii)$ is due to $\alpha_w > 0$, $(iii)$ holds by Assumption 1. In the last step we apply the definition of $P^t = \frac{1}{n}\sum_{i=1}^{n}\|\widetilde{g}_i^t - \nabla f_i(x^t)\|^2$, and $R^t = \|x^{t+1} - x^t\|^2$

Finally, plugging (70) into (68), we conclude the proof. □

### G.1 Convergence for General Non-Convex Functions

**Theorem 9.** *Let Assumption 1 hold, and let the stepsize in Algorithm 5 be set as*

$$0 < \gamma \leq \left( L + \widetilde{L}\sqrt{\frac{16\beta_M}{\theta_M} + \frac{2\beta_w}{\theta_w}\left(1 + \frac{8\beta_M}{\theta_M}\right)} \right)^{-1} \tag{71}$$

*Fix $T \geq 1$ and let $\hat{x}^T$ be chosen from the iterates $x^0, x^1, \dots, x^{T-1}$ uniformly at random. Then*

$$\mathbb{E}\left[\|\nabla f(\hat{x}^T)\|^2\right] \leq \frac{2\mathbb{E}\left[\Psi^0\right]}{\gamma T}, \tag{72}$$

*where $\Psi^t \overset{def}{=} f(x^t) - f^{\inf} + \frac{\gamma}{\theta_M}\|g^t - \widetilde{g}^t\|^2 + \frac{\gamma}{\theta_w}\left(1 + \frac{8\beta_M}{\theta_M}\right)P^t$, $\widetilde{L} = \sqrt{\frac{1}{n}\sum_{i=1}^{n} L_i^2}$, $\theta_w \overset{def}{=} 1 - (1 - \alpha_w)(1+s)$, $\beta_w \overset{def}{=} (1 - \alpha_w)\left(1 + s^{-1}\right)$, $\theta_M \overset{def}{=} 1 - (1 - \alpha_M)(1 + \rho)$, $\beta_M \overset{def}{=} (1 - \alpha_M)\left(1 + \rho^{-1}\right)$ for any $\rho, s > 0$.*

*Proof.* We apply Lemma 16 and split the error $\|g^t - \nabla f(x^t)\|^2$ in two parts

$$
\begin{aligned}
f(x^{t+1}) &\leq f(x^t) - \frac{\gamma}{2}\|\nabla f(x^t)\|^2 - \left(\frac{1}{2\gamma} - \frac{L}{2}\right)R^t + \frac{\gamma}{2}\|g^t - \nabla f(x^t)\|^2 \\
&\leq f(x^t) - \frac{\gamma}{2}\|\nabla f(x^t)\|^2 - \left(\frac{1}{2\gamma} - \frac{L}{2}\right)R^t \\
&\quad + \gamma\|g^t - \widetilde{g}^t\|^2 + \gamma\|\widetilde{g}^t - \nabla f(x^t)\|^2 \\
&\leq f(x^t) - \frac{\gamma}{2}\|\nabla f(x^t)\|^2 - \left(\frac{1}{2\gamma} - \frac{L}{2}\right)R^t \\
&\quad + \gamma\frac{1}{n}\sum_{i=1}^{n}\|g^t - \widetilde{g}^t\|^2 + \gamma\frac{1}{n}\sum_{i=1}^{n}\|\widetilde{g}_i^t - \nabla f_i(x^t)\|^2 \\
&= f(x^t) - \frac{\gamma}{2}\|\nabla f(x^t)\|^2 - \left(\frac{1}{2\gamma} - \frac{L}{2}\right)R^t + \gamma\|g^t - \widetilde{g}^t\|^2 + \gamma P^t, \tag{73}
\end{aligned}
$$

where we used notation $R^t = \|\gamma g^t\|^2 = \|x^{t+1} - x^t\|^2$, $P^t = \frac{1}{n}\sum_{i=1}^{n}\|\widetilde{g}_i^t - \nabla f_i(x^t)\|^2$ and applied (118) and (119).

Subtracting $f^{\inf}$ from both sides of the above inequality, taking expectation and using the notation $\delta^t = f(x^{t+1}) - f^{\inf}$, we get

$$\mathbb{E}\left[\delta^{t+1}\right] \leq \mathbb{E}\left[\delta^t\right] - \frac{\gamma}{2}\mathbb{E}\left[\|\nabla f(x^t)\|^2\right] - \left(\frac{1}{2\gamma} - \frac{L}{2}\right)\mathbb{E}\left[R^t\right] + \gamma\mathbb{E}\left[\|g^t - \widetilde{g}^t\|^2\right] + \gamma\mathbb{E}\left[P^t\right]. \tag{74}$$

Further, Lemma 7 and 8 provide the recursive bounds for the last two terms of (74)

$$\mathbb{E}\left[P^{t+1}\right] \leq (1 - \theta_w)\mathbb{E}\left[P^t\right] + \beta_w\widetilde{L}^2\mathbb{E}\left[R_t\right], \tag{75}$$

$$\mathbb{E}\left[\|g^{t+1} - \widetilde{g}^{t+1}\|^2\right] \leq (1 - \theta_M)\mathbb{E}\left[\|g^t - \widetilde{g}^t\|^2\right] + 8\beta_M\widetilde{L}^2\mathbb{E}\left[R_t\right] + 8\beta_M\mathbb{E}\left[P^t\right]. \tag{76}$$

Summing up (74) with a $\frac{\gamma}{\theta_M}$ multiple of (76) we obtain

$$
\begin{aligned}
\mathbb{E}\left[\delta^{t+1}\right] + \frac{\gamma}{\theta_M}\mathbb{E}\left[\left\|g^{t+1} - \widetilde{g}^{t+1}\right\|^2\right] \leq{} & \mathbb{E}\left[\delta^t\right] - \frac{\gamma}{2}\mathbb{E}\left[\left\|\nabla f\left(x^t\right)\right\|^2\right] - \left(\frac{1}{2\gamma} - \frac{L}{2}\right)\mathbb{E}\left[R^t\right] \\
& + \gamma\mathbb{E}\left[\left\|g^t - \widetilde{g}^t\right\|^2\right] + \gamma\mathbb{E}\left[P^t\right] \\
& + \frac{\gamma}{\theta_M}\left((1 - \theta_M)\mathbb{E}\left[\left\|g^t - \widetilde{g}^t\right\|^2\right]\right) \\
& + \frac{\gamma}{\theta_M}\left(8\beta_M\widetilde{L}^2\mathbb{E}\left[R^t\right] + 8\beta_M\mathbb{E}\left[P^t\right]\right) \\
\leq{} & \mathbb{E}\left[\delta^t\right] + \frac{\gamma}{\theta_M}\mathbb{E}\left[\left\|g^t - \widetilde{g}^t\right\|^2\right] - \frac{\gamma}{2}\mathbb{E}\left[\left\|\nabla f\left(x^t\right)\right\|^2\right] \\
& - \left(\frac{1}{2\gamma} - \frac{L}{2} - \frac{8\gamma\beta_M\widetilde{L}^2}{\theta_M}\right)\mathbb{E}\left[R^t\right] \\
& + \gamma\left(1 + \frac{8\beta_M}{\theta_M}\right)\mathbb{E}\left[P^t\right].
\end{aligned}
$$

Then adding the above inequality with a $\frac{\gamma}{\theta_w}\left(1 + \frac{8\beta_M}{\theta_M}\right)$ multiple of (75), we get

$$
\begin{aligned}
\mathbb{E}\left[\Psi^{t+1}\right] ={} & \mathbb{E}\left[\delta^{t+1}\right] + \frac{\gamma}{\theta_M}\mathbb{E}\left[\left\|g^{t+1} - \widetilde{g}^{t+1}\right\|^2\right] + \frac{\gamma}{\theta_w}\left(1 + \frac{8\beta_M}{\theta_M}\right)\mathbb{E}\left[P^{t+1}\right] \\
\leq{} & \mathbb{E}\left[\delta^t\right] + \frac{\gamma}{\theta_M}\mathbb{E}\left[\left\|g^t - \widetilde{g}^t\right\|^2\right] - \frac{\gamma}{2}\mathbb{E}\left[\left\|\nabla f\left(x^t\right)\right\|^2\right] - \left(\frac{1}{2\gamma} - \frac{L}{2} - \frac{8\gamma\beta_M\widetilde{L}^2}{\theta_M}\right)\mathbb{E}\left[R^t\right] \\
& + \gamma\left(1 + \frac{8\beta_M}{\theta_M}\right)\mathbb{E}\left[P^t\right] \\
& + \frac{\gamma}{\theta_w}\left(1 + \frac{8\beta_M}{\theta_M}\right)\left((1 - \theta_w)\mathbb{E}\left[P^t\right] + \beta_w\widetilde{L}^2\mathbb{E}\left[R^t\right]\right) \\
\leq{} & \mathbb{E}\left[\delta^t\right] + \frac{\gamma}{\theta_M}\mathbb{E}\left[\left\|g^t - \widetilde{g}^t\right\|^2\right] + \frac{\gamma}{\theta_w}\left(1 + \frac{8\beta_M}{\theta_M}\right)\mathbb{E}\left[P^t\right] - \frac{\gamma}{2}\mathbb{E}\left[\left\|\nabla f\left(x^t\right)\right\|^2\right] \\
& - \left(\frac{1}{2\gamma} - \frac{L}{2} - \frac{8\gamma\beta_M\widetilde{L}^2}{\theta_M} - \frac{\gamma}{\theta_w}\left(1 + \frac{8\beta_M}{\theta_M}\right)\beta_w\widetilde{L}^2\right)\mathbb{E}\left[R^t\right] \\
={} & \mathbb{E}\left[\Psi^t\right] - \frac{\gamma}{2}\mathbb{E}\left[\left\|\nabla f\left(x^t\right)\right\|^2\right] \\
& - \left(\frac{1}{2\gamma} - \frac{L}{2} - \frac{8\gamma\beta_M\widetilde{L}^2}{\theta_M} - \frac{\gamma\beta_w\widetilde{L}^2}{\theta_w}\left(1 + \frac{8\beta_M}{\theta_M}\right)\right)\mathbb{E}\left[R^t\right]. \quad (77)
\end{aligned}
$$

Thus by Lemma 15 and the stepsize choice

$$
0 < \gamma \leq \left(L + \widetilde{L}\sqrt{\frac{16\beta_M}{\theta_M} + \frac{2\beta_w}{\theta_w}\left(1 + \frac{8\beta_M}{\theta_M}\right)}\right)^{-1} \quad (78)
$$

the last term in (77) is not positive. By summing up inequalities for $t = 0, \ldots, T - 1$, we get

$$
0 \leq \mathbb{E}\left[\Psi^T\right] \leq \mathbb{E}\left[\Psi^0\right] - \frac{\gamma}{2}\sum_{t=0}^{T-1}\mathbb{E}\left[\left\|\nabla f(x^t)\right\|^2\right].
$$

Multiplying both sides by $\frac{2}{\gamma T}$ and rearranging we get

$$
\sum_{t=0}^{T-1}\frac{1}{T}\mathbb{E}\left[\left\|\nabla f(x^t)\right\|^2\right] \leq \frac{2\mathbb{E}\left[\Psi^0\right]}{\gamma T}.
$$

It remains to notice that the left hand side can be interpreted as $\mathbb{E}\left[\left\|\nabla f(\hat{x}^T)\right\|^2\right]$, where $\hat{x}^T$ is chosen from $x^0, x^1, \ldots, x^{T-1}$ uniformly at random.

$\square$

**Corollary 14.** *Let assumption of Theorem 9 hold,*

$$g^0 = \nabla f(x^0), \qquad \widetilde{g}_i^0 = \nabla f_i(x^0), \qquad i = 1, \ldots, n,$$

$$\gamma = \left(L + \widetilde{L}\sqrt{\frac{16\beta_M}{\theta_M} + \frac{2\beta_w}{\theta_w}\left(1 + \frac{8\beta_M}{\theta_M}\right)}\right)^{-1},$$

*Then, after $T$ iterations/communication rounds of* EF21-BC *we have* $\mathbb{E}\left[\left\|\nabla f(\hat{x}^T)\right\|^2\right] \leq \varepsilon^2$. *It requires*

$$T = \#grad = \mathcal{O}\left(\frac{\widetilde{L}\delta^0}{\alpha_w \alpha_M \varepsilon^2}\right) \tag{79}$$

*iterations/communications rounds/gradint computations at each node.*

*Proof.* Note that by Lemma 17 and $\alpha_M, \alpha_w \leq 1$, we have

$$\frac{16\beta_M}{\theta_M} + \frac{2\beta_w}{\theta_w}\left(1 + \frac{8\beta_M}{\theta_M}\right) \leq 16\frac{4}{\alpha_M^2} + 2\frac{4}{\alpha_w^2}\left(1 + 8\frac{4}{\alpha_M^2}\right)$$

$$\leq \frac{64}{\alpha_M^2} + \frac{8}{\alpha_w^2}\frac{33}{\alpha_M^2}$$

$$\leq \frac{64 + 8 \cdot 33}{\alpha_w^2 \alpha_M^2}.$$

It remains to apply the steps similar to those in the proof of Corollary 2. $\square$

### G.2 CONVERGENCE UNDER POLYAK-ŁOJASIEWICZ CONDITION

**Theorem 10.** *Let Assumptions 1 and 4 hold, and let the stepsize in Algorithm 3 be set as*

$$0 < \gamma \leq \min\left\{\gamma_0, \frac{\theta_M}{2\mu}, \frac{\theta_w}{2\mu}\right\}, \tag{80}$$

*where* $\gamma_0 \overset{def}{=} \left(L + \widetilde{L}\sqrt{\frac{32\beta_M}{\theta_M} + \frac{4\beta_w \widetilde{L}^2}{\theta_w}\left(1 + \frac{16\beta_M}{\theta_M}\right)}\right)^{-1}$, $\widetilde{L} = \sqrt{\frac{1}{n}\sum_{i=1}^n L_i^2}$, $\theta_w \overset{def}{=} 1 - (1-\alpha_w)(1+ s)$, $\beta_w \overset{def}{=} (1-\alpha_w)\left(1 + s^{-1}\right)$, $\theta_M \overset{def}{=} 1 - (1 - \alpha_M)(1 + \rho)$, $\beta_M \overset{def}{=} (1 - \alpha_M)\left(1 + \rho^{-1}\right)$ *for any* $\rho, s > 0$.

*Let* $\Psi^t \overset{def}{=} f(x^t) - f^{\inf} + \frac{\gamma}{\theta_M}\left\|g^t - \widetilde{g}^t\right\|^2 + \frac{\gamma}{\theta_w}\left(1 + \frac{8\beta_M}{\theta_M}\right)P^t$. *Then for any* $T \geq 0$, *we have*

$$\mathbb{E}\left[\Psi^T\right] \leq (1 - \gamma\mu)^T \mathbb{E}\left[\Psi^0\right]. \tag{81}$$

*Proof.* Similarly to the proof of Theorem 9 the inequalities (74), (75), (76) hold with $\delta^t = f(x^t) - f(x^\star)$.

It remains to apply the steps similar to those in the proof of Theorem 6. $\square$

**Corollary 15.** *Let assumption of Theorem 10 hold,*

$$g^0 = \nabla f(x^0), \qquad \widetilde{g}_i^0 = \nabla f_i(x^0), \qquad i = 1, \ldots, n,$$

$$\gamma = \min\left\{\gamma_0, \frac{\theta_M}{2\mu}, \frac{\theta_w}{2\mu}\right\}, \qquad \gamma_0 = \left(L + \widetilde{L}\sqrt{\frac{32\beta_M}{\theta_M} + \frac{4\beta_w \widetilde{L}^2}{\theta_w}\left(1 + \frac{16\beta_M}{\theta_M}\right)}\right)^{-1},$$

*Then, after $T$ iterations of* EF21-PAGE *we have* $\mathbb{E}\left[f(x^T) - f^{\mathrm{inf}}\right] \le \varepsilon$. *It requires*

$$T = \#grad = \mathcal{O}\left(\frac{\widetilde{L}}{\mu\alpha_w\alpha_M}\ln\left(\frac{\delta^0}{\varepsilon}\right)\right)$$

*iterations/communications rounds/gradint computations at each node.*

# H   HEAVY BALL MOMENTUM

**Notations for this section:** $R^t = \|\gamma g^t\|^2 = (1-\eta)^2 \|z^{t+1} - z^t\|^2$.

In this section, we study the momentum version of EF21. In particular, we focus on Polyak style momentum (Polyak, 1964; Yang et al., 2016). Let $g^t$ be a gradient estimator at iteration $t$, then the update rule of *heavy ball* (HB) is given by

$$x^{t+1} = x^t - \gamma g^t + \eta \left( x^t - x^{t-1} \right),$$

where $x^{-1} = x^0$, $\eta \in [0, 1)$ is called the *momentum parameter*, and $\gamma > 0$ is the stepsize. The above update rule can be viewed as a combination of the classical *gradient step*

$$y^t = x^t - \gamma g^t$$

followed by additional *momentum step*

$$x^{t+1} = y^t + \eta \left( x^t - x^{t-1} \right).$$

Here the momentum term is added to accelerate the convergence and make the trajectory look like a smooth descent to the bottom of the ravine, rather than zigzag.

Equivalently, the update of HB can be implemented by the following two steps (Yang et al., 2016):

$$\begin{cases} x^{t+1} = x^t - \gamma v^t \\ v^{t+1} = \eta v^t + g^{t+1}. \end{cases}$$

We are now ready to present the distributed variant of heavy ball method enhanced with a contractive compressor $\mathcal{C}$, and EF21 mechanism, which we call EF21-HB (Algorithm 6). We present the complexity results in Theorem 11 and Corollary 16.

---

**Algorithm 6** EF21-HB

1: **Input:** starting point $x^0 \in \mathbb{R}^d$; $g_i^0 \in \mathbb{R}^d$ for $i = 1, \dots, n$ (known by nodes); $v^0 = g^0 = \frac{1}{n} \sum_{i=1}^n g_i^0$ (known by master); learning rate $\gamma > 0$; momentum parameter $0 \le \eta < 1$
2: **for** $t = 0, 1, 2, \dots, T - 1$ **do**
3:     Master computes $x^{t+1} = x^t - \gamma v^t$ and broadcasts $x^{t+1}$ to all nodes
4:     **for all nodes** $i = 1, \dots, n$ **in parallel do**
5:         Compress $c_i^t = \mathcal{C}(\nabla f_i(x^{t+1}) - g_i^t)$ and send $c_i^t$ to the master
6:         Update local state $g_i^{t+1} = g_i^t + c_i^t$
7:     **end for**
8:     Master computes $g^{t+1} = \frac{1}{n} \sum_{i=1}^n g_i^{t+1}$ via $g^{t+1} = g^t + \frac{1}{n} \sum_{i=1}^n c_i^t$, and $v^{t+1} = \eta v^t + g^{t+1}$
9: **end for**

---

In the analysis of EF21-HB, we assume by default that $v^{-1} = 0$.

**Lemma 9.** *Let sequences $\{x^t\}_{t \ge 0}$, and $\{v^t\}_{t \ge 0}$ be generated by Algorithm 6 and let the sequence $\{z^t\}_{t \ge 0}$ be defined as $z^{t+1} \stackrel{def}{=} x^{t+1} - \frac{\gamma\eta}{1-\eta} v^t$ with $0 \le \eta < 1$. Then for all $t \ge 0$*

$$z^{t+1} = z^t - \frac{\gamma}{1-\eta} g^t.$$

*Proof.*

$$
\begin{aligned}
z^{t+1} &\stackrel{(i)}{=} x^{t+1} - \frac{\gamma\eta}{1-\eta} v^t \\
&\stackrel{(ii)}{=} x^t - \gamma v^t - \frac{\gamma\eta}{1-\eta} v^t \\
&\stackrel{(iii)}{=} z^t + \frac{\gamma\eta}{1-\eta} v^{t-1} - \frac{\gamma}{1-\eta} v^t \\
&= z^t - \frac{\gamma}{1-\eta} \left( v^t - \eta v^{t-1} \right) \\
&= z^t - \frac{\gamma}{1-\eta} g^t,
\end{aligned}
$$

where in $(i)$ and $(iii)$ we use the definition of $z^{t+1}$ and $z^t$, in $(ii)$ we use the step $x^{t+1} = x^t - \gamma v^t$ (line 3 of Algorithm 6). Finally, the last equality follows by the update $v^{t+1} = \eta v^t + g^{t+1}$ (line 8 of Algorithm 6).

$\square$

**Lemma 10.** *Let the sequence $\{v^t\}_{t\geq 0}$ be defined as $v^{t+1} = \eta v^t + g^{t+1}$ with $0 \leq \eta < 1$. Then*

$$\sum_{t=0}^{T-1} \left\| v^t \right\|^2 \leq \frac{1}{(1-\eta)^2} \sum_{t=0}^{T-1} \left\| g^t \right\|^2 .$$

*Proof.* Unrolling the given recurrence and noticing that $v^{-1} = 0$, we have $v^t = \sum_{l=0}^{t} \eta^{t-l} g^l$. Define $H \stackrel{\text{def}}{=} \sum_{l=0}^{t} \eta^l \leq \frac{1}{1-\eta}$. Then by Jensen's inequality

$$
\begin{aligned}
\sum_{t=0}^{T-1} \left\| v^t \right\|^2 &= H^2 \sum_{t=0}^{T-1} \left\| \sum_{l=0}^{t} \frac{\eta^{t-l}}{H} g^l \right\|^2 \\
&\leq H^2 \sum_{t=0}^{T-1} \sum_{l=0}^{t} \frac{\eta^{t-l}}{H} \left\| g^l \right\|^2 \\
&= H \sum_{t=0}^{T-1} \sum_{l=0}^{t} \eta^{t-l} \left\| g^l \right\|^2 \\
&\leq \frac{1}{1-\eta} \sum_{t=0}^{T-1} \sum_{l=0}^{t} \eta^{t-l} \left\| g^l \right\|^2 \\
&= \frac{1}{1-\eta} \sum_{l=0}^{T-1} \left\| g^l \right\|^2 \sum_{t=l}^{T-1} \eta^{t-l} \\
&\leq \frac{1}{(1-\eta)^2} \sum_{t=0}^{T-1} \left\| g^t \right\|^2 .
\end{aligned}
$$

$\square$

**Lemma 11.** *Let the sequence $\{z^t\}_{t\geq 0}$ be defined as $z^{t+1} \stackrel{\text{def}}{=} x^{t+1} - \frac{\gamma\eta}{1-\eta} v^t$ with $0 \leq \eta < 1$. Then*

$$\sum_{t=0}^{T-1} \mathbb{E}\left[G^{t+1}\right] \leq (1-\theta) \sum_{t=0}^{T-1} \mathbb{E}\left[G^t\right] + 2\beta\widetilde{L}^2(1+4\eta^2) \sum_{t=0}^{T-1} \mathbb{E}\left[\left\| z^{t+1} - z^t \right\|^2\right],$$

*where $\theta = 1 - (1-\alpha)(1+s)$, $\quad \beta = (1-\alpha)\left(1+s^{-1}\right)$ for any $s > 0$.*

*Proof.* Summing up the inequality in Lemma 1 (for EF21 estimator) for $t = 0, \ldots, T-1$, we have

$$\sum_{t=0}^{T-1} \mathbb{E}\left[G^{t+1}\right] \leq (1-\theta) \sum_{t=0}^{T-1} \mathbb{E}\left[G^t\right] + \beta\widetilde{L}^2 \sum_{t=0}^{T-1} \mathbb{E}\left[\left\| x^{t+1} - x^t \right\|^2\right]. \tag{82}$$

It remains to bound $\sum_{t=0}^{T-1} \mathbb{E}\left[\left\| x^{t+1} - x^t \right\|^2\right]$. Notice that by definition of $\{z^t\}_{t\geq 0}$, we have

$$x^{t+1} - x^t = z^{t+1} - z^t + \frac{\gamma\eta}{1-\eta}\left(v^t - v^{t-1}\right).$$

Thus

$$
\begin{aligned}
\sum_{t=0}^{T-1} \mathbb{E}\left[\left\|x^{t+1}-x^t\right\|^2\right] &\leq 2\sum_{t=0}^{T-1} \mathbb{E}\left[\left\|z^{t+1}-z^t\right\|^2\right] + \frac{2\gamma^2\eta^2}{(1-\eta)^2}\sum_{t=0}^{T-1}\mathbb{E}\left[\left\|v^t-v^{t-1}\right\|^2\right] \\
&= 2\sum_{t=0}^{T-1} \mathbb{E}\left[\left\|z^{t+1}-z^t\right\|^2\right] + \frac{2\gamma^2\eta^2}{(1-\eta)^2}\sum_{t=0}^{T-1}\mathbb{E}\left[\left\|g^t-(1-\eta)v^{t-1}\right\|^2\right] \\
&\leq 2\sum_{t=0}^{T-1} \mathbb{E}\left[\left\|z^{t+1}-z^t\right\|^2\right] + \frac{4\gamma^2\eta^2}{(1-\eta)^2}\sum_{t=0}^{T-1}\mathbb{E}\left[\left\|g^t\right\|^2\right] \\
&\quad + \frac{4\gamma^2\eta^2}{(1-\eta)^2}\sum_{t=0}^{T-1}(1-\eta)^2\mathbb{E}\left[\left\|v^{t-1}\right\|^2\right] \\
&\overset{(i)}{\leq} 2\sum_{t=0}^{T-1} \mathbb{E}\left[\left\|z^{t+1}-z^t\right\|^2\right] + \frac{4\gamma^2\eta^2}{(1-\eta)^2}\sum_{t=0}^{T-1}\mathbb{E}\left[\left\|g^t\right\|^2\right] \\
&\quad + \frac{4\gamma^2\eta^2}{(1-\eta)^2}\sum_{t=0}^{T-1}\mathbb{E}\left[\left\|g^t\right\|^2\right] \\
&= 2\sum_{t=0}^{T-1} \mathbb{E}\left[\left\|z^{t+1}-z^t\right\|^2\right] + \frac{8\gamma^2\eta^2}{(1-\eta)^2}\sum_{t=0}^{T-1}\mathbb{E}\left[\left\|g^t\right\|^2\right] \\
&\overset{(ii)}{=} 2\sum_{t=0}^{T-1} \mathbb{E}\left[\left\|z^{t+1}-z^t\right\|^2\right] + 8\eta^2\sum_{t=0}^{T-1}\mathbb{E}\left[\left\|z^{t+1}-z^t\right\|^2\right] \\
&= 2(1+4\eta^2)\sum_{t=0}^{T-1}\mathbb{E}\left[\left\|z^{t+1}-z^t\right\|^2\right],
\end{aligned}
$$

where in $(i)$ we apply Lemma 10, and in $(ii)$ Lemma 9 is utilized.

It remains to plug in the above inequality into (82) $\qquad\square$

**Lemma 12.** *Let the sequence $\{z^t\}_{t\geq 0}$ be generated as in Lemma 9, i.e., $z^{t+1}=z^t-\frac{\gamma}{1-\eta}g^t$, then for all $t \geq 0$*

$$
\left\|\nabla f(x^t)\right\|^2 \leq 2G^t + \frac{2(1-\eta)^2}{\gamma^2}\left\|z^{t+1}-z^t\right\|^2
$$

*with $G^t = \frac{1}{n}\sum_{i=1}^n \left\|\nabla f_i(x^t) - g_i^t\right\|^2$.*

*Proof.* Notice that for $\gamma > 0$ we have $\nabla f(x^t) = \nabla f(x^t) - g^t - \frac{1-\eta}{\gamma}(z^{t+1}-z^t)$. Then

$$
\begin{aligned}
\left\|\nabla f(x^t)\right\|^2 &\leq 2\left\|\nabla f(x^t)-g^t\right\|^2 + 2\frac{(1-\eta)^2}{\gamma^2}\left\|z^{t+1}-z^t\right\|^2 \\
&\leq \frac{2}{n}\sum_{i=1}^n\left\|\nabla f_i(x^t)-g_i^t\right\|^2 + \frac{2(1-\eta)^2}{\gamma^2}\left\|z^{t+1}-z^t\right\|^2,
\end{aligned}
$$

where the inequalities hold due to (118) with $s = 1$, and (119). $\qquad\square$

## H.1 Convergence for General Non-Convex Functions

**Theorem 11.** *Let Assumption 1 hold, and let the stepsize in Algorithm 6 be set as*

$$
0 < \gamma < \left(\frac{(1+\eta)L}{2(1-\eta)^2} + \frac{\widetilde{L}}{1-\eta}\sqrt{\frac{2\beta}{\theta}\left(1+4\eta^2\right)}\right)^{-1} \overset{def}{=} \gamma_0, \tag{83}
$$

where $0 \leq \eta < 1$, $\theta = 1 - (1 - \alpha)(1 + s)$, $\beta = (1 - \alpha)\left(1 + s^{-1}\right)$, and $s > 0$.

Fix $T \geq 1$ and let $\hat{x}^T$ be chosen from the iterates $x^0, x^1, \ldots, x^{T-1}$ uniformly at random. Then

$$\sum_{t=0}^{T-1} \mathbb{E}\left[\left\|\nabla f(x^t)\right\|^2\right] \leq \frac{3\delta^0(1 - \eta)}{T\gamma\left(1 - \frac{\gamma}{\gamma_0}\right)} + \frac{\mathbb{E}\left[G^0\right]}{\theta T}\left(2 + \frac{1}{2\lambda_1}\frac{3(1 - \eta)}{\gamma\left(1 - \frac{\gamma}{\gamma_0}\right)}\right), \qquad (84)$$

where $\lambda_1 \overset{def}{=} \widetilde{L}\sqrt{\frac{2\beta}{\theta}\left(1 + 4\eta^2\right)}$. If the stepsize is set to $0 < \gamma \leq \gamma_0/2$, then

$$\sum_{t=0}^{T-1} \mathbb{E}\left[\left\|\nabla f(x^t)\right\|^2\right] \leq \frac{6\delta^0(1 - \eta)}{\gamma T} + \frac{\mathbb{E}\left[G^0\right]}{T\theta}\left(2 + \frac{3(1 - \eta)}{\gamma\widetilde{L}\sqrt{\frac{2\beta}{\theta}\left(1 + 4\eta^2\right)}}\right). \qquad (85)$$

*Proof.* Consider the sequence $z^{t+1} \overset{def}{=} x^{t+1} - \frac{\gamma\eta}{1-\eta}v^t$ with $0 \leq \eta < 1$. Then Lemma 9 states that $z^{t+1} = z^t - \frac{\gamma}{1-\eta}g^t$. By $L$-smoothness of $f(\cdot)$

$$
\begin{aligned}
f(z^{t+1}) - f(z^t) &\leq \langle\nabla f(z^t), z^{t+1} - z^t\rangle + \frac{L}{2}\left\|z^{t+1} - z^t\right\|^2 \\
&= \langle\nabla f(z^t) - g^t, z^{t+1} - z^t\rangle + \langle g^t, z^{t+1} - z^t\rangle + \frac{L}{2}\left\|z^{t+1} - z^t\right\|^2 \\
&\overset{(i)}{=} \langle\nabla f(z^t) - g^t, z^{t+1} - z^t\rangle - \frac{1 - \eta}{\gamma}\left\|z^{t+1} - z^t\right\|^2 + \frac{L}{2}\left\|z^{t+1} - z^t\right\|^2 \\
&= \langle\nabla f(z^t) - g^t, z^{t+1} - z^t\rangle - \left(\frac{1 - \eta}{\gamma} - \frac{L}{2}\right)\left\|z^{t+1} - z^t\right\|^2 \\
&= \langle\nabla f(x^t) - g^t, z^{t+1} - z^t\rangle + \langle\nabla f(z^t) - \nabla f(x^t), z^{t+1} - z^t\rangle \\
&\qquad - \left(\frac{1 - \eta}{\gamma} - \frac{L}{2}\right)\left\|z^{t+1} - z^t\right\|^2 \\
&\overset{(ii)}{\leq} \frac{1}{2\lambda_1}\left\|\nabla f(x^t) - g^t\right\|^2 + \frac{\lambda_1}{2}\left\|z^{t+1} - z^t\right\|^2 + \frac{1}{2\lambda_2}\left\|\nabla f(z^t) - \nabla f(x^t)\right\|^2 \\
&\qquad + \frac{\lambda_2}{2}\left\|z^{t+1} - z^t\right\|^2 - \left(\frac{1 - \eta}{\gamma} - \frac{L}{2}\right)\left\|z^{t+1} - z^t\right\|^2 \\
&= \frac{1}{2\lambda_1}\left\|\nabla f(x^t) - g^t\right\|^2 + \frac{1}{2\lambda_2}\left\|\nabla f(z^t) - \nabla f(x^t)\right\|^2 \\
&\qquad - \left(\frac{1 - \eta}{\gamma} - \frac{L}{2} - \frac{\lambda_1}{2} - \frac{\lambda_2}{2}\right)\left\|z^{t+1} - z^t\right\|^2 \\
&\overset{(iii)}{\leq} \frac{1}{2\lambda_1}\left\|\nabla f(x^t) - g^t\right\|^2 + \frac{L^2}{2\lambda_2}\left\|z^t - x^t\right\|^2 \\
&\qquad - \left(\frac{1 - \eta}{\gamma} - \frac{L}{2} - \frac{\lambda_1}{2} - \frac{\lambda_2}{2}\right)\left\|z^{t+1} - z^t\right\|^2 \\
&\overset{(iv)}{\leq} \frac{1}{2\lambda_1}\left\|\nabla f(x^t) - g^t\right\|^2 + \frac{\gamma^2\eta^2 L^2}{2\lambda_2(1 - \eta)^2}\left\|v^{t-1}\right\|^2 \\
&\qquad - \left(\frac{1 - \eta}{\gamma} - \frac{L}{2} - \frac{\lambda_1}{2} - \frac{\lambda_2}{2}\right)\left\|z^{t+1} - z^t\right\|^2,
\end{aligned}
$$

where in $(i)$ Lemma 9 is applied, in $(ii)$ the inequality (115) is applied twice for $\lambda_1, \lambda_2 > 0$, $(iii)$ holds due to Assumption 1, and $(iv)$ holds by definition of $z^t = x^t - \frac{\gamma\eta}{1-\eta}v^{t-1}$.

Summing up the above inequalities for $t = 0, \ldots, T-1$ (assuming $v^{-1} = 0$), we have

$$
\begin{aligned}
f(z^T) \quad \leq \quad & f(z^0) + \frac{1}{2\lambda_1} \sum_{t=0}^{T-1} \left\| \nabla f(x^t) - g^t \right\|^2 + \frac{\gamma^2 \eta^2 L^2}{2\lambda_2 (1-\eta)^2} \sum_{t=0}^{T-1} \left\| v^t \right\|^2 \\
& - \left( \frac{1-\eta}{\gamma} - \frac{L}{2} - \frac{\lambda_1}{2} - \frac{\lambda_2}{2} \right) \sum_{t=0}^{T-1} \left\| z^{t+1} - z^t \right\|^2 \\
\overset{(i)}{\leq} \quad & f(z^0) + \frac{1}{2\lambda_1} \sum_{t=0}^{T-1} \left\| \nabla f(x^t) - g^t \right\|^2 + \frac{\gamma^2 \eta^2 L^2}{2\lambda_2 (1-\eta)^4} \sum_{t=0}^{T-1} \left\| g^t \right\|^2 \\
& - \left( \frac{1-\eta}{\gamma} - \frac{L}{2} - \frac{\lambda_1}{2} - \frac{\lambda_2}{2} \right) \sum_{t=0}^{T-1} \left\| z^{t+1} - z^t \right\|^2 \\
\overset{(ii)}{=} \quad & f(z^0) + \frac{1}{2\lambda_1} \sum_{t=0}^{T-1} \left\| \nabla f(x^t) - g^t \right\|^2 + \frac{\gamma^2 \eta^2 L^2}{2\lambda_2 (1-\eta)^4} \sum_{t=0}^{T-1} \frac{(1-\eta)^2}{\gamma^2} \left\| z^{t+1} - z^t \right\|^2 \\
& - \left( \frac{1-\eta}{\gamma} - \frac{L}{2} - \frac{\lambda_1}{2} - \frac{\lambda_2}{2} \right) \sum_{t=0}^{T-1} \left\| z^{t+1} - z^t \right\|^2 \\
= \quad & f(z^0) + \frac{1}{2\lambda_1} \sum_{t=0}^{T-1} \left\| \nabla f(x^t) - g^t \right\|^2 \\
& - \left( \frac{1-\eta}{\gamma} - \frac{L}{2} - \frac{\lambda_1}{2} - \frac{\lambda_2}{2} - \frac{\eta^2 L^2}{2\lambda_2 (1-\eta)^2} \right) \sum_{t=0}^{T-1} \left\| z^{t+1} - z^t \right\|^2 \\
\overset{(iii)}{\leq} \quad & f(z^0) + \frac{1}{2\lambda_1} \sum_{t=0}^{T-1} G^t \\
& - \left( \frac{1-\eta}{\gamma} - \frac{L}{2} - \frac{\lambda_1}{2} - \frac{\lambda_2}{2} - \frac{\eta^2 L^2}{2\lambda_2 (1-\eta)^2} \right) \sum_{t=0}^{T-1} \left\| z^{t+1} - z^t \right\|^2 \\
= \quad & f(z^0) + \frac{1}{2\lambda_1} \sum_{t=0}^{T-1} G^t \\
& - \left( \frac{1-\eta}{\gamma} - \frac{L}{2} - \frac{\lambda_1}{2} - \frac{\eta L}{(1-\eta)} \right) \sum_{t=0}^{T-1} \left\| z^{t+1} - z^t \right\|^2 \\
= \quad & f(z^0) + \frac{1}{2\lambda_1} \sum_{t=0}^{T-1} G^t - \left( \frac{1-\eta}{\gamma} - \frac{L}{2} - \frac{\lambda_1}{2} - \frac{\eta L}{(1-\eta)} \right) \frac{1}{(1-\eta)^2} \sum_{t=0}^{T-1} R^t,
\end{aligned}
$$

where $(i)$ holds due to Lemma 10, in $(ii)$ Lemma 9 is applied, in $(iii)$ we apply $\left\| \nabla f(x^t) - g^t \right\|^2 \leq G^t$. Finally, in the last two steps we choose $\lambda_2 = \frac{\eta L}{1-\eta}$, and recall the definition $R^t = \left\| \gamma g^t \right\|^2 = (1-\eta)^2 \left\| z^{t+1} - z^t \right\|^2$.

Subtracting $f^{\inf}$ from both sides of the above inequality, taking expectation and using the notation $\delta^t = f(z^t) - f^{\inf}$, we get

$$
\mathbb{E}\left[ \delta^T \right] \quad \leq \quad \mathbb{E}\left[ \delta^0 \right] + \frac{1}{2\lambda_1} \sum_{t=0}^{T-1} \mathbb{E}\left[ G^t \right] - \left( \frac{1-\eta}{\gamma} - \frac{L}{2} - \frac{\lambda_1}{2} - \frac{\eta L}{(1-\eta)} \right) \frac{1}{(1-\eta)^2} \sum_{t=0}^{T-1} \mathbb{E}\left[ R^t \right].
\tag{86}
$$

By Lemma 11, we have

$$
\sum_{t=0}^{T-1} \mathbb{E}\left[ G^{t+1} \right] \leq (1-\theta) \sum_{t=0}^{T-1} \mathbb{E}\left[ G^t \right] + \frac{2\beta \widetilde{L}^2 (1 + 4\eta^2)}{(1-\eta)^2} \sum_{t=0}^{T-1} \mathbb{E}\left[ R^t \right].
\tag{87}
$$

Next, we are going to add (86) with a $\frac{1}{2\theta\lambda_1}$ multiple of (87). First, let us "forget", for a moment, about all the terms involving $R^t$ and denote their sum appearing on the right hand side by $\mathcal{R}$, then

$$
\begin{aligned}
\mathbb{E}\left[\delta^T\right] + \frac{1}{2\theta\lambda_1}\sum_{t=0}^{T-1}\mathbb{E}\left[G^{t+1}\right] &\leq \mathbb{E}\left[\delta^0\right] + \frac{1}{2\lambda_1}\sum_{t=0}^{T-1}\mathbb{E}\left[G^t\right] + (1-\theta)\frac{1}{2\lambda_1}\sum_{t=0}^{T-1}\mathbb{E}\left[G^t\right] + \mathcal{R} \\
&= \mathbb{E}\left[\delta^0\right] + \frac{1}{2\theta\lambda_1}\sum_{t=0}^{T-1}\mathbb{E}\left[G^t\right] + \mathcal{R}.
\end{aligned}
$$

Canceling out the same terms in both sides of the above inequality, we get

$$
\mathbb{E}\left[\delta^T\right] + \frac{1}{2\theta\lambda_1}\mathbb{E}\left[G^T\right] \leq \mathbb{E}\left[\delta^0\right] + \frac{1}{2\theta\lambda_1}\mathbb{E}\left[G^0\right] + \mathcal{R},
$$

where $\mathcal{R} \stackrel{\text{def}}{=} -\left(\frac{1-\eta}{\gamma} - \frac{L}{2}\left(1 + \frac{2\eta}{1-\eta}\right) - \frac{\lambda_1}{2} - \frac{\beta\widetilde{L}^2(1+4\eta^2)}{\theta\lambda_1}\right)\frac{1}{(1-\eta)^2}\sum_{t=0}^{T-1}\mathbb{E}\left[R^t\right]$.

Now choosing $\lambda_1 = \widetilde{L}\sqrt{\frac{2\beta}{\theta}(1+4\eta^2)}$ and using the definition of $\gamma_0$ given by (83), i.e., $\gamma_0 \stackrel{\text{def}}{=} \left(\frac{(1+\eta)L}{2(1-\eta)^2} + \frac{\widetilde{L}}{1-\eta}\sqrt{\frac{2\beta}{\theta}(1+4\eta^2)}\right)^{-1}$, we have

$$
\begin{aligned}
&\left(\frac{1-\eta}{\gamma} - \frac{L}{2}\left(1 + \frac{2\eta}{1-\eta}\right) - \frac{\lambda_1}{2} - \frac{\beta\widetilde{L}^2(1+4\eta^2)}{\theta\lambda_1}\right)\frac{1}{(1-\eta)^2} \\
&= \left(\frac{1-\eta}{\gamma} - \frac{L}{2}\left(1 + \frac{2\eta}{1-\eta}\right) - \widetilde{L}\sqrt{\frac{2\beta}{\theta}(1+4\eta^2)}\right)\frac{1}{(1-\eta)^2} \\
&= \left(\frac{1}{\gamma} - \frac{L}{2}\frac{1+\eta}{(1-\eta)^2} - \frac{\widetilde{L}}{1-\eta}\sqrt{\frac{2\beta}{\theta}(1+4\eta^2)}\right)\frac{1}{1-\eta} \\
&= \left(\frac{1}{\gamma} - \frac{1}{\gamma_0}\right)\frac{1}{1-\eta}.
\end{aligned}
$$

Then

$$
\begin{aligned}
0 \leq \mathbb{E}\left[\Phi^T\right] &\stackrel{\text{def}}{=} \mathbb{E}\left[\delta^T + \frac{1}{2\theta\lambda_1}G^T\right] \\
&\leq \mathbb{E}\left[\delta^0 + \frac{1}{2\theta\lambda_1}G^0\right] - \left(\frac{1}{\gamma} - \frac{1}{\gamma_0}\right)\frac{1}{1-\eta}\sum_{t=0}^{T-1}\mathbb{E}\left[R^t\right] \\
&= \mathbb{E}\left[\Phi^0\right] - \left(\frac{1}{\gamma} - \frac{1}{\gamma_0}\right)\frac{1}{1-\eta}\sum_{t=0}^{T-1}\mathbb{E}\left[R^t\right].
\end{aligned}
$$

After rearranging, we get

$$
\frac{1}{\gamma^2}\sum_{t=0}^{T-1}\mathbb{E}\left[R^t\right] \leq \frac{\mathbb{E}\left[\Phi^0\right](1-\eta)}{\gamma\left(1 - \frac{\gamma}{\gamma_0}\right)}.
$$

Summing the result of Lemma 12 over $t = 0, \ldots, T-1$ and applying expectation, we get

$$
\sum_{t=0}^{T-1}\mathbb{E}\left[\left\|\nabla f(x^t)\right\|^2\right] \leq 2\sum_{t=0}^{T-1}\mathbb{E}\left[G^t\right] + \frac{2}{\gamma^2}\sum_{t=0}^{T-1}\mathbb{E}\left[R^t\right].
$$

Due to Lemma 11, the conditions of Lemma 18 hold with $C \stackrel{\text{def}}{=} 2\beta\widetilde{L}^2\frac{1+4\eta^2}{(1-\eta)^2}$, $s^t = \mathbb{E}\left[G^t\right]$, $r^t = \mathbb{E}\left[R^t\right]$, thus

$$\sum_{t=0}^{T-1} \mathbb{E}\left[G^t\right] \leq \frac{\mathbb{E}\left[G^0\right]}{\theta} + \frac{C}{\theta}\sum_{t=0}^{T-1}\mathbb{E}\left[R^t\right].$$

Combining the above inequalities, we can continue with

$$
\begin{aligned}
\sum_{t=0}^{T-1}\mathbb{E}\left[\left\|\nabla f(x^t)\right\|^2\right] &\leq 2\sum_{t=0}^{T-1}\mathbb{E}\left[G^t\right] + \frac{2}{\gamma^2}\sum_{t=0}^{T-1}\mathbb{E}\left[R^t\right] \\
&\leq \frac{2\mathbb{E}\left[G^0\right]}{\theta} + \left(2 + \frac{\gamma^2 C}{\theta}\right)\frac{1}{\gamma^2}\sum_{t=0}^{T-1}\mathbb{E}\left[R^t\right] \\
&\leq \frac{2\mathbb{E}\left[G^0\right]}{\theta} + \left(2 + \frac{\gamma^2 C}{\theta}\right)\frac{\mathbb{E}\left[\Phi^0\right](1-\eta)}{\gamma\left(1-\frac{\gamma}{\gamma_0}\right)}.
\end{aligned}
$$

Note that for $\gamma < \gamma_0 = \left(\frac{(1+\eta)L}{2(1-\eta)^2} + \sqrt{\frac{C}{\theta}}\right)^{-1}$, we have

$$\frac{\gamma^2 C}{\theta} \quad < \quad \frac{\frac{C}{\theta}}{\left(\frac{(1+\eta)L}{2(1-\eta)^2} + \sqrt{\frac{C}{\theta}}\right)^2} \leq 1. \tag{88}$$

Thus

$$
\begin{aligned}
\sum_{t=0}^{T-1}\mathbb{E}\left[\left\|\nabla f(x^t)\right\|^2\right] &\leq \frac{2\mathbb{E}\left[G^0\right]}{\theta} + \frac{3\mathbb{E}\left[\Phi^0\right](1-\eta)}{\gamma\left(1-\frac{\gamma}{\gamma_0}\right)} \\
&= \frac{3\delta^0(1-\eta)}{\gamma\left(1-\frac{\gamma}{\gamma_0}\right)} + \frac{\mathbb{E}\left[G^0\right]}{\theta}\left(2 + \frac{1}{2\lambda_1}\frac{3(1-\eta)}{\gamma\left(1-\frac{\gamma}{\gamma_0}\right)}\right),
\end{aligned}
$$

where $\lambda_1 = \widetilde{L}\sqrt{\frac{2\beta}{\theta}(1+4\eta^2)}$. $\qquad\square$

**Corollary 16.** *Let assumptions of Theorem 11 hold,*

$$
\begin{aligned}
g_i^0 &= \nabla f_i(x^0), \qquad i = 1,\ldots,n, \\
\gamma &= \left(\frac{(1+\eta)L}{2(1-\eta)^2} + \frac{\widetilde{L}}{1-\eta}\sqrt{\frac{2\beta}{\theta}(1+4\eta^2)}\right)^{-1}.
\end{aligned}
$$

*Then, after $T$ iterations/communication rounds of* EF21-HB *we have* $\mathbb{E}\left[\left\|\nabla f(\hat{x}^T)\right\|^2\right] \leq \varepsilon^2$. *It requires*

$$T = \#grad = \mathcal{O}\left(\frac{\widetilde{L}\delta^0}{\varepsilon^2}\left(\frac{1}{\alpha} + \frac{1}{1-\eta}\right)\right) \tag{89}$$

*iterations/communications rounds/gradint computations at each node.*

*Proof.* Notice that by using $L \leq \widetilde{L}$, $\eta < 1$ and Lemma 17, we have

$$
\begin{aligned}
\frac{(1+\eta)L}{2(1-\eta)^2} + \frac{\widetilde{L}}{1-\eta}\sqrt{\frac{2\beta}{\theta}(1+4\eta^2)} &\leq \frac{\widetilde{L}}{(1-\eta)^2} + \frac{\widetilde{L}}{1-\eta}\sqrt{\frac{10\beta}{\theta}} \\
&\leq \frac{\widetilde{L}}{1-\eta}\left(\frac{1}{1-\eta} + \frac{2\sqrt{10}}{\alpha}\right).
\end{aligned}
$$

Using the above inequality, (85), and (83), we get

$$
\begin{aligned}
\#\text{grad} \quad &= \quad T \leq \frac{6\delta^0(1-\eta)}{\gamma\varepsilon^2} \leq \frac{6\delta^0(1-\eta)}{\varepsilon^2} \frac{\widetilde{L}}{1-\eta} \left( \frac{1}{1-\eta} + \frac{2\sqrt{10}}{\alpha} \right) \\
&\leq \quad \frac{6\widetilde{L}\delta^0}{\varepsilon^2} \left( \frac{1}{1-\eta} + \frac{2\sqrt{10}}{\alpha} \right).
\end{aligned}
$$

$\square$

## I  COMPOSITE CASE

Now we focus on solving a composite optimization problem

$$\min_{x \in \mathbb{R}^d} \Phi(x) \overset{\text{def}}{=} \frac{1}{n} \sum_{i=1}^{n} f_i(x) + r(x), \tag{90}$$

where each $f_i(\cdot)$ is $L_i$-smooth (possibly non-convex), $r(\cdot)$ is convex, and $\Phi^{\text{inf}} = \inf_{x \in \mathbb{R}^d} \Phi(x) > -\infty$. This is a standard and important generalization of setting (1). Namely, it includes three special cases.

- **Smooth unconstrained optimization.** Set $r \equiv 0$, then we recover the initially stated problem formulation (1).

- **Smooth optimization over convex set.** Let $r = \delta_Q$ (indicator function of the set $Q$), where $Q$ is a nonempty closed convex set. Then (90) reduces to the problem of minimizing finite a sum of smooth (possibly non-convex) functions over a nonempty closed convex set

$$\min_{x \in Q} \left\{ \frac{1}{n} \sum_{i=1}^{n} f_i(x) \right\}.$$

- $l_1$-**regularized optimization.** Choose $r(x) = \lambda \|x\|_1$ with $\lambda > 0$, then (90) amounts to the $l_1$-regularized (also known as LASSO) problem

$$\min_{x \in \mathbb{R}^d} \left\{ \frac{1}{n} \sum_{i=1}^{n} f_i(x) + \lambda \|x\|_1 \right\}.$$

For any $\gamma > 0$, $x \in \mathbb{R}^d$, define a proximal mapping of function $r(\cdot)$ (prox-operator) as

$$\text{prox}_{\gamma r}(x) \overset{\text{def}}{=} \arg\min_{y \in \mathbb{R}^d} \left\{ r(y) + \frac{1}{2\gamma} \|y - x\|^2 \right\}. \tag{91}$$

Throughout this section, we assume that the master node can efficiently compute prox-operator at every iteration. This is a reasonable assumption, and in many cases (choices of $r(\cdot)$) appearing in applications, there exists an analytical solution of (91), or its computation is cheap compared to the aggregation step.

To evaluate convergence in composite case, we define the *generalized gradient mapping* at a point $x \in \mathbb{R}^d$ with a parameter $\gamma$

$$\mathcal{G}_\gamma(x) \overset{\text{def}}{=} \frac{1}{\gamma} \left( x - \text{prox}_{\gamma r}(x - \gamma \nabla f(x)) \right). \tag{92}$$

One can verify that the above quantity is a well-defined evaluation metric (Beck, 2017). Namely, for any $x^* \in \mathbb{R}^d$, it holds that $\mathcal{G}_\gamma(x) = 0$ if and only if $x^*$ is a stationary point of (90), and in a special case when $r \equiv 0$, we have $\mathcal{G}_\gamma(x) = \nabla f(x)$.

**Notations for this section:** in this section we re-define $\delta^t \overset{\text{def}}{=} \Phi(x^t) - \Phi^{\text{inf}}$

**Lemma 13** (Gradient mapping bound). *Let* $x^{t+1} \overset{def}{=} \text{prox}_{\gamma r}(x^t - \gamma v^t)$, *then*

$$\mathbb{E}\left[ \left\| \mathcal{G}_\gamma(x^t) \right\|^2 \right] \le \frac{2}{\gamma^2} \mathbb{E}\left[ \left\| x^{t+1} - x^t \right\|^2 \right] + 2\mathbb{E}\left[ \left\| v^t - \nabla f(x^t) \right\|^2 \right]. \tag{93}$$

*Proof.*

$$
\begin{aligned}
\mathbb{E}\left[\left\|\mathcal{G}_\gamma\left(x^t\right)\right\|^2\right] &= \frac{1}{\gamma^2}\mathbb{E}\left[\left\|x^t - \mathrm{prox}_{\gamma r}(x^t - \gamma\nabla f(x^t))\right\|^2\right] \\
&\leq \frac{2}{\gamma^2}\mathbb{E}\left[\left\|x^{t+1} - x^t\right\|^2\right] + \frac{2}{\gamma^2}\mathbb{E}\left[\left\|x^{t+1} - \mathrm{prox}_{\gamma r}(x^t - \gamma\nabla f(x^t))\right\|^2\right] \\
&= \frac{2}{\gamma^2}\mathbb{E}\left[\left\|x^{t+1} - x^t\right\|^2\right] \\
&\quad + \frac{2}{\gamma^2}\mathbb{E}\left[\left\|\mathrm{prox}_{\gamma r}(x^t - \gamma v^t) - \mathrm{prox}_{\gamma r}(x^t - \gamma\nabla f(x^t))\right\|^2\right] \\
&\leq \frac{2}{\gamma^2}\mathbb{E}\left[\left\|x^{t+1} - x^t\right\|^2\right] \\
&\quad + \frac{2}{\gamma^2}\mathbb{E}\left[\left\|(x^t - \gamma v^t) - (x^t - \gamma\nabla f(x^t))\right\|^2\right] \\
&= \frac{2}{\gamma^2}\mathbb{E}\left[\left\|x^{t+1} - x^t\right\|^2\right] + 2\mathbb{E}\left[\left\|v^t - \nabla f(x^t)\right\|^2\right], \tag{94}
\end{aligned}
$$

where in the last inequality we apply non-expansiveness of prox-operator. $\qquad\square$

**Lemma 14.** *Let* $x^{t+1} \overset{def}{=} \mathrm{prox}_{\gamma r}(x^t - \gamma v^t)$, *then for any* $\lambda > 0$,

$$
\Phi\left(x^{t+1}\right) \leq \Phi\left(x^t\right) + \frac{1}{2\lambda}\left\|v^t - \nabla f(x^t)\right\|^2 - \left(\frac{1}{\gamma} - \frac{L}{2} - \frac{\lambda}{2}\right)\left\|x^{t+1} - x^t\right\|^2. \tag{95}
$$

*Proof.* Define $\tilde{r}(x) \overset{def}{=} r(x) + \frac{1}{2\gamma}\left\|x - x^t + \gamma v^t\right\|^2$, and note that $x^{t+1} = \arg\min_{x\in\mathbb{R}^d}\{\tilde{r}(x)\}$. Since $\tilde{r}(\cdot)$ is $1/\gamma$ - strongly convex, we have

$$
\tilde{r}(x^t) \geq \tilde{r}(x^{t+1}) + \frac{1}{2\gamma}\left\|x^{t+1} - x^t\right\|^2,
$$

$$
r(x^t) + \frac{1}{2\gamma}\left\|\gamma v^t\right\|^2 \geq r(x^{t+1}) + \frac{1}{2\gamma}\left\|x^{t+1} - x^t + \gamma v^t\right\|^2 + \frac{1}{2\gamma}\left\|x^{t+1} - x^t\right\|^2.
$$

Thus

$$
r(x^{t+1}) - r(x^t) \leq -\frac{1}{\gamma}\left\|x^{t+1} - x^t\right\|^2 - \langle v^t, x^{t+1} - x^t\rangle. \tag{96}
$$

By $L$ smoothness of $f(\cdot)$,

$$
f\left(x^{t+1}\right) - f\left(x^t\right) \leq \left\langle\nabla f\left(x^t\right), x^{t+1} - x^t\right\rangle + \frac{L}{2}\left\|x^{t+1} - x^t\right\|^2. \tag{97}
$$

Summing up (97) with (96) we obtain

$$
\begin{aligned}
\Phi\left(x^{t+1}\right) - \Phi\left(x^t\right) &\leq \langle\nabla f(x^t) - v^t, x^{t+1} - x^t\rangle - \left(\frac{1}{\gamma} - \frac{L}{2}\right)\left\|x^{t+1} - x^t\right\|^2 \\
&\leq \frac{1}{2\lambda}\left\|\nabla f(x^t) - v^t\right\|^2 - \left(\frac{1}{\gamma} - \frac{L}{2} - \frac{\lambda}{2}\right)\left\|x^{t+1} - x^t\right\|^2.
\end{aligned}
$$

$\qquad\square$

We are now ready to present EF21-Prox and provide its convergence guarantees in general non-convex case.

## I.1   CONVERGENCE FOR GENERAL NON-CONVEX FUNCTIONS

---

**Algorithm 7** EF21-Prox

---

1: **Input:** starting point $x^0 \in \mathbb{R}^d$; $g_i^0 \in \mathbb{R}^d$ for $i = 1, \ldots, n$ (known by nodes); $g^0 = \frac{1}{n} \sum_{i=1}^{n} g_i^0$ (known by master); learning rate $\gamma > 0$
2: **for** $t = 0, 1, 2, \ldots, T - 1$ **do**
3:     Master computes $x^{t+1} = \text{prox}_{\gamma r} (x^t - \gamma g^t)$
4:     **for all nodes** $i = 1, \ldots, n$ **in parallel do**
5:         Compress $c_i^t = \mathcal{C}(\nabla f_i(x^{t+1}) - g_i^t)$ and send $c_i^t$ to the master
6:         Update local state $g_i^{t+1} = g_i^t + c_i^t$
7:     **end for**
8:     Master computes $g^{t+1} = \frac{1}{n} \sum_{i=1}^{n} g_i^{t+1}$ via $g^{t+1} = g^t + \frac{1}{n} \sum_{i=1}^{n} c_i^t$
9: **end for**
10: **Output:** $\hat{x}_T$ chosen uniformly from $\{x^t\}_{t \in [T]}$

---

**Theorem 12.** *Let Assumption 1 hold, $r(\cdot)$ be convex and $\Phi^{\text{inf}} = \inf_{x \in \mathbb{R}^d} \Phi(x) > -\infty$. Set the stepsize in Algorithm 7 as*

$$0 < \gamma < \left( \frac{L}{2} + \widetilde{L} \sqrt{\frac{\beta}{\theta}} \right)^{-1} \stackrel{def}{=} \gamma_0, \tag{98}$$

*where $\widetilde{L} = \sqrt{\frac{1}{n} \sum_{i=1}^{n} L_i^2}$, $\theta = 1 - (1 - \alpha)(1 + s)$, $\beta = (1 - \alpha)\left(1 + s^{-1}\right)$ for any $s > 0$.*

*Fix $T \geq 1$ and let $\hat{x}^T$ be chosen from the iterates $x^0, x^1, \ldots, x^{T-1}$ uniformly at random. Then*

$$\mathbb{E}\left[ \left\| \mathcal{G}_\gamma(\hat{x}^T) \right\|^2 \right] \leq \frac{4\left(\Phi^0 - \Phi^{\text{inf}}\right)}{T\gamma\left(1 - \frac{\gamma}{\gamma_0}\right)} + \frac{2\mathbb{E}\left[G^0\right]}{\theta T} \left( 1 + \frac{1}{\gamma\left(1 - \frac{\gamma}{\gamma_0}\right)} \frac{1}{\widetilde{L}} \sqrt{\frac{\theta}{\beta}} \right). \tag{99}$$

*If the stepsize is set to $0 < \gamma \leq \gamma_0/2$, then*

$$\mathbb{E}\left[ \left\| \mathcal{G}_\gamma(\hat{x}^T) \right\|^2 \right] \leq \frac{8\left(\Phi^0 - \Phi^{\text{inf}}\right)}{\gamma T} + \frac{2\mathbb{E}\left[G^0\right]}{\theta T} \left( 1 + \frac{2}{\gamma\widetilde{L}} \sqrt{\frac{\theta}{\beta}} \right). \tag{100}$$

*Proof.* First, let us apply Lemma 14 with $v^t = g^t$, $\lambda > 0$

$$\Phi\left(x^{t+1}\right) \leq \Phi\left(x^t\right) + \frac{1}{2\lambda}\left\| g^t - \nabla f\left(x^t\right) \right\|^2 - \left( \frac{1}{\gamma} - \frac{L}{2} - \frac{\lambda}{2} \right)\left\| x^{t+1} - x^t \right\|^2. \tag{101}$$

Subtract $\Phi^{\text{inf}}$ from both sides, take expectation, and define $\delta^t = \Phi(x^t) - \Phi^{\text{inf}}$, $G^t = \frac{1}{n} \sum_{i=1}^{n} \left\| g_i^t - \nabla f_i(x^t) \right\|^2$, $R^t = \left\| x^{t+1} - x^t \right\|^2$, then

$$\mathbb{E}\left[\delta^{t+1}\right] \leq \mathbb{E}\left[\delta^t\right] - \left( \frac{1}{\gamma} - \frac{L}{2} - \frac{\lambda}{2} \right)\mathbb{E}\left[R^t\right] + \frac{1}{2\lambda}\mathbb{E}\left[G^t\right]. \tag{102}$$

Note that the proof of Lemma 1 does not rely on the update rule for $x^{t+1}$, but only on the way the estimator $g_i^{t+1}$ is constructed. Therefore, (14) also holds for the composite case

$$\mathbb{E}\left[G^{t+1}\right] \leq (1 - \theta)\mathbb{E}\left[G^t\right] + \beta\widetilde{L}^2\mathbb{E}\left[R^t\right]. \tag{103}$$

Adding (102) with a $\frac{1}{2\theta\lambda}$ multiple of (103), we obtain

$$\mathbb{E}\left[\delta^{t+1}\right] + \frac{1}{2\theta\lambda}\mathbb{E}\left[G^{t+1}\right] \leq \mathbb{E}\left[\delta^t\right] + \frac{1}{2\lambda}\mathbb{E}\left[G^t\right] + \frac{1 - \theta}{2\theta\lambda}\mathbb{E}\left[G^t\right] - \left( \frac{1}{\gamma} - \frac{L}{2} - \frac{\lambda}{2} \right)\mathbb{E}\left[R^t\right]$$

$$+ \frac{1}{2\theta\lambda}\beta\widetilde{L}^2\mathbb{E}\left[R^t\right]$$

$$= \mathbb{E}\left[\delta^t\right] + \frac{1}{2\theta\lambda}\mathbb{E}\left[G^t\right] - \left( \frac{1}{\gamma} - \frac{L}{2} - \frac{\lambda}{2} - \frac{\beta}{2\theta\lambda}\widetilde{L}^2 \right)\mathbb{E}\left[R^t\right].$$

By summing up inequalities for $t = 0, \ldots, T - 1$, we arrive at

$$0 \leq \mathbb{E}\left[\delta^T\right] + \frac{1}{2\theta\lambda}\mathbb{E}\left[G^T\right] \leq \delta^0 + \frac{1}{2\theta\lambda}\mathbb{E}\left[G^0\right] - \left(\frac{1}{\gamma} - \frac{L}{2} - \frac{\lambda}{2} - \frac{\beta}{2\theta\lambda}\widetilde{L}^2\right)\sum_{t=0}^{T-1}\mathbb{E}\left[R^t\right].$$

Thus

$$
\begin{aligned}
\sum_{t=0}^{T-1}\mathbb{E}\left[R^t\right] &\leq \left(\delta^0 + \frac{1}{2\theta\lambda}\mathbb{E}\left[G^0\right]\right)\left(\frac{1}{\gamma} - \frac{L}{2} - \frac{\lambda}{2} - \frac{\beta}{2\theta\lambda}\widetilde{L}^2\right)^{-1} \\
&= \left(\delta^0 + \frac{1}{2\theta}\sqrt{\frac{\theta}{\beta\widetilde{L}^2}}\mathbb{E}\left[G^0\right]\right)\left(\frac{1}{\gamma} - \frac{L}{2} - \sqrt{\frac{\beta}{\theta}\widetilde{L}^2}\right)^{-1} \\
&= \gamma^2 F^0 B.
\end{aligned}
$$
(104)

where in the first equality we choose $\lambda = \sqrt{\frac{\beta}{\theta}\widetilde{L}^2}$, and in the second we define $F^0 \overset{\text{def}}{=} \delta^0 + \frac{1}{2\theta}\sqrt{\frac{\theta}{\beta\widetilde{L}^2}}\mathbb{E}\left[G^0\right]$, $B \overset{\text{def}}{=} \left(\gamma - \frac{L\gamma^2}{2} - \sqrt{\frac{\beta}{\theta}\widetilde{L}^2}\gamma^2\right)^{-1} = \left(\gamma - \frac{\gamma^2}{\gamma_0}\right)^{-1}$.

By Lemma 13 with $v^t = g^t$ we have

$$
\begin{aligned}
\mathbb{E}\left[\left\|\mathcal{G}_\gamma\left(\hat{x}^T\right)\right\|^2\right] &= \frac{1}{T}\sum_{t=0}^{T-1}\mathbb{E}\left[\left\|\mathcal{G}_\gamma\left(x^t\right)\right\|^2\right] \\
&\leq \frac{2}{\gamma^2 T}\sum_{t=0}^{T-1}\mathbb{E}\left[R^t\right] + \frac{2}{T}\sum_{t=0}^{T-1}\mathbb{E}\left[G^t\right] \\
&\overset{(i)}{\leq} \frac{2}{\gamma^2 T}\sum_{t=0}^{T-1}\mathbb{E}\left[R^t\right] + \frac{2}{T}\frac{\mathbb{E}\left[G^0\right]}{\theta} + \frac{2}{T}\frac{\beta\widetilde{L}^2}{\theta}\sum_{t=0}^{T-1}\mathbb{E}\left[R^t\right] \\
&\overset{(ii)}{\leq} \frac{2F^0 B}{T} + \frac{2}{T}\frac{\mathbb{E}\left[G^0\right]}{\theta} + \frac{2}{T}\frac{\beta\widetilde{L}^2}{\theta}\gamma^2 F^0 B \\
&= \frac{2F^0 B}{T}\left(1 + \frac{\gamma^2\beta\widetilde{L}^2}{\theta}\right) + \frac{2}{T}\frac{\mathbb{E}\left[G^0\right]}{\theta} \\
&= \frac{2F^0}{T\gamma\left(1 - \frac{\gamma}{\gamma_0}\right)}\left(1 + \frac{\gamma^2\beta\widetilde{L}^2}{\theta}\right) + \frac{2}{T}\frac{\mathbb{E}\left[G^0\right]}{\theta},
\end{aligned}
$$

where in $(i)$ we apply Lemma 18 with $C \overset{\text{def}}{=} \beta\widetilde{L}^2$, $s^t \overset{\text{def}}{=} \mathbb{E}\left[G^t\right]$, $r^t \overset{\text{def}}{=} \mathbb{E}\left[R^t\right]$. $(ii)$ is due to (104).

Note that for $\gamma < \left(\frac{L}{2} + \sqrt{\frac{\beta}{\theta}}\widetilde{L}\right)^{-1}$, we have

$$\frac{\gamma^2\beta\widetilde{L}^2}{\theta} < \frac{\frac{\beta}{\theta}\widetilde{L}^2}{\left(\frac{L}{2} + \sqrt{\frac{\beta}{\theta}}\widetilde{L}\right)^2} \leq 1.$$
(105)

Thus

$$
\begin{aligned}
\mathbb{E}\left[\left\|\mathcal{G}_\gamma(\hat{x}^T)\right\|^2\right] &\leq \frac{4F^0}{T\gamma\left(1 - \frac{\gamma}{\gamma_0}\right)} + \frac{2}{T}\frac{\mathbb{E}\left[G^0\right]}{\theta} \\
&= \frac{4\delta^0}{T\gamma\left(1 - \frac{\gamma}{\gamma_0}\right)} + \frac{2\mathbb{E}\left[G^0\right]}{\theta T} + \frac{2\mathbb{E}\left[G^0\right]}{T\gamma\left(1 - \frac{\gamma}{\gamma_0}\right)}\frac{1}{\theta}\sqrt{\frac{\theta}{\beta\widetilde{L}^2}}.
\end{aligned}
$$
(106)

Set $\gamma \leq \gamma_0/2$, then the bound simplifies to

$$\mathbb{E}\left[\left\|\mathcal{G}_\gamma(\hat{x}^T)\right\|^2\right] \leq \frac{8\delta^0}{\gamma T} + \frac{2\mathbb{E}\left[G^0\right]}{\theta T}\left(1 + \frac{2}{\gamma}\sqrt{\frac{\theta}{\beta\widetilde{L}^2}}\right).$$

□

**Corollary 17.** *Let assumptions of Theorem 12 hold,*

$$g_i^0 = \nabla f_i(x^0), \qquad i = 1, \ldots, n,$$
$$\gamma = \left(L + 2\widetilde{L}\sqrt{\beta/\theta}\right)^{-1}.$$

*Then, after $T$ iterations/communication rounds of* EF21-Prox *we have* $\mathbb{E}\left[\left\|\nabla f(\hat{x}^T)\right\|^2\right] \leq \varepsilon^2$. *It requires*

$$\#grad = \mathcal{O}\left(\frac{\widetilde{L}\delta^0}{\alpha\varepsilon^2}\right),$$

*where $\widetilde{L} = \sqrt{\frac{1}{n}\sum_{i=1}^n L_i^2}$, $\delta^0 = \Phi(x^0) - \Phi^{inf}$.*

*Proof.* The proof is the same as for Corollary 2. □

### I.2 CONVERGENCE UNDER POLYAK-ŁOJASIEWICZ CONDITION

In order to extend the analysis of Polyak-Łojasiewicz functions to composite optimization, we use the following Assumption 5 from (Li & Li, 2018; Wang et al., 2018).

**Assumption 5** (Polyak-Łojasiewicz). *There exists $\mu > 0$ such that*

$$\|\mathcal{G}_\gamma(x)\|^2 \geq 2\mu\left(\Phi(x) - \Phi(x^\star)\right)$$

*for all $x \in \mathbb{R}^d$, where $x^\star = \arg\min_x \Phi(x)$.*

**Theorem 13.** *Let Assumptions 1 and 5 hold, $r(\cdot)$ be convex and $\Phi^{\inf} = \inf_{x \in \mathbb{R}^d} \Phi(x) > -\infty$. Set the stepsize in Algorithm 7 as*

$$\gamma \leq \min\left\{\left(L + 2\widetilde{L}\sqrt{\frac{2\beta}{\theta}}\right)^{-1}, \frac{\theta}{\mu + \theta\widetilde{L}\sqrt{\frac{2\beta}{\theta}}}\right\}. \tag{107}$$

*Let $\Psi^t \stackrel{def}{=} \Phi(x^t) - \Phi(x^\star) + \frac{1}{\theta\lambda}G^t$ with $\lambda = \sqrt{\frac{2\beta}{\theta}}\widetilde{L}$. Then for any $T \geq 0$, we have*

$$\mathbb{E}\left[\Psi^T\right] \leq \left(1 - \frac{\gamma\mu}{2}\right)^T \mathbb{E}\left[\Psi^0\right], \tag{108}$$

*where $\widetilde{L} = \sqrt{\frac{1}{n}\sum_{i=1}^n L_i^2}$, $\theta = 1 - (1-\alpha)(1+s)$, $\beta = (1-\alpha)\left(1 + s^{-1}\right)$ for any $s > 0$.*

*Proof.* We start as in the previous proof, but subtract $\Phi(x^\star)$ from both sides of (101) and define $\delta^t \stackrel{def}{=} \Phi(x^t) - \Phi(x^\star)$. Recall that $G^t = \frac{1}{n}\sum_{i=1}^n \|g_i^t - \nabla f_i(x^t)\|^2$, $R^t = \|x^{t+1} - x^t\|^2$. Then

$$\mathbb{E}\left[\delta^{t+1}\right] \leq \mathbb{E}\left[\delta^t\right] - \left(\frac{1}{\gamma} - \frac{L}{2} - \frac{\lambda}{2}\right)\mathbb{E}\left[R^t\right] + \frac{1}{2\lambda}\mathbb{E}\left[G^t\right]. \tag{109}$$

By Lemma 1, we have

$$\mathbb{E}\left[G^{t+1}\right] \leq (1-\theta)\mathbb{E}\left[G^t\right] + \beta\widetilde{L}^2\mathbb{E}\left[R^t\right]. \tag{110}$$

Then by adding (109) with a $\frac{1}{\theta\lambda}$ multiple of (110) we obtain

$$
\begin{aligned}
\mathbb{E}\left[\delta^{t+1}\right] + \frac{1}{\theta\lambda}\mathbb{E}\left[G^{t+1}\right] \;\leq\;& \mathbb{E}\left[\delta^{t}\right] + \frac{1}{\theta\lambda}\left(1-\theta+\frac{\theta}{2}\right)\mathbb{E}\left[G^{t}\right] - \left(\frac{1}{\gamma}-\frac{L}{2}-\frac{\lambda}{2}\right)\mathbb{E}\left[R^{t}\right] \\
&+ \frac{1}{\theta\lambda}\beta\widetilde{L}^{2}\mathbb{E}\left[R^{t}\right] \\
=\;& \mathbb{E}\left[\delta^{t}\right] + \frac{1}{\theta\lambda}\left(1-\frac{\theta}{2}\right)\mathbb{E}\left[G^{t}\right] - \left(\frac{1}{\gamma}-\frac{L}{2}-\frac{\lambda}{2}-\frac{\beta}{\theta\lambda}\widetilde{L}^{2}\right)\mathbb{E}\left[R^{t}\right] \\
\overset{(i)}{=}\;& \mathbb{E}\left[\delta^{t}\right] + \frac{1}{\theta\lambda}\left(1-\frac{\theta}{2}\right)\mathbb{E}\left[G^{t}\right] - \left(\frac{1}{\gamma}-\frac{L}{2}-\sqrt{\frac{2\beta}{\theta}}\widetilde{L}\right)\mathbb{E}\left[R^{t}\right] \\
\overset{(ii)}{\leq}\;& \mathbb{E}\left[\delta^{t}\right] + \frac{1}{\theta\lambda}\left(1-\frac{\theta}{2}\right)\mathbb{E}\left[G^{t}\right] - \frac{1}{2\gamma}\mathbb{E}\left[R^{t}\right], \qquad (111)
\end{aligned}
$$

where in $(i)$ we choose $\lambda = \sqrt{\frac{2\beta}{\theta}\widetilde{L}^{2}}$, $(ii)$ is due to the stepsize choice (the first term in minimum). Next, combining Assumption 5 with Lemma 13, we have

$$
2\mu\delta^{t} = 2\mu\left(\Phi(x^{t})-\Phi(x^{\star})\right) \leq \left\|\mathcal{G}_{\gamma}(x^{t})\right\|^{2} \leq \frac{2}{\gamma^{2}}R^{t}+2G^{t},
$$

and

$$
-R^{t} \leq -\mu\gamma^{2}\delta^{t}+\gamma^{2}G^{t}. \qquad (112)
$$

Thus (111) can be further bounded as

$$
\begin{aligned}
\mathbb{E}\left[\Psi\right] \;=\;& \mathbb{E}\left[\delta^{t+1}+\frac{1}{\theta\lambda}G^{t+1}\right] \\
\leq\;& \mathbb{E}\left[\delta^{t}\right] + \frac{1}{\theta\lambda}\left(1-\frac{\theta}{2}\right)\mathbb{E}\left[G^{t}\right] - \frac{1}{2\gamma}\mathbb{E}\left[R^{t}\right] \\
\overset{(112)}{\leq}\;& \mathbb{E}\left[\delta^{t}\right] + \frac{1}{\theta\lambda}\left(1-\frac{\theta}{2}\right)\mathbb{E}\left[G^{t}\right] - \frac{\gamma\mu}{2}\mathbb{E}\left[\delta^{t}\right] + \frac{\gamma}{2}\mathbb{E}\left[G^{t}\right] \\
=\;& \left(1-\frac{\gamma\mu}{2}\right)\mathbb{E}\left[\delta^{t}\right] + \frac{1}{\theta\lambda}\left(1-\frac{\theta}{2}+\frac{\gamma\theta\lambda}{2}\right)\mathbb{E}\left[G^{t}\right] \\
\leq\;& \left(1-\frac{\gamma\mu}{2}\right)\mathbb{E}\left[\delta^{t}+\frac{1}{\theta\lambda}G^{t}\right], \qquad (113)
\end{aligned}
$$

where the last inequality follows by our assumption on the stepsize (the second term in minimum). It remains to unroll the recurrence. $\qquad\square$

**Corollary 18.** *Let assumptions of Theorem 13 hold,*

$$
\begin{aligned}
g_{i}^{0} \;=\;& \nabla f_{i}(x^{0}), \qquad i=1,\dots,n, \\
\gamma \;=\;& \min\left\{\left(L+2\widetilde{L}\sqrt{\frac{2\beta}{\theta}}\right)^{-1}, \; \frac{\theta}{\mu+\theta\widetilde{L}\sqrt{\frac{2\beta}{\theta}}}\right\}.
\end{aligned}
$$

*Then, after $T$ iterations/communication rounds of* EF21-Prox *we have $\mathbb{E}\left[f(x^{T})-f(x^{\star})\right]\leq\varepsilon$. It requires*

$$
T = \#grad = \mathcal{O}\left(\frac{\mu+\widetilde{L}}{\alpha\mu}\log\left(\frac{\delta^{0}}{\varepsilon}\right)\right) \qquad (114)
$$

*iterations/communications rounds/gradint computations at each node, where $\widetilde{L}=\sqrt{\frac{1}{n}\sum_{i=1}^{n}L_{i}^{2}}$, $\delta^{0}=\Phi(x^{0})-\Phi^{inf}$.*

*Proof.* Note that by Lemma 17 we have

$$
\begin{aligned}
\frac{\mu}{\theta} + \widetilde{L}\sqrt{\frac{2\beta}{\theta}} &\leq \frac{4\mu}{\alpha} + \widetilde{L}\frac{2\sqrt{2}}{\alpha} \\
&\leq \frac{4\left(\mu + \widetilde{L}\right)}{\alpha}.
\end{aligned}
$$

The remainder of the proof is the same as for Corollary 3. □

## J  USEFUL AUXILIARY RESULTS

### J.1  BASIC FACTS

For all $a, b, x_1, \ldots, x_n \in \mathbb{R}^d$, $s > 0$ and $p \in (0,1]$ the following inequalities hold

$$\langle a, b \rangle \quad \leq \quad \frac{\|a\|^2}{2s} + \frac{s\|b\|^2}{2}, \tag{115}$$

$$\langle a - b, a + b \rangle \quad = \quad \|a\|^2 - \|b\|^2, \tag{116}$$

$$\frac{1}{2}\|a\|^2 - \|b\|^2 \quad \leq \quad \|a + b\|^2, \tag{117}$$

$$\|a + b\|^2 \quad \leq \quad (1+s)\|a\|^2 + (1 + 1/s)\|b\|^2, \tag{118}$$

$$\left\| \frac{1}{n} \sum_{i=1}^{n} x_i \right\|^2 \quad \leq \quad \frac{1}{n} \sum_{i=1}^{n} \|x_i\|^2, \tag{119}$$

$$\left(1 - \frac{p}{2}\right)^{-1} \quad \leq \quad 1 + p, \tag{120}$$

$$\left(1 + \frac{p}{2}\right)(1 - p) \quad \leq \quad 1 - \frac{p}{2}, \tag{121}$$

$$\log(1 - p) \quad \leq \quad -p. \tag{122}$$

**Bias-variance decomposition**  For a random vector $\xi \in \mathbb{R}^d$ and any deterministic vector $x \in \mathbb{R}^d$, the variance of $\xi$ can be decomposed as

$$\mathbb{E}\left[\|\xi - \mathbb{E}[\xi]\|^2\right] = \mathbb{E}\left[\|\xi\|^2\right] - \|\mathbb{E}[\xi]\|^2 \tag{123}$$

**Tower property of mathematical expectation.**  For random variables $\xi, \eta \in \mathbb{R}^d$ we have

$$\mathbb{E}[\xi] = \mathbb{E}[\mathbb{E}[\xi \mid \eta]] \tag{124}$$

under assumption that all expectations in the expression above are well-defined.

### J.2  USEFUL LEMMAS

**Lemma 15** (Lemma 5 of (Richtárik et al., 2021))**.**  *If* $0 \leq \gamma \leq \frac{1}{\sqrt{a}+b}$*, then* $a\gamma^2 + b\gamma \leq 1$*. Moreover, the bound is tight up to the factor of 2 since* $\frac{1}{\sqrt{a}+b} \leq \min\left\{\frac{1}{\sqrt{a}}, \frac{1}{b}\right\} \leq \frac{2}{\sqrt{a}+b}$*.*

**Lemma 16** (Lemma 2 of (Li et al., 2021))**.**  *Suppose that function $f$ is $L$-smooth and let* $x^{t+1} \overset{def}{=} x^t - \gamma g^t$*, where $g^t \in \mathbb{R}^d$ is any vector, and $\gamma > 0$ any scalar. Then we have*

$$f(x^{t+1}) \leq f(x^t) - \frac{\gamma}{2}\left\|\nabla f(x^t)\right\|^2 - \left(\frac{1}{2\gamma} - \frac{L}{2}\right)\left\|x^{t+1} - x^t\right\|^2 + \frac{\gamma}{2}\left\|g^t - \nabla f(x^t)\right\|^2. \tag{125}$$

**Lemma 17** (Lemma 3 of (Richtárik et al., 2021))**.**  *Let $0 < \alpha < 1$ and for $s > 0$ let $\theta(s)$ and $\beta(s)$ be defined as*

$$\theta(s) \overset{def}{=} 1 - (1 - \alpha)(1 + s), \qquad \beta(s) \overset{def}{=} (1 - \alpha)(1 + s^{-1}).$$

*Then the solution of the optimization problem*

$$\min_s \left\{ \frac{\beta(s)}{\theta(s)} \ : \ 0 < s < \frac{\alpha}{1 - \alpha} \right\} \tag{126}$$

*is given by* $s^* = \frac{1}{\sqrt{1-\alpha}} - 1$*. Furthermore,* $\theta(s^*) = 1 - \sqrt{1 - \alpha}$*,* $\beta(s^*) = \frac{1-\alpha}{1-\sqrt{1-\alpha}}$ *and*

$$\sqrt{\frac{\beta(s^*)}{\theta(s^*)}} = \frac{1}{\sqrt{1-\alpha}} - 1 = \frac{1}{\alpha} + \frac{\sqrt{1-\alpha}}{\alpha} - 1 \leq \frac{2}{\alpha} - 1. \tag{127}$$

*In the trivial case $\alpha = 1$, we have $\frac{\beta(s)}{\theta(s)} = 0$ for any $s > 0$, and (127) is satisfied.*

**Lemma 18.** *Let (arbitrary scalar) non-negative sequences $\{s^t\}_{t\geq 0}$, and $\{r^t\}_{t\geq 0}$ satisfy*

$$\sum_{t=0}^{T-1} s^{t+1} \leq (1-\theta) \sum_{t=0}^{T-1} s^t + C \sum_{t=0}^{T-1} r^t$$

*for some parameters $\theta \in (0,1]$, $C > 0$. Then for all $T \geq 0$*

$$\sum_{t=0}^{T-1} s^t \leq \frac{s^0}{\theta} + \frac{C}{\theta} \sum_{t=0}^{T-1} r^t. \tag{128}$$

*Proof.* We have

$$
\begin{aligned}
\sum_{t=0}^{T-1} s^t - s^0 &\leq \sum_{t=0}^{T-1} s^t + s^T - s^0 \\
&= \sum_{t=0}^{T-1} s^{t+1} \\
&\leq (1-\theta) \sum_{t=0}^{T-1} s^t + C \sum_{t=0}^{T-1} r^t \\
&= \sum_{t=0}^{T-1} s^t - \theta \sum_{t=0}^{T-1} s^t + C \sum_{t=0}^{T-1} r^t.
\end{aligned}
$$

Dividing both sides by $\theta > 0$ and rearranging the terms, we get

$$\sum_{t=0}^{T-1} s^t \leq \frac{s^0}{\theta} + \frac{C}{\theta} \sum_{t=0}^{T-1} r^t.$$

$\square$

