# OpenReview forum: "EF21 with Bells & Whistles: Practical Algorithmic Extensions of Modern Error Feedback"
_ICLR.cc/2022/Conference — ICLR 2022 Submitted_

### Official Review · Reviewer_YjdL · 2021-10-31

**Correctness:** 3
**Technical Novelty And Significance:** 2
**Empirical Novelty And Significance:** 2
**Recommendation:** 6
**Confidence:** 4

**Main Review:**

This paper studies extended versions of an existing algorithm called EF21, and show its convergence complexity when combined with other techniques, such as momentum, variance reduction, etc. The theoretical analysis is detailed and given for each variant. The analysis on partial participation and proximal settings are new for error-feedback type algorithms.

Overall I have two concerns:

- The paper provides convergence proof for each variant, which is good, however, it's still unclear to me what role EF21 plays on these techniques. More specifically, to me this paper seems to focus on the fundamental question: how would the convergence rate of any existing algorithm adapt if an EF21 gradient oracle is used? For instance, if we want to use both stochastic approximation and momentum (which is usually the case in practice), can we obtain a rate from your analysis without additional proof?

- The experiments seem a little thin. The main experiments are conducted on small scale linear regression problems (where communication is not really an issue). The appendix provides additional results for Resnet18 on CIFAR10. However, only the first few epochs are shown and it's unclear what hypothesis is verified there (Resnet18 should reach 93% test accuracy on CIFAR10).

Some other minor concerns:

- In the abstract (and also in the paper), it's claimed that EF21-BC is better than a previous result from (Tang et al., 2020). But note that in (Tang et al., 2020), stochastic gradient is used while in  EF21-BC, full gradient is still used. So it's not clear whether EF21-BC can really improve the rate if they are compared in the same setting.

- In the comparison to ChocoSGD on page 8, the authors states $G$ usually depends on dimension while $\sigma^2$ and $\Delta$ are dimension free, which seems a little off to me. As they are all bounded on the norm, the statements on their orders are unjustified.

**Summary Of The Paper:**

This paper studies the extended versions of a previously proposed algorithm called EF21. The paper includes several variants of EF21 such as stochastic optimization, partial participation, variance reduction, momentum, etc. The theoretical analysis for each extension is given and experiments on real world datasets are conducted.

**Summary Of The Review:**

Please refer to the main review for the details.

I recommend the authors to update the manuscript to make these points clearer.

---

> ### Author Response · Authors · 2021-11-23
> **Official Response to Reviewer YjdL**
>
> We thank the reviewer for their feedback. Below we respond to all the comments by the reviewer.
>
> > The paper provides convergence proof for each variant, which is good, however, it's still unclear to me what role EF21 plays on these techniques. More specifically, to me this paper seems to focus on the fundamental question: how would the convergence rate of any existing algorithm adapt if an EF21 gradient oracle is used? For instance, if we want to use both stochastic approximation and momentum (which is usually the case in practice), can we obtain a rate from your analysis without additional proof?
>
> We thank the reviewer for raising this question. Due to the flexibility of our analysis, several of the proposed extensions can be combined together. For instance, in order to obtain the analysis for EF21-SGD-HB one needs to use the results of Lemma 2 (for EF21-SGD) instead of Lemma 1 (for EF21) and apply the steps in the proof of Theorem 11 (for EF21-HB). Similar combinations can be obtained with other methods as well. However, for the sake of readability, we do not combine EF21 with more than one technique considered in the paper.
>
> > The experiments seem a little thin. The main experiments are conducted on small scale linear regression problems (where communication is not really an issue). The appendix provides additional results for Resnet18 on CIFAR10. However, only the first few epochs are shown and it's unclear what hypothesis is verified there (Resnet18 should reach 93% test accuracy on CIFAR10).
>
> In our work, we focus on improving the theory of methods with error feedback. We conducted several numerical experiments to illustrate our theoretical findings. We believe these results can be used as a good starting point for the implementation of the proposed methods for larger problems using a more practical distributed setup. In order to strengthen our paper further, we conducted additional experiments on a larger dataset (real-sim) and added the new plots in the revised version. Regarding the experiments for Resnet18 on CIFAR10 we refer to our reply to the fourth comment of Reviewer yZfa.
>
> > In the abstract (and also in the paper), it's claimed that EF21-BC is better than a previous result from (Tang et al., 2020). But note that in (Tang et al., 2020), stochastic gradient is used while in EF21-BC, full gradient is still used. So it's not clear whether EF21-BC can really improve the rate if they are compared in the same setting.
>
> First of all, taking $\sigma = 0$ in the rate from (Tang et al., 2020), one gets the rate with worse dependence on $\varepsilon$ than we have for EF21-BC. Secondly, as we explained above, our analysis is quite flexible. In particular, we can combine EF21-BC with SGD or PAGE and compare this to (Tang et al., 2020). It is straightforward from the provided analysis to obtain the analysis for EF21-SGD-BC.
>
> > In the comparison to ChocoSGD on page 8, the authors states $G$ usually depends on dimension while $\sigma^2$ and $\Delta$ are dimension free, which seems a little off to me. As they are all bounded on the norm, the statements on their orders are unjustified.
>
> We thank the reviewer for pointing this. We do not claim that it is always the case, but when $G$ is finite (typically it is not) it has an implicit dependence on the dimension when the problem is not sparse. However, $\sigma^2$ can be dimension-independent, e.g., when the noise is small. Next, $\Delta_{\inf}$ depends on the discrepancy between lower bounds on the functional values of local functions that is in general dimension-independent.

---

### Official Review · Reviewer_yZfa · 2021-11-01

**Correctness:** 3
**Technical Novelty And Significance:** 3
**Empirical Novelty And Significance:** 1
**Recommendation:** 5
**Confidence:** 4

**Main Review:**

While the authors definitely did a lot of works, as this is a 70 page paper, the novelty of this work seems to largely depend on EF21. And the main results are obtained by combining the key results in of EF21 with existing analytical techniques such as PAGE and momentum. It is not clear from the main context what are the new analytical breakthrough introduced specifically in this work.

More experiments are needed to support the theoretical findings. In particular,

a) Comparison with other benchmarks is needed since tighter bounds in theory does not necessarily imply better empirical performance.

b) More large-scale experiments should be added. Currently, many logistic regression problems have dimension less than 300.

c) In the ResNet18 experiment, have the authors decreased the step sizes? Typically it is not hard for resnet18 to get a test accuracy more than 90%.

d) One of the goals of communication efficient methods is to make training faster. Communication rounds does not necessarily mean fast in runtime since there are also implementation overhead for (de)compression. Hence, the runtime vs accuracy is also needed in numerical results.

**Summary Of The Paper:**

This work studies communication efficient methods by extending EF21 to six more practical settings including momentum, variance reduction, etc. The main results are the convergence rates under these settings.

**Summary Of The Review:**

More discussions are needed to clarify the theoretical novelty, and numerical results should be extended.

---

> ### Author Response · Authors · 2021-11-23
> **Official Response to Reviewer yZfa**
>
> We thank the reviewer for their feedback. Below we respond to all the comments by the reviewer.
>
> > … the novelty of this work seems to largely depend on EF21. And the main results are obtained by combining the key results in of EF21 with existing analytical techniques such as PAGE and momentum. It is not clear from the main context what are the new analytical breakthrough introduced specifically in this work.
>
> We politely disagree with the evaluation of our methods. As we explain in our response to the first comment of Reviewer DvwK, our paper addresses important open questions and provides a systematic study of error feedback improving/generalizing previously known results. We emphasize that it was unclear beforehand how the known techniques should be combined in order to get the results that we derived. Error feedback is a popular tool in distributed learning with compression. Therefore, it is important to have such a detailed study (with a clean comparison with existing work) of it as our paper contains.
>
> > More experiments are needed to support the theoretical findings. In particular,
> a) Comparison with other benchmarks is needed since tighter bounds in theory does not necessarily imply better empirical performance.
>
> The goal of our numerical experiments is to illustrate our theoretical results. We do not have a goal to provide an extensive comparison with various benchmarks, since we focus on the theoretical analysis of error feedback.
>
> > b) More large-scale experiments should be added. Currently, many logistic regression problems have dimension less than 300.
>
> Although our work is mainly theoretical, we take seriously the criticism of our numerical experiments. In order to strengthen our paper further, we conducted additional experiments on a larger dataset (real-sim) and added the new plots in the revised version.
>
> > c) In the ResNet18 experiment, have the authors decreased the step sizes? Typically it is not hard for resnet18 to get a test accuracy more than 90%.
>
> We did not apply stepsize decreasing strategies, but we tuned the stepsize for each method extensively. Our main goal of the experiments on the deep learning problem is to demonstrate how does the distributed training using EF21-SGD-HB and EF21+-SGD-HB compare to EF21-SGD and other existing methods. We choose not to apply stepsize scheduling or other additional techniques for the sake of clarity of comparison. We refer to the experiments on EF with top-K compressor in [Samuel Horváth and Peter Richtárik. A better alternative to error feedback for communication efficient distributed learning. In 9th International Conference on Learning Representations (ICLR), 2021].
>
> Our experimental results are consistent with their findings in that they also achieve test accuracy just above 80% in the same setting. To achieve better test accuracy one needs to apply a more sophisticated stepsize schedule.
>
> > d) One of the goals of communication efficient methods is to make training faster. Communication rounds does not necessarily mean fast in runtime since there are also implementation overhead for (de)compression. Hence, the runtime vs accuracy is also needed in numerical results.
>
> We do not have runtime vs accuracy comparison, because such a comparison is infrastructure-dependent and, therefore, is not fair. Moreover, the mentioned compression/decompression overhead appears in all methods with compression, so, there is no need to take it into account when we compare methods with the same compression as we do in our experiments.

---

### Official Review · Reviewer_2qZ7 · 2021-11-02

**Correctness:** 4
**Technical Novelty And Significance:** 2
**Empirical Novelty And Significance:** 2
**Recommendation:** 5
**Confidence:** 3

**Main Review:**

The paper is well-written. The convergence analysis and experiments shows that the proposed algorithms work well in both theory and practice.
However, I have the following concerns:
1. All the proposed algorithms are extensions of EF21, the overall modification seems minor to me, which weakens the contribution of this paper.
2. Most importantly, I hardly find the experiment results convincing due to the experiment settings. Most of the datasets and models are small and simple, which typically does not require distributed training (even being trained on CPUs), not to mention communication compression. The results on such simple and small learning problems can barely show the significance of any distributed training algorithm with communication compression.
3. Even for the largest problems: Resnet on cifar10, is usually considered the smallest one in the distributed scenarios.  On cifar10, although the proposed algorithms shows better test accuracy than the baselines, the baselines have better training loss. Thus, it is hard to justify which algorithm is actually better. Such phenomena is also not explained in the paper.
4. A very important baseline is missing in the experiments: SGD without communication compression, which could be compared to the proposed algorithms in the number of epochs. Without such a baseline, it is hard to justify the performance of the proposed algorithms.

**Summary Of The Paper:**

In this paper, the authors propose six practical extensions of EF21, with convergence analysis and good experiment results.

**Summary Of The Review:**

The paper is well-written, and good in theory and practice.
However, I have some concerns in the contribution and the experiment settings.

---

> ### Author Response · Authors · 2021-11-23
> **Official Response to Reviewer 2qZ7**
>
> We thank the reviewer for their feedback. Below we respond to all the comments by the reviewer.
>
> > 1. All the proposed algorithms are extensions of EF21, the overall modification seems minor to me, which weakens the contribution of this paper.
>
> We politely disagree with the evaluation of our methods. As we explain in our response to the first comment of Reviewer DvwK, our paper addresses important open questions and provides a systematic study of error feedback improving/generalizing previously known results. Error feedback is a popular tool in distributed learning with compression. Therefore, it is important to have such a detailed study (with a clean comparison with existing work) of it as our paper contains.
>
> > 2. Most importantly, I hardly find the experiment results convincing due to the experiment settings. Most of the datasets and models are small and simple, which typically does not require distributed training (even being trained on CPUs), not to mention communication compression. The results on such simple and small learning problems can barely show the significance of any distributed training algorithm with communication compression.
>
> In our work, we focus on improving the theory of methods with error feedback. We conducted several numerical experiments to illustrate our theoretical findings. We believe these results can be used as a good starting point for the implementation of the proposed methods for larger problems using a more practical distributed setup.
>
> > 3. Even for the largest problems: Resnet on cifar10, is usually considered the smallest one in the distributed scenarios. On cifar10, although the proposed algorithms shows better test accuracy than the baselines, the baselines have better training loss. Thus, it is hard to justify which algorithm is actually better. Such phenomena is also not explained in the paper.
>
> Such phenomena (worse train loss but better test accuracy) are quite common for not over-parameterized problems. Indeed, good performance on the train data does not imply good performance in the test data.
>
> > 4. A very important baseline is missing in the experiments: SGD without communication compression, which could be compared to the proposed algorithms in the number of epochs. Without such a baseline, it is hard to justify the performance of the proposed algorithms.
>
> We thank the reviewer for the suggestion. We have added a comparison with non-compressed SGD in the revised version in terms of the sent bits from clients to the server (see Figures 7 and 8). Since we focus on reducing the number of transmitted information to achieve the desired accuracy of the solution, we do not provide a comparison in terms of the epochs.

---

### Official Review · Reviewer_DvwK · 2021-11-08

**Correctness:** 3
**Technical Novelty And Significance:** 2
**Empirical Novelty And Significance:** 2
**Recommendation:** 5
**Confidence:** 4

**Main Review:**

I appreciate the authors' efforts for analyzing EF21 for various settings and provide their convergence results as well as the experimental results. However, it is not clear to me what challenge the paper addresses and novelty the paper presents. All the six variants including stochastic gradient, variance reduction, heavy momentum, partial participation, bidirectional compression, and proximal gradient have been extensively studied in the literature. Merely combining them with EF21 and providing theoretical analyses do not seems novel to me.

EF has worse error bound than the full-precision counterpart when the full gradient is used. However, it has the same asymptotic bound with stochastic gradient. How does EF21-SGD compare to EF-SGD? Does EF21 still yield better bound when stochastic gradient is adopted?

The aforementioned question is from theoretical side. From empirical side, Figure 7 in the appendix shows that both EF-SGD and EF-SGD-HB converge faster than EF21-SGD and EF21-SGD-HB, respectively. Only the hybrid method EF21+-SGD-HB obtains higher testing accuracy. These seems to contradict the claim that EF21 is better than EF for the stochastic setting. How do we explain these findings?

Some minor comments:
1. It is clearer to split the updates in Table 2 to worker's and server's, which I think are more friendly to the readers.
2. $L_j$ in Example 2 should be $L_{ij}$?

**Summary Of The Paper:**

Error feedback (EF) is a technique for ensuring convergence of biased contractive compressor. However, it achieves suboptimal convergence rate when full gradient is used. Recently, EF21 was introduced to mitigate the theoretical deficiencies of EF. This paper studies several extensions of EF21 including EF21 with stochastic gradient, variance reduction, heavy momentum, partial participation, bidirectional compression, and proximal gradient. The paper establishes the convergence results for the six variants and conducts experiments on several small datasets.

**Summary Of The Review:**

This paper provides solid theoretical analysis for several EF21 variants. But the novelty is limited.

---

> ### Author Response · Authors · 2021-11-23
> **Official Response to Reviewer DvwK [Part 2/2]**
>
> > 1. It is clearer to split the updates in Table 2 to worker's and server's, which I think are more friendly to the readers.
>
> We thank the reviewer for the suggestion: indeed if we had more space, we would make the table better this way. However, taking into account the space limitations, we did our best in making the table short and reader-friendly.
>
> > 2. $L_{j}$ in Example 2 should be $L_{ij}$?
>
> Thank you for noticing the typo in Example 2. We fixed it in the revised version.

---

> ### Author Response · Authors · 2021-11-23
> **Official Response to Reviewer DvwK [Part 1/2]**
>
> We thank the reviewer for their feedback. Below we respond to all the comments by the reviewer.
>
> > However, it is not clear to me what challenge the paper addresses and novelty the paper presents. All the six variants including stochastic gradient, variance reduction, heavy momentum, partial participation, bidirectional compression, and proximal gradient have been extensively studied in the literature. Merely combining them with EF21 and providing theoretical analyses do not seems novel to me.
>
> Indeed, each of the six techniques (extensions of GD) are well established in optimization and federated learning, they have been extensively analyzed due to their high importance for both theory and practice. Richtarik et. al (2021) propose and study EF21 mechanism, however, they present and analyze it in pure form. The paper leaves an important open question of whether the proposed mechanism is flexible enough to be combined with other techniques in federated learning. In particular, we addressed the following questions: How these combinations should be implemented? Can one obtain strong convergence results for these methods without additional restrictive assumptions? Our paper answers all these questions in the affirmative and presents a clean analysis and the rates. We emphasize that it is NOT always the case that the combination of different techniques provably works in optimization (for instance, it is not all apparent whether or not the original EF can be combined with bidirectional compression, partial participation, or variance reduction under the same assumptions as in the analysis of non-compressed methods). The theoretical investigation that we propose in our work is an important contribution to the field of error feedback. It shows that EF21 is not just a standalone technique but rather a natural way to design a distributed training along with other popular techniques. Moreover, it shows further theoretical advantages of EF21 over EF since it is not known whether EF can be combined with all the above extensions without restrictive assumptions.
>
> > EF has worse error bound than the full-precision counterpart when the full gradient is used. However, it has the same asymptotic bound with stochastic gradient. How does EF21-SGD compare to EF-SGD? Does EF21 still yield better bound when stochastic gradient is adopted?
>
> To the best of our knowledge, the existing results of EF-SGD rely on a strong assumption the gradients are bounded (see the “Comment” column in Table 1 for Choco-SGD). This is a restrictive assumption since, for example, a simple quadratic function does not satisfy this property. Richtarik et. al (2021) derive the result under a weaker assumption that the variance of the stochastic gradient is bounded. We prove the result under a much more general assumption (Assumption 2) than in any analysis of EF, therefore, it is not trivial to compare the asymptotic bound of EF-SGD and EF21-SGD. In particular, although our rate has worse dependence on the number of workers $n$ than the one derived for Choco-SGD, the quantity $G$ is often infinite meaning that the rate is not well-defined in these cases. In contrast, our rates are well-defined for much wider class of problems.
>
> > From empirical side, Figure 7 in the appendix shows that both EF-SGD and EF-SGD-HB converge faster than EF21-SGD and EF21-SGD-HB, respectively. Only the hybrid method EF21+-SGD-HB obtains higher testing accuracy. These seems to contradict the claim that EF21 is better than EF for the stochastic setting. How do we explain these findings?
>
> In our paper, we do not claim that the methods with EF21 are always better than previously known methods with EF. What we claim is that for EF21 we prove better theoretical results than previously best-known ones. So, there is no contradiction here. Regarding the DL experiments in Figure 7, we write: *“momentum methods show a considerable improvement in the accuracy score on the test set over the existing EF21-SGD and EF-SGD.”* We do not claim that EF21 is better than EF in a stochastic setting. We mean that the methods with momentum (EF21-SGD-HB, EF21+-SGD-HB, and EF-SGD-HB) show better test accuracy results than EF21-SGD, EF-SGD (non-momentum methods, that was introduced and compared for the same DL problems in the original EF21 paper).
>
> Moreover, our empirical findings imply that, as with many other methods, EF21-SGD-HB should be used properly, and, in practice, it is worth applying some heuristic, e.g., EF21+, to get better performance for a particular problem. This phenomenon does not diminish our theoretical results,  but rather gives useful information for the practical use of the method.

---

### Author Response · Authors · 2021-11-23
**General Response to Reviewers**

We thank the reviewers for their time and efforts to study our paper and for their feedback.

We appreciate the reviewers acknowledging that our analysis is solid (Reviewer DvwK), our algorithms *“work well in both theory and practice”*, our paper is well-written (Reviewer 2qZ7), our analysis is detailed, and the combinations of error feedback with partial participation and proximal setting are novel in the field of error feedback methods (Reviewer YjdL).

We also added the replies to all the reviewers’ questions, comments, and concerns. Following the reviewers’ requests, we conducted additional experiments on logistic regression with a larger dataset (real-sim dataset from LIBSVM) and provided a numerical comparison of EF21-SGD and EF21-PAGE with non-compressed SGD and PAGE. New experimental results are given in Appendix A.1. All changes are highlighted in blue color.

---

### Author Response · Authors · 2021-11-28
**Feedback request**

We thank the reviewers once again for their reviews. In our responses, we addressed all concerns of the reviewers.

Considering that the discussion period ends tomorrow, we kindly ask the reviewers to let us know whether our replies are convincing and whether additional clarifications are required. Furthermore, we would be happy to address new questions and criticism in case of any.

We thank the reviewers in advance.

---

### Author Response · Authors · 2021-12-01
**To all reviewers: We feel our paper and results were not understood and hence not appreciated**

Dear reviewers,

Thanks for your valuable comments. **We feel our paper and our results were not understood and hence not appreciated. We propose several new and practical enhancements of the newly proposed EF21 error feedback method, obtaining state-of-the-art theoretical results for the error feedback mechanism.**

**Is theoretical SOTA not appreciated by the ICLR community anymore? Please study our Table 1.** Several of the proposed variants ("bells and whistles") of error feedback were not considered in the literature before because this was hard or impossible to do before the new analysis approach pioneered by Richtarik et al in their EF21 paper. In some cases, similar variants existed, but our rates are the new SOTA.
Some examples:

- Prior theoretical best rate for error feedback with compression also at the master (which we call bidirectional compression, or BC) was $O(1/\epsilon^4)$ for the double squeeze method. Their rate does not apply to quantization nor sparsification operators commonly used in practice since they do not satisfy the assumption $\mathbb{E}||C(x)-x|| \leq \Delta$. Our results apply to the commonly used contractive compression operators, and we also get the improved rate $O(1/\epsilon^2)$. **We believe that this alone is a result publishable in ICLR.**  Moreover, Tang et al do not obtain any rate in the PL setting, we do.

- Prior to our work, to the best of our knowledge, there was no analysis of error feedback with a proximal operator. We obtain the first results in this setting. This alone is an answer to an open problem since 2014 when error feedback was first proposed by Seide et al, and **we believe that this alone is a result publishable in ICLR.**

- Prior to our work, the best rate for error feedback with the very popular momentum mechanism was obtained by Xie et al (2020). They obtained a $O(G/\epsilon^3)$ rate, where $G$ is a uniform bound on the gradients. However, this is a strong assumption since gradients of many functions used in practice are not uniformly bounded (e.g., quadratics). Methods such as gradient descent and Nesterov momentum are normally analyzed without the need for such an assumption. We resolve this problem: we do not need this assumption. Moreover, our rate is better by an order of magnitude: $O(1/\epsilon^2)$. We proved better rates for PL functions. **We believe that this alone is a result publishable in ICLR.**

- Prior to our work, no EF mechanism was analyzed in the important-to-federated learning *partial participation regime*. This is because previous analysis techniques were not good enough to allow for such an analysis. Our method EF21-PP is the first error feedback mechanism provably combinable with partial participation! Since communication compression, error feedback and partial participation are of key importance to modern federated learning, we resolve an important open problem in the field. Again, we obtain an even better rate in the PL regime. **We believe that this alone is a result publishable in ICLR.**

- Our rate for EF21-PAGE is a massive improvement on the best error feedback results with minibatching (=subsampling) on the clients. In modern machine learning, each client/machine owns many training datapoints, and it is customary to rely on stochastic gradients rather than full gradients. So, it is very important to study this problem. In EF21-PAGE  we managed to combine (and this was not trivial) he EF21 method with the optimal PAGE estimator of Li et al which allows for optimal variance reduced subsampling/minbatching on the clients. Our rate $O(1/\epsilon^2)$ is a massive improvement on the previous best rate for error feedback with stochastic gradients due to Koloskova et al (2020) and Richtarik et al (2021), which were both $O(1/\epsilon^4)$. Again, this resolves an important problem in modern distributed and federated learning. **We believe that this alone is a result publishable in ICLR.**

**In summary, our work contains several independent contributions each of which we deem to be publishable in a top ML venue as each resolves an important theoretical problem in the area of modern communication efficient distributed training.**

**We do not understand why the reviewers do not appreciate such contributions.** Some reviewers have the impression that combining techniques is easy and not novel enough. This can't be further from the truth. If this was the case, error feedback would already have been combined with all the techniques we propose here, and the improved complexity results would have been obtained. There are hundreds of examples in the field where it is not known whether certain  techniques, understood in isolation, are combinable in the sense that the individual benefits they each provide combine for an even greater benefit. For example, we do not know whether adaptive stepsizes for gradient descent (proposed by Mishchenko and Malitsky) can be combined with stochastic approximation.

Authors

---

### Decision · Program_Chairs · 2022-01-20

**Decision:**

Reject

**Comment:**

This paper presents several variants and extensions (including stochastic and proximal) of the error-feedback method EF21 and provides convergence rates for each of them and shows that they improve upon previous state of the arts. Despite the much broadened application scenarios and SOTA  in convergence rates/complexity, the main and common concern from the reviewers is the novelty of the paper beyond the original EF21 work. There are also concerns on the empirical evaluations that do not fully support the theoretical promises. I agree with the reviewers and regrettably have to recommend rejection for ICLR.